# Actively Identifying Causal Effects with Latent Variables Given Only Response Variable Observable

**Tian-Zuo Wang**[1,2] **and Zhi-Hua Zhou**[1]
[1] National Key Laboratory for Novel Software Technology,
Nanjing University, Nanjing, 210023, China,
[2] Pazhou Lab, Guangzhou, 510330, China.
{wangtz, zhouzh}@lamda.nju.edu.cn

## Abstract

In many real tasks, it is generally desired to study the causal effect on a specific target (response variable) only, with no need to identify the thorough causal effects involving all variables. In this paper, we attempt to identify such effects by a few active interventions where only the response variable is observable. This task is challenging because the causal graph is unknown and even there may exist latent confounders. To learn the necessary structure for identifying the effects, we provide the graphical characterization that allows us to efficiently estimate all possible causal effects in a partially mixed ancestral graph (PMAG) by generalized back-door criterion. The characterization guides learning a local structure with the interventional data. Theoretical analysis and empirical studies validate the effectiveness and efficiency of our proposed approach.

## 1 Introduction

Identifying causal effects is one prominent task throughout empirical sciences. In many real problems, we are generally desired to study the causal effect on a specific target only, i.e. *response variable*, with no need to identify the thorough causal effects involving all variables. For example, a business person wants to maximize profit. The person has many intervention methods such as lowering price, increasing advertisement investment, improving quality, and so on. A valid approach to guide the decisions is to estimate the causal effect of each variable on the response variable. We call it *target effect identification* in this paper. After identifying all of these causal effects, the person could see which intervention could lead to the most desired response (profit).

To achieve target effect identification, an ideal method is to intervene on each decision variable with different attainable values and collect the interventional data of the response variable. However, when there are many decision variables and many attainable values for each variable, it will take enormous number of interventions, which is economically unfeasible. Considering that there is usually a large amount of observational data in reality, we hope to exploit mainly the observational data to achieve it.

The main obstacle here is that the causal graph of all involved variables is not available, which makes causal effect identification intractable. Hence, we learn the structure at first. As shown by Verma and Pearl [1], only a Markov equivalence class could be learned with observational data without further assumptions, where there are possibly unidentified causal relations resulting in unidentified causal effects. To reveal these effects, we *further* learn the causal relations in the Markov equivalence class by introducing a small amount of *active* hard interventions, i.e., we force some variable to a known value [2, 3, 4]. In practice, generally one can only observe part of variables under interventions rather than all of those, e.g., a business person can observe his profit, but does not know other companies' profit. To make the method applicable in a broader scenario, we consider an extreme situation that only the response variable is observable, which could be trivially extended to any settings with

35th Conference on Neural Information Processing Systems (NeurIPS 2021).

part of variables observed. In addition, there are usually latent confounders in practical problems. For instance, economic level influences both price and advertising cost, but it is hard to evaluate accurately and thus a latent confounder. In summary, in this paper we learn the causal knowledge where there are latent confounders and only the response variable is observable under interventions.

This problem is very challenging, because the fact that only the response variable is observable disables the approaches designed for situations where all variables are observable [2, 5, 6], while the existence of latent confounders disables the typical solutions [7]. To get a brief understanding of the challenges taken by latent confounders in techniques, consider a graph learned with observational data by traditional causal discovery methods [8]: If there is no latent confounder, the definite adjacent edges of intervened variable $X$ lead to a definite interventional distribution of the response variable $Y$ by back-door criterion [9], which makes it possible to learn the edges of $X$ by the interventional data of $Y$; when latent confounders exist, however, the mapping no longer holds. Besides, a *partial mixed ancestral graph (PMAG)* is commonly used to describe the relation between variables in this condition. The back-door criterion may fail in this case since there possibly exist latent confounders influencing the two variables of some directed edges. Hence, we also need to identify whether such latent confounders exist behind the directed edges, in addition to just learning the causal edges.

To overcome the difficulties above, we provide a graphical characterization that allows us to efficiently find all possible causal effects of the intervened variable on the response variable in a partial mixed ancestral graph. Guided by the characterization, we learn a local structure and identify the presence of latent confounders behind some directed edges by each intervention, which leads to identifying some causal effects on the response variable. After a few active interventions on different variables, our method **ACIC**, short for **AC**tive target effect **I**dentification with latent **C**onfounding, could achieve target effect identification. Due to space limit, we present related work and all the proofs in Appendix.

## 2   Preliminary

We guide readers to Appendix A.1 for the notions including *causal graph* ($G = (\mathbf{V}, \mathbf{E})$), *mixed graph*, *mark*, *arrowhead*, *tail*, *circle*($\circ$), *partial mixed graph*, *parent*, *child*, *spouse*, *possible directed path*, *(possible) ancestor*, $\text{PossAn}(X, G)$, $\text{An}(X, G)$, *(possible) descendant*, $\text{PossDe}(X, G)$, $\text{De}(X, G)$, *adjacent*, $\text{Adj}(X, G)$, *almost directed cycle*, *collider path*, *minimal path*, *ancestral graph*, *m-separation (m-connecting or active)*, *maximum ancestral graph (MAG, $\mathcal{M}$)*, *discriminating path*, *Markov equivalent*, *Markov equivalence class (MEC)*, *partial ancestral graph (PAG)*, *visible*, $\mathcal{M}_{\underline{X}}$, $\mathcal{P}_{\underline{X}}$, *causal effect* ($P(Y|do(X))$), and the common assumptions including *positivity* and *no selection bias*. $*$ is a wildcard that denotes any of the marks. For a partial mixed graph, we say it is a *partial mixed ancestral graph (PMAG)* if there is no directed or almost directed cycle, and denote it by $\mathcal{P}$. Note *both MAG and PAG are special cases of PMAG*, thus $\mathcal{P}$ could also denote PAG. The relation between PMAG and PAG is similar to that between partial directed acyclic graph (PDAG) and completed partial directed acyclic graph (CPDAG) when there is no latent confounder [10]. MAG $\mathcal{M}$ is *consistent to PMAG* $\mathcal{P}$ if it belongs to the MEC represented by $\mathcal{P}$ (detailed in Appendix A.1).

For a directed edge $V_i \to V_j$ in a graph, we say the edge is *pure* if the two variables are not influenced by common *latent confounders*, denoted by $V_i \xrightarrow{p} V_j$. We use a *purity matrix* to characterize the purity of each edge in a PMAG $G$, in which 1 is in $(i, j)$ entry *only if* the directed edge $V_i \to V_j$ in $G$ is pure, and 0 otherwise. Note that 0 *does not* imply that the edge is impure. It is also possible that we are unaware of whether it is pure or the edge is not directed. In the literature, a graphical characterization to imply that there are no latent confounders behind a directed edge in an MAG or PAG is proposed by Zhang [11], where they call it by *visibility*. Visibility is sufficient but not necessary for purity, which is detailed in Appendix A.2. Thus, given a PMAG, we initialize the purity matrix by setting $(i, j)$ entry to 1 if $V_i \to V_j$ is visible and 0 otherwise. For an MAG $\mathcal{M}$, $\mathcal{M}_{\underline{X}}$ denotes the graph by removing the directed edges out of $X$ in $\mathcal{M}$ that are labeled to be pure in purity matrix. For a PMAG $\mathcal{P}$, see Appendix A.4 for the definition of $\mathcal{P}_{\underline{X}}$.

Since the graph is continuously learned by interventions in this paper, the graph in the process is a PMAG. Hence we present Prop. 1 to guide causal effect identification in a PMAG with the consideration of purity matrix rather than only in a PAG or MAG. It is based on the generalized back-door criterion (GBC) and the graphical condition for the causal effect identifiability by GBC in MAG or PAG proposed by Maathuis et al. [12]. Before that, we introduce D-SEP($X, Y, G$) in Def. 1.

**Definition 1** (D-SEP$(X, Y, G)$ [12])**.** Let $X$ and $Y$ be two distinct vertices in a mixed graph $G$. We say that $V \in$ D-SEP$(X, Y, G)$ if $V \neq X$, and there is a collider path between $X$ and $V$ in $G$, such that every vertex on this path (including $V$) is an ancestor of $X$ or $Y$ in $G$.

**Proposition 1.** *Let $G$ be a PMAG and $W$ be a purity matrix of $G$. Suppose $X \in$ An$(Y, G)$ and $Y$ are two distinct vertices in $G$. There exists a generalized back-door set relative to $(X, Y)$ and $(G, W)$ if and only if D-SEP$(X, Y, G_{\underline{X}}) \cap$ PossDe$(X, G) = \emptyset$. Moreover, if the set exists, D-SEP$(X, Y, G_{\underline{X}})$ is such a set. Denote D-SEP$(X, Y, G_{\underline{X}})$ by $\mathbf{D}$, the causal effect is*

$$P(Y|do(X = x)) = \int_{\mathbf{D}} P(\mathbf{D})P(Y|\mathbf{D}, X = x) \, \mathrm{d}\mathbf{D}. \tag{1}$$

There are two possible unidentifiable cases for $P(Y|do(X))$ by GBC in Prop. 1. One is that there are many possible causal effects due to the missing of exact structure information, but we do not know which is correct. In this case, we could address it by learning the structure with interventional data. The other unidentifiable case is that the causal effect is unidentifiable by GBC *even if we know the MAG and the purity of each edge*. We return "Fail" for such $X$ because GBC is not sufficient for identifying the causal effect in this case, and we say *GBC fails* ( to identify $P(Y|do(X))$).

## 3 The Proposed Approach

In this paper, we assume faithfulness, positivity, and no selection bias. Denote the decision variables by $X_1, \cdots, X_p$ and the response variable by $Y$. Given the observational data of these variables, our goal is to achieve target effect identification by generalized back-door criterion, i.e., we aim to identify $P(Y|do(X_i))$ by GBC for each variable $X_i, i = 1, 2, \cdots, p$ if GBC does not fail. With the observational data, we can learn a PAG $\mathcal{P}$ that the true MAG is consistent to by FCI algorithm [8]. And we initialize a purity matrix by setting $(i, j)$ entry to 1 if the directed edge between the two variables is visible in $\mathcal{P}$ and 0 otherwise. The causal effects of some variables on $Y$ are possibly unidentifiable in $\mathcal{P}$ by Prop. 1. To identify these effects, we further introduce *active* interventions and observe $Y$ under those. Beginning from $\mathcal{P}$, in each round our method selects one variable $X$ from $X_i, i = 1, \cdots, p$ to intervene and exploits the interventional data of $Y$ to learn the structure, including learning an updated PMAG by revealing circles and updating the purity matrix. The process repeats until identifying all effects $P(Y|do(X_i)), i = 1, 2, \cdots, p$ (identifying failing of GBC is also included). Since it is recursive, we just present the method to learn the structure by interventional data in one round. The criterion to select the intervention variable in each round is given at the end.

### 3.1 Two direct methods

A naive method to learn the structure by interventional data is enumerating each MAG $\mathcal{M}$ consistent to the PMAG $\mathcal{P}$. For each $\mathcal{M}$, $P(Y|do(X))$ can be estimated with observational data by (1) and we judge whether the estimated causal effect is consistent with the interventional data. By such judgment, we could rule out the MAGs with inconsistent causal effects. Yet, this method is usually impractical since it takes a huge computation complexity mainly from two parts. One part is the exhaustive search in the space of MAGs. As is known to all, the space of MAGs is extremely large. For each searched MAG, we also need to judge whether it is consistent to $\mathcal{P}$[1]. The other costly part is looking for D-SEP$(X, Y, \mathcal{M}_{\underline{X}})$ according to Def. 1 to estimate causal effects by (1) for each MAG.

A cleverer approach instead of enumerating is to directly learn a local structure by interventional data, inspired by ACI proposed by Wang et al. [7]. ACI tackles a similar task but assumes no latent confounders. The core of this approach is to find an equivalent condition that *could be obtained based on a local structure* for causal effect. In this way, there is a bijection between the condition and $P(Y|do(X))$, by which we could learn a local structure with the interventional data of $Y$. Specifically, ACI is comprised of three steps: (a) propose Minimal Parental back-door admissible Set (MPS), a variable set that could be obtained by only the orientation of adjacent edges of $X$, as the equivalent condition for $P(Y|do(X))$; (b) find all possible MPSs in the partial graph to be learned; (c) identify which MPS is correct by interventional data and learn the adjacent edges of $X$ implied by the MPS.

---

[1]For such judgment, we could first obtain an MAG based on $\mathcal{P}$ by the procedure of Theorem 2 by Zhang [13], as a representative of $\mathcal{P}$. Then we judge whether the searched MAG is Markov equivalent to this MAG by the necessary and sufficient condition for Markov equivalence by Ali et al. [14], Hu and Evans [15].

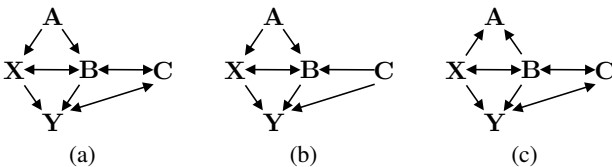

Figure 1: Suppose $X \to Y$ is pure. The adjacent edges of $X$ are the same in (a) and (b), but the causal effects are different due to the different marks at $C$. In addition, although D-SEP$(X, Y, \mathcal{M}_{\underset{\sim}{X}})$ are different in (a) and (c), the causal effects of $X$ on $Y$ are the same by (1), since $A \perp Y \mid X, B$.

However, ACI fails if there are latent confounders. In this situation the adjacent edges of $X$ is not sufficient for identifying causal effects. According to Prop. 1, even though there are the same adjacent edges of $X$ in two MAGs, the causal effects of $X$ on $Y$ are possibly different if they have different D-SEP$(X, Y, \mathcal{M}_{\underset{\sim}{X}})$. See Fig. 1(a) and Fig. 1(b) for example. The method of Malinsky and Spirtes [16] is possibly useful here, which estimates all possible causal effects in $\mathcal{P}$ by enumerating the structures in a relatively local region, but the two parts of computational costs above are still large.

Referring to the framework of ACI, we propose the three steps for our setting in the next three sections. Firstly, we propose MCS, as an *equivalent condition for $P(Y|do(X))$ in MAG by GBC* in Sec. 3.2. Secondly, we present how to *find all MCSs in $\mathcal{P}$ by graphical characterization* in Sec. 3.3. It is the main challenge here in contrast to ACI. ACI achieves it by considering all possible orientations of adjacent edges of $X$ due to the fact that their proposed MPS merely depends on these edges. In our setting, however, it is unclear which edges MCS depends on. And obviously, enumerating all edges is not efficient. Hence we propose a graphical characterization to guide it. Lastly, we give the method to *learn marks and purity matrix by interventional data* in Sec. 3.4 with all MCSs found before.

### 3.2  The equivalent condition for causal effects

By Prop. 1, D-SEP$(X, Y, \mathcal{M}_{\underset{\sim}{X}})$ can be as the adjustment set to identify $P(Y|do(X))$ when GBC does not fail. Thus, we construct the equivalent condition for $P(Y|do(X))$ based on D-SEP$(X, Y, \mathcal{M}_{\underset{\sim}{X}})$. Note that D-SEP$(X, Y, \mathcal{M}_{\underset{\sim}{X}})$ cannot be used directly since the causal effects are possibly equal in the MAGs with different D-SEP$(X, Y, \mathcal{M}_{\underset{\sim}{X}})$. See Fig. 1 for an example. The D-SEP$(X, Y, \mathcal{M}_{\underset{\sim}{X}})$ in (a) and (c) are $\{A, B\}$ and $\{B\}$, respectively, but $P(Y|do(X))$ are equal. The reason is $A \perp Y|X, B$ in the two MAGs. When calculating (1), the results are equal after integration on the generalized back-door set. To formalize this situation, we introduce *Minimal Conditional Set (MCS) regarding* D-SEP$(X, Y, \mathcal{M}_{\underset{\sim}{X}})$ *in* $\mathcal{M}$. The algorithm to find MCS and the related properties are in Appendix B.

**Definition 2** (Minimal conditional set regarding D-SEP$(X, Y, \mathcal{M}_{\underset{\sim}{X}})$ (MCS))**.** Let $\mathcal{M}$ be an MAG. $X, Y$, and $\mathbf{D}$ denote the intervened variable, response variable, and the set D-SEP$(X, Y, \mathcal{M}_{\underset{\sim}{X}})$ in $\mathcal{M}$, respectively. $\overline{\mathbf{D}}$ is a subset of $\mathbf{D}$. $\overline{\mathbf{D}}$ is a minimal conditional set regarding $\mathbf{D}$ in $\mathcal{M}$ if

(1) $(Y \perp \mathbf{D}\backslash\overline{\mathbf{D}} \,|\overline{\mathbf{D}} \cup \{X\})_{\mathcal{M}}$,

(2) $(Y \not\perp \mathbf{D}\backslash\overline{\mathbf{D}}'|\overline{\mathbf{D}}' \cup \{X\})_{\mathcal{M}}$, for any $\overline{\mathbf{D}}' \subset \overline{\mathbf{D}}$.

We call it MCS regarding D-SEP$(X, Y, \mathcal{M}_{\underset{\sim}{X}})$ or MCS for short. Note that MCS is essentially a set of variables. For all MAGs $\mathcal{M}$ satisfying the conditions of Prop. 1 with the same $\overline{\mathbf{D}}$, where $\overline{\mathbf{D}}$ denotes the MCS regarding D-SEP$(X, Y, \mathcal{M}_{\underset{\sim}{X}})$ in respective $\mathcal{M}$, they have the same causal effect as

$$P(Y|do(X = x)) = \int_{\overline{\mathbf{D}}} P(\overline{\mathbf{D}})P(Y|\overline{\mathbf{D}}, X = x)\,\mathrm{d}\overline{\mathbf{D}}. \tag{2}$$

Since structure inference with interventional data is involved, we make an additional assumption as Assumption 1 to match interventional data to causal graph, playing as the role of faithfulness given only observational data or some intervention-related assumptions[17, 7, 6, 18]. The purpose is to avoid that different causal effects in general (e.g. the causal effects $P(Y|do(X))$ in Fig. 1(a) and Fig. 1(b) ) *happen to* be equal given some *specific* observational distributions. Under the assumption,

there is a bijection between MCS and $P(Y|do(X))$ through (2). Finding all possible causal effects could be converted to finding all possible MCSs. In the next section, we present how to find all possible MCSs in a PMAG by our proposed graphical characterization.

**Assumption 1.** *For two Markov equivalent MAGs $\mathcal{M}^1$ and $\mathcal{M}^2$ with the same observational distribution, if there are different minimal conditional sets regarding* D-SEP$(X, Y, \mathcal{M}^1_{\underset{\sim}{X}})$ *and* D-SEP$(X, Y, \mathcal{M}^2_{\underset{\sim}{X}})$ *in the respective graphs, then $P\big(Y|do(X=x)\big)$ are different in the two MAGs.*

### 3.3 Finding all MCSs in $\mathcal{P}$ by graphical characterization

At the beginning, we provide an outline of finding possible MCSs regarding D-SEP$(X, Y, \mathcal{M}_{\underset{\sim}{X}})$ in all $\mathcal{M}$ consistent to $\mathcal{P}$ if GBC does not fail to identify $P(Y|do(X))$ in $\mathcal{M}$. The justification for the restriction on $\mathcal{M}$ is that if $\mathcal{M}$ is the true MAG and GBC fails to identify $P(Y|do(X))$ in $\mathcal{M}$, there is no MCS matched to the interventional data. We thus cannot identify an MCS by the intervention. In this case it is unnecessary to find MCS in such $\mathcal{M}$. Hence, we consider the MAG $\mathcal{M}$ that satisfies D-SEP$(X, Y, \mathcal{M}_{\underset{\sim}{X}}) \cap$ De$(X, \mathcal{M}) = \emptyset$, according to Prop. 1. There are three main parts to achieve it. The first part is to enumerate all mark combinations of $X$. The true MAG is with one mark combination among them. As mentioned earlier, the marks of $X$ are not sufficient for identifying $P(Y|do(X))$. In another word, MCSs could be different in the MAGs with the same marks at $X$. Hence, in the second part we propose the graphical condition to indicate when there are different MCSs in the MAGs with the same marks at $X$. And this graphical condition can be obtained in a partial graph. Based on that, we present the algorithm to find all possible MCSs in $\mathcal{P}$ in the third part.

We first enumerate all possible mark combinations of $X$. To denote the PMAGs with deterministic marks at $X$, we introduce *local MAG of $X$ based on $\mathcal{P}$* in Def. 3. For a local MAG $M$ and an MAG $\mathcal{M}$ consistent to $M$, we then present a sufficient and necessary condition for $V \in$ D-SEP$(X, Y, \mathcal{M}_{\underset{\sim}{X}})$ in Thm. 2, which plays a vital role for the following result involving MCS regarding D-SEP$(X, Y, \mathcal{M}_{\underset{\sim}{X}})$.

**Definition 3** (Local MAG of $X$ based on $\mathcal{P}$). Given a PMAG $\mathcal{P}$ and a variable $X$, a PMAG $M$ is a *local MAG of $X$ based on $\mathcal{P}$* if (1) $M$ is with definite marks (arrowheads or tails) at $X$; (2) $M$ is obtained from $\mathcal{P}$ by marking some circles without generating new unshielded colliders or directed or almost directed cycles. We call it *local MAG* for short if there is no ambiguity and denote it by $M$, which is different from calligraphic $\mathcal{M}$ that denotes MAG.

**Theorem 2.** Let $M$ be a local MAG of $X$ and $\mathcal{M}$ be an MAG consistent to $M$. Suppose $V$ ($V \neq X, Y$) is a variable in $\mathcal{M}$. If D-SEP$(X, Y, \mathcal{M}_{\underset{\sim}{X}}) \cap$ De$(X, \mathcal{M}) = \emptyset$, then $V \in$ D-SEP$(X, Y, \mathcal{M}_{\underset{\sim}{X}})$ holds if and only if there is at least one collider path from $X$ to $V$ starting by an arrowhead at $X$ in $M$ such that each variable except for $X$ on the path is an ancestor of $X$ or $Y$ in $\mathcal{M}$.

The advantage of considering local MAG of $X$ rather than $\mathcal{P}$ with circles at $X$ is that based on local MAG we could obtain a set of variables, PD-SEP$(X, Y, M)$, as Def. 4. This set implies some variables in D-SEP$(X, Y, \mathcal{M}_{\underset{\sim}{X}})$ and has a good property that if PD-SEP$(X, Y, M) \backslash$D-SEP$(X, Y, \mathcal{M}_{\underset{\sim}{X}}) \neq \emptyset$, there must be at least one variable $V \in$ PD-SEP$(X, Y, M)$ which is not an ancestor of $X$ or $Y$ in $\mathcal{M}$, as shown by the combination of Thm. 2 and Def. 4. This property is utilized to prove the main result.

**Definition 4** (PD-SEP$(X, Y, M)$). Let $M$ be a local MAG of $X$ and $\mathcal{M}$ be an MAG consistent to $M$. Variable $V \in$ PD-SEP$(X, Y, M)$ if and only if $V \in$ PossAn$(Y, M) \backslash$De$(X, \mathcal{M})$[2] and there exists a collider path between $X$ and $V$ in $M$, where each non-endpoint variable is an ancestor of $X$ or $Y$ in $M$ but not a descendant of $X$ in $\mathcal{M}$.

*Remark.* In the literature, there is a well-defined notion Possible-D-SEP$(X, Y)$ [8, 19], which is introduced with the similar intention that indicates some variables possibly belonging to D-SEP$(X, Y, \mathcal{M})$. Since ours is pretty different in both required conditions and the restriction on ancestral relations, we use a new name PD-SEP$(X, Y, M)$ to distinguish them.

With the knowledge above, we present our main result in Thm. 3. Perhaps surprisingly, it implies that given a local MAG $M$, although there are possibly different D-SEP$(X, Y, \mathcal{M}_{\underset{\sim}{X}})$ in distinct MAG $\mathcal{M}$

---

[2]Note all the descendants of $X$ in $\mathcal{M}$ consistent to $M$ are knowable in $M$, which is detailed by Lemma 19 in Appendix C.1. Hence PD-SEP$(X, Y, M)$ can be obtained based on $M$ without the further knowledge about $\mathcal{M}$.

---

**Algorithm 1** Find all possible MCSs in $\mathcal{P}$

---

**input:** Intervention variable $X$, PAG $\mathcal{P}$
1: $L \leftarrow \emptyset$                  // It is to record each local MAG $M^j$ and corresponding $\text{MCS}^j$
2: **for** each local MAG $M^j$ of $X$ based on $\mathcal{P}$ by merely marking the marks at $X$ **do**
3:      **if** there is critical variable set $\mathbf{C}^j$ for $(X, Y)$ in $M^j$ **then**
4:          CRITICAL($M^j, \mathbf{C}^j, \mathbf{S}^j$)      // $\mathbf{S}^j$ ($\mathbf{S}^{j_k}$ below) is the set $\mathbf{S}$ defined in Def. 5 in $M^j$ ($M^{j_k}$)
5:      **else**
6:          $L = L \cup (M^j, \text{MCS}^j)$
7:      **end if**
8: **end for**
9: **function** CRITICAL($M^j, \mathbf{C}^j, \mathbf{S}^j$)
10:      **for** each element $\mathbf{C}^{j_k}$ in the power set of $\mathbf{C}^j$ **do**
11:          Obtain a new local MAG $M^{j_k}$ by orienting $F \rightarrow S$ for $\forall F \in \mathbf{C}^{j_k}, \forall S \in \mathbf{S}^j$ and marking the critical marks of $F \in \mathbf{C}^j \backslash \mathbf{C}^{j_k}$ as arrowheads
12:          **if** there is critical variable set $\mathbf{C}^j$ for $(X, Y)$ in $M^{j_k}$ **then**
13:              CRITICAL($M^{j_k}, \mathbf{C}^{j_k}, \mathbf{S}^{j_k}$)
14:          **else**
15:              $L = L \cup (M^{j_k}, \text{MCS}^{j_k})$
16:          **end if**
17:      **end for**
18: **end function**
**output:** $L$

---

consistent to $M$, the MCSs regarding them are the same if there are no *critical variables for $(X, Y)$* in $M$. We define critical variables for $(X, Y)$ in Def. 5, and show an example in Fig. 2, where $F_t$ is a critical variable and the circles colored by red are critical marks.

**Definition 5** (Critical variable for $(X, Y)$). In a local MAG $M$ with a path $X \leftrightarrow F_1 \leftrightarrow \cdots \leftrightarrow F_{t-1} \leftrightarrow F_t$ or $X \leftrightarrow F_1 \leftrightarrow \cdots \leftrightarrow F_{t-1} \leftarrow\circ F_t$, $t \geq 1$, where $F_1, \cdots, F_{t-1} \in \text{PD-SEP}(X, Y, M)$, $F_t$ is called a *critical variable* for $(X, Y)$ if there is a non-empty variable set $\mathbf{S}$ relative to $F_t$ defined as follows: $S \in \mathbf{S}$ if and only if in $M$ (1) $S$ is a child of $X, F_1, \cdots, F_{t-1}$, (2) there is $F_t \circ\!\!-\!\!* S$, (3) $S$ is at one minimal possible directed path from $F_t$ to $Y$, and no variable on the path belongs to PD-SEP$(X, Y, M)$. Each circle at $F_t$ on the edge with $F_{t-1}$ or $S \in \mathbf{S}$ is called a *critical mark* of $F_t$.

**Theorem 3.** Let $M$ be a local MAG of $X$ based on a PMAG $\mathcal{P}$. Then condition (1) below is sufficient for condition (2):

(1) there is no critical variable for $(X, Y)$ in $M$.

(2) for any an MAG $\mathcal{M}$ consistent to $M$ such that (a) D-SEP$(X, Y, \mathcal{M}_{\underline{X}}) \cap \text{De}(X, \mathcal{M}) = \emptyset$, (b) $X \in \text{An}(Y, \mathcal{M})$, it holds that $\mathbf{A}_{\mathcal{M}} = \mathbf{A}'_{\mathcal{M}}$, where $\mathbf{A}_{\mathcal{M}}$ denotes the MCS regarding D-SEP$(X, Y, \mathcal{M}_{\underline{X}})$ in $\mathcal{M}$ and $\mathbf{A}'_{\mathcal{M}}$ denotes the MCS regarding PD-SEP$(X, Y, M)$ in $\mathcal{M}$;

*Remark.* It is noteworthy that all the MAGs consistent to $M$ are Markov equivalent. Hence the MCSs regarding PD-SEP$(X, Y, M)$ in these MAGs are the same, namely, $\mathbf{A}'_{\mathcal{M}}$ is invariant for different $\mathcal{M}$. If there is no critical variable as condition (1), then for any MAG $\mathcal{M}$ consistent to $M$, $\mathbf{A}_{\mathcal{M}} = \mathbf{A}'_{\mathcal{M}}$ holds according to Thm. 3. We thus conclude $\mathbf{A}_{\mathcal{M}}$ is invariant for different $\mathcal{M}$ consistent to $M$, i.e., different D-SEP$(X, Y, \mathcal{M}_{\underline{X}})$ in distinct $\mathcal{M}$ consistent to $M$ share the same MCS regarding them.

Def. 5 and Thm. 3 form a graphical characterization that allows us to obtain all possible MCSs in all MAGs consistent to a local MAG $M$. We propose Alg. 1 to find all MCSs in $\mathcal{P}$ based on it. Given a PMAG $\mathcal{P}$, it is easy to find local MAGs $M$ based on $\mathcal{P}$ by differently marking the circles at $X$ without generating new unshielded colliders and directed or almost directed cycles (Line 2). For each $M$, if there is no critical variable in $M$, the MCS regarding D-SEP$(X, Y, \mathcal{M}_{\underline{X}})$ is unique, and it equals to

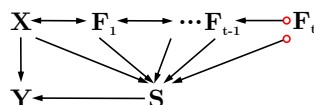

Figure 2: An illustration for critical variable $F_t$ and critical marks (red).

the MCS regarding PD-SEP$(X, Y, M)$ (Line 6). When there is a non-empty critical variable set $\mathbf{C}$, the MCSs in the MAGs consistent to $M$ could be different. The reason is that the variable in $\mathbf{C}$ could

be the ancestor of $X$ or $Y$ in some MAGs but the non-ancestor in other MAGs, due to the unidentified critical marks. Conditioning on the non-ancestor variables above makes some generalized back-door paths from $X$ to $Y$ active, thus these variables cannot appear in MCS. Hence, we further discuss the critical marks. For any subset $\mathbf{C}_k$ of the critical variable set $\mathbf{C}$, there could possibly be some MAG $\mathcal{M}$ consistent to $M$ where $\mathbf{C}_k \subseteq$ D-SEP$(X, Y, \mathcal{M}_{\underline{X}})$ and $(\mathbf{C} \backslash \mathbf{C}_k) \cap$ D-SEP$(X, Y, \mathcal{M}_{\underline{X}}) = \emptyset$. We use a new local MAG to represent the common parts of these MAGs (Line 10,11). And in the new one, we further consider whether critical variable exists (Line 12). The soundness of Alg. 1 is shown in Lemma 4, i.e. no matter what the true MAG $\mathcal{M}$ is, the MCS regarding D-SEP$(X, Y, \mathcal{M}_{\underline{X}})$ in $\mathcal{M}$ could be returned as long as D-SEP$(X, Y, \mathcal{M}_{\underline{X}}) \cap$ De$(X, \mathcal{M}) = \emptyset$.

**Lemma 4.** *Let $\mathcal{P}$ be a PMAG of MAG $\mathcal{M}$. If $X \in$ An$(Y, \mathcal{M})$ and D-SEP$(X, Y, \mathcal{M}_{\underline{X}}) \cap$ De$(X, \mathcal{M}) = \emptyset$, then the MCS regarding D-SEP$(X, Y, \mathcal{M}_{\underline{X}})$ in $\mathcal{M}$ and the corresponding local MAG are contained in the output of Algorithm 1.*

### 3.4 Learning marks and purity matrix by interventional data

In this section we propose how to learn marks and purity matrix by the interventional data of $Y$. After intervening on $X$, there are three possible situations implied by the interventional data of $Y$.

(1) $X$ has no causal effect on $Y$, i.e., $P(Y|do(X)) = P(Y)$,

(2) $X$ has causal effect on $Y$ and it can be identified by GBC, i.e., there exists a generalized back-door set $\mathbf{D}$ such that $P(Y|do(X)) = \int P(\mathbf{D})P(Y|X, \mathbf{D}) \, d\mathbf{D}$,

(3) $X$ has causal effect on $Y$ but GBC fails to identify it.

For situation (1), it is trivial to learn the marks by the interventional data according to Prop. 5.

**Proposition 5.** *If $P(Y|do(X)) = P(Y)$, the marks at $X$ are arrowheads in all the minimal possible directed paths from $X$ to $Y$ in a partially mixed ancestral graph.*

If $P(Y|do(X))$ is not equal to $P(Y)$, it implies that the interventional data accords with situation (2) or situation (3) but is uncertain to us. Hence, we need to find all possible MCSs based on $\mathcal{P}$, then judge whether one of the estimated causal effects with the observational data by (2) is consistent to the interventional data. If so, we can see that the causal effect of $X$ on $Y$ can be identified by GBC in the true MAG. Specifically, we first obtain a group of pairs $(M^j, \text{MCS}^j)$ by Alg. 1, where $M^j$ is a local MAG and MCS$^j$ is the corresponding MCS. Let $\hat{P}_{M^j}(Y|do(X = x))$ and $\hat{P}(Y|do(X = x))$ denote the estimated causal effect in each $M^j$ by MCS$^j$ and that under real intervention respectively. $Disc(P, Q)$ is the distribution discrepancy between $P$ and $Q$. If there are $m$ intervention samples $\big((do(X = x_1), Y_1), \cdots, (do(X = x_s), Y_s), \cdots, (do(X = x_m), Y_m)\big)$, we take MCS$^\star$ by

$$\text{MCS}^\star = \arg\min_{\text{MCS}^j} \sum_{s=1}^{m} Disc\Big( \hat{P}_{\text{MCS}^j}(Y|do(X = x_s)), \hat{P}(Y|do(X = x_s)) \Big). \tag{3}$$

Any distance metric can be used here. Since the attention is not on calculation, we take the expectation difference as the metric for convenience, i.e., $\text{MCS}^\star = \arg\min_{\text{MCS}^j} \sum_{s=1}^{m} |\hat{\mathbb{E}}_{\text{MCS}^j}(Y|do(X = x_s)) - Y_s|$. If the distance between the estimated causal effect by MCS$^\star$ and interventional data is larger than a given threshold, we think that GBC fails to identify such causal effect. We thus mark such $X$ as "FAIL" and orient some edges by Prop. 6. Otherwise, if there is only one local MAG $M^*$ corresponding to MCS$^\star$, we orient $\mathcal{P}$ by $M^*$. If there are more than one local MAG corresponding to MCS$^\star$, we orient $\mathcal{P}$ by the common marks of the local MAGs with MCS$^\star$. Besides, when GBC does not fail to identify the causal effect, all the edges out of $X$ in the minimal directed paths from $X$ to $Y$ in the PMAG are learned to be pure. Thus we could update the purity matrix accordingly.

**Proposition 6.** *In situation (3), let $\mathbf{T}$ denote all variables adjacent to $X$ in the minimal possible directed paths from $X$ to $Y$. For $T \in \mathbf{T}$, if for $\forall V \in \mathbf{T} \backslash T$, it holds either $T \notin$ Adj$(V, \mathcal{P})$ or there is a variable $S \notin$ Adj$(V, \mathcal{P})$ such that there is a collider path $X \circ\!\!\rightarrow T \leftrightarrow \cdots \leftarrow\!\!* S$ and every vertex except $S$ on the path is a parent of $V$, then $X \rightarrow T$.*

After the learning process above, we can further update the PMAG according to the property of MAG. Zhang [13] proposed ten complete rules with only observational data. However, how to orient PMAG

---
**Algorithm 2** ACIC (ACtive target effect Identification with latent Confounding)
---
**input:** PAG $\mathcal{P}$ by FCI algorithm;

1: Initialize $\mathcal{I}_1 = \{V | V \in \mathbf{V}(\mathcal{P}), V \in \mathrm{PossAn}(Y, \mathcal{P})$ and $P(Y|do(V))$ is unidentifiable by Prop. 1
    in $\mathcal{P}\}$             // Record the variables whose causal effects on $Y$ are unidentifiable by Prop. 1
2: Initialize $\mathcal{I}_2 = \emptyset$ // Record the variables whose causal effects are failed to be identified by GBC
3: **while** $\mathcal{I}_1 \backslash \mathcal{I}_2 \neq \emptyset$ **do**
4:     Select a variable $X$ with the maximum number of circles from $\mathcal{I}_1 \backslash \mathcal{I}_2$ to intervene
5:     **if** $P(Y|do(X))$ equals to $P(Y)$ **then**
6:         Update $\mathcal{P}$ by Prop. 5
7:     **else**
8:         Find all possible MCSs in $\mathcal{P}$ by Alg. 1 and select MCS$^*$ by Eq. 3
9:         **if** $\frac{1}{m}\sum_{s=1}^{m}|\hat{\mathbb{E}}_{\mathrm{MCS}^*}(Y|do(X=x_s)) - Y_s| \leq \tau$ **then**         // $\tau$ is a pre-set threshold
10:             **if** there is only local MAG $M^*$ with MCS$^*$ **then**
11:                 Update $\mathcal{P}$ to $M^*$, and label all the edges out of $X$ in the minimal directed paths
    from $X$ to $Y$ pure
12:             **else** there is more than one local MAG $M_1^*, M_2^*, \cdots, M_k^*$
13:                 Update $\mathcal{P}$ with the common marks in $M_1^*, M_2^*, \cdots, M_k^*$, and label all the edges
    out of $X$ in the minimal directed paths from $X$ to $Y$ pure
14:             **end if**
15:             Update $\mathcal{P}$ further based on the 11 rules for orienting PAGs with background knowl-
    edge, and update the purity matrix if some directed edges are newly identified to be visible
16:         **else**                              // In this case GBC fails to identify $P(Y|do(X))$
17:             Update $\mathcal{P}$ by Prop. 6
18:             $\mathcal{I}_2 = \mathcal{I}_2 \cup \{X\}$
19:         **end if**
20:     **end if**
21:     Update $\mathcal{I}_1$ by Prop. 1 with the updated PMAG $\mathcal{P}$ and the updated purity matrix
22: **end while**
**output:** The estimated causal effect of each variable on $Y$.
---

completely with the learned knowledge by interventional data is still an open problem. We add an additional rule referring to the known results for CPDAG [20]. But regretfully, whether the eleven rules are complete is unknown. In the updated graph, some directed edges are newly identified to be visible, thus they are identified to be pure. Hence we could also update the purity matrix accordingly.

> Rule 11: If $a \circ\!\!\rightarrow b \rightarrow c$, $a, b, c \in \mathrm{Adj}(d)$, $a \notin \mathrm{Adj}(c)$, and $a, d, c$ do not form an unshielded collider, then $d \rightarrow c$.

**Proposition 7.** *Rule 11 is sound.*

Combining all the parts above, we present the whole process in Alg. 2. In a PMAG obtained by FCI or learned after interventions, we judge which variables are possible ancestors of $Y$ by Prop 8. If there are variables with unidentified causal effects on $Y$ (Line 1) among them and we are not sure whether GBC fails to identify the effects (Line 3), we select one variable from them to intervene (Line 4) and learn structure by the interventional data (Line 5 - Line 20). Since intervention is expensive in reality, we hope achieving target effect identification with fewer intervention times. Hence, among the variables mentioned above, we greedily intervene on the one with the maximum number of circles. A running example is given in Appendix F to illustrate the detailed procedure of the proposed method.

**Proposition 8.** *In a PMAG $\mathcal{P}$, if there is no minimal possible directed path from $X$ to $Y$, then $X$ cannot be ancestor of $Y$ in any MAG consistent to $\mathcal{P}$. And it holds that $X \notin \mathrm{PossAn}(Y, \mathcal{P})$.*

## 4 Theoretical Results

In this section, we first prove the identifiability of causal effects[3]. Then we provide an analysis about the computation complexity of estimating all possible causal effects of each variable $X$ on $Y$.

---

[3]Note when we intervene, we have the information of $P(Y|do(X = x))$. However, we aim to identify $P(Y|do(X))$ when we say causal effect identification.

**Theorem 9.** Given the observational distribution of the observed variables, if there exists a valid generalized back-door set for $(X, Y)$ in the true MAG with the knowledge of the purity of each directed edge, then we can identify this set by only additional data of $Y$ under intervention on $X$.

Thm. 9 implies causal effect identifiability of each $X$ on $Y$ by interventions. In practice, when we intervene on $X$, in addition to identifying $P(Y|do(X))$, our method could learn some marks and the purity of some edges, which lead to the causal effect identification of *other variables* on $Y$. Hence we usually make target effect identification by far fewer intervention times than the variable number.

Estimating all possible causal effects takes the main computational cost. The process of estimating possible effects in a PMAG comprises finding all MCSs in Sec. 3.3 and using (2). We analyze the complexity of Sec. 3.3. Since the complexity is strongly related to the graph, it is hard to analyze for a general graph due to the randomness of the skeleton. We consider a special case - the complete graph, which is often considered as the most difficult graph to learn, because it has the most edges and we can learn no marks by only observational data as a result of no conditional independent relationship between the variables. To ensure that the interventional data accords with situation (2) or (3) when finding all MCSs in the PMAG is necessary, we set $Y$ as the descendant of all of $X_1, \cdots, X_p$. Due to space limit, we present a brief analysis here, while a detailed version is provided in Appendix E.2.

**Proposition 10.** *Let $\mathcal{M}$ be a complete MAG with $p + 1$ variables $X_1, \cdots, X_p, Y$, where the causal order of the variables except $Y$ is completely random and $Y$ is at the last. Denote the graph obtained by FCI with observational data by $\mathcal{P}$ and intervention variable by $X_i$. And let $M$ be a local MAG of $X_i$ with $p - 1 - k$ tails and $k$ arrowheads at $X_i$. The computational complexity of finding all possible causal effects $P(Y|do(X_i))$ in all the MAGs consistent to $M$ is $O(2^k)$. Further, the computational complexity of finding all causal effects $P(Y|do(X_i))$ in all the MAGs consistent to $\mathcal{P}$ is $O(3^p)$.*

Let $\mathbf{S}$ denote the set of variable that has an edge with an arrowhead at $X_i$. Since the graph is complete, for any subset $\mathbf{S}_1$ of $\mathbf{S}$, we could construct an MAG $\mathcal{M}$ based on $M$ such that MCS in $\mathcal{M}$ is $\mathbf{S}_1$, which is detailed in Appendix E.2. Hence there are $2^k$ causal effects of $X_i$ on $Y$ in the MAGs consistent to $M$. Our method thus achieves the minimum complexity in finding possible causal effects in $M$.

However, when estimating causal effects of $X_i$ on $Y$ in $\mathcal{P}$, the complexity of our method is $O(3^p)$, while there are $2^{p-1}$ causal effects in $\mathcal{P}$. The gap is from two aspects. One is the search for critical variables. The other is that the causal effects in MAGs from different local MAGs are possibly equal, while these causal effects are considered separately based on each local MAG. The complexity of the latter part

|  | local MAG $M$ | PAG $\mathcal{P}$ |
| --- | --- | --- |
| # Possible causal effects | $2^k$ | $2^{p-1}$ |
| Ours | $O(2^k)$ | $O(3^p)$ |
| Malinsky and Spirtes (2016) | None | $O(3^{p^2})$ |

Table 1: Complexity of estimating all effects.

dominates. While if we do not consider the graphical characterization based on critical variable and adopt the local method of Malinsky and Spirtes [16] in estimating causal effects that enumerates MAGs in a subregion, the complexity is exponentially larger than ours, as shown in Table 1.

## 5 Experiments

In this section, we apply our method on synthetic dataset to validate the effectiveness and efficiency of the proposed method. The code is developed based on R package "pcalg" [21].

We generate 100 random causal graphs and evaluate the number of correctly identified causal effects. In each graph, there are $p = 15$ variables and an edge occurs between two variables with probability 0.3. We randomly take 3 variables as latent confounders. And the last observed variable in the causal order is set to the response variable. We generate linear Gaussian data[4] according to the causal graph. Based on these information, we know not only the true MAG, but also whether each edge is pure. Hence we have the ground truth causal effect of each variable on the response variable by GBC. Beginning from the PAG obtained by the true MAG, we record the number of correctly identified causal effects under different intervention times with different methods.

---

[4]When variables are continuous, the positivity assumption tends to be violated. Here we ignore such risks because the identifiability has been proven and we just want to test whether the proposed method could accurately learn the structure and identify the generalized back-door set.

In our method, we adopt a greedy strategy to select the intervention variable. To verify the feasibility, we design a baseline method *ACIC-simple*, where we randomly select the intervention variable of which the causal effect on $Y$ has not been identified. To distinguish the methods with different strategies, we call our method *ACIC-greedy*. There are two additional baseline methods, *Do* and *ACI* [7].

For Do, it identifies causal relations by judging whether the distribution of observed variable takes a change under intervention, which idea is applied widely in many active causal discovery methods [2, 3]. Since in these methods all the variables could be observed under intervention, we allow Do to select which variable $X_j$ to observe under intervention instead of observing $Y$. The results are shown in Figure 3. The superiority of ACIC-greedy to ACIC-simple implies the greedy strategy to intervene helps saving intervention times. The phenomenon that ACIC-greedy is more efficient than Do verifies that exploiting causal effect elaborately could take us more message about the structure. There are many effects wrongly identified by ACI, which indicates the risk of ignoring latent confounders.

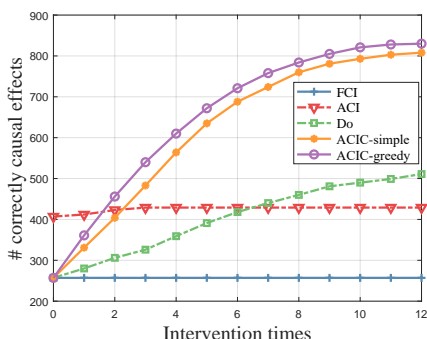

Figure 3: The number of correctly identified causal effects by different methods.

## 6   Conclusion

In this paper, we tackle the problem of identifying the causal effect of each variable on the response variable when the causal graph is unknown and there may exist latent confounders. We present the graphical characterization that allows us to find all possible causal effects in a PMAG. The characterization guides learning structure with the interventional data of only the response variable. Our method can achieve target effect identification effectively and efficiently in both saving intervention times and reducing computational complexity. And they are verified theoretically and empirically.

There are two main aspects to improve the method. One is on how to completely update a PMAG with the learned knowledge by interventional data. The other is on how to reduce the computational complexity further if possible. We look forward to future work that could address these problems.

## Acknowledgment

This research was supported by NSFC (61921006) and the Collaborative Innovation Center of Novel Software Technology and Industrialization. The authors would like to thank Tian Qin, Jie Qiao, Wei Chen, Peng Zhao, and Zhi-Hao Tan for helpful discussions. We are also grateful to the reviewers for their valuable comments.

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
