# Appendix A  Preliminary

In this part, we first show some well-defined notions in Section A.1. The relation between pure and visible is displayed in Section A.2. Then, we review the generalized back-door criterion and some related results such as the necessary and sufficient condition for the existence of a generalized back-door set in Section A.3. Finally, we generalize these results to adapt to partial mixed ancestral graph taking purity into account in Section A.4.

## A.1  Well-defined notions

In Pearl's causality framework, we use a graph $G = (\mathbf{V}(G), \mathbf{E}(G))$ to describe the causal relation between the variables, where $\mathbf{V}(G)$ denotes the variable set and $\mathbf{E}(G)$ denotes the causal relation. A *mixed graph* is a graph containing two kinds of edges: directed edges $\rightarrow$ and bi-directed edges $\leftrightarrow$. The two ends of an edge are called *mark* and have two types *arrowhead* or *tail*. If there are some *circles* ($\circ$) in a graph, the graph is a *partial mixed graph*. The circle implies that the mark could be either arrowhead or tail but they are uncertain to us. $V_i$ is a *parent/child/spouse* of $V_j$ if $V_i \rightarrow V_j / V_i \leftarrow V_j / V_i \leftrightarrow V_j$. A *possible directed path* from $V_i$ to $V_j$ is a path without arrowheads at the marks near $V_i$ on every edge. $V_i$ is an *(possible) ancestor* of $V_j$ if there is a *(possible) directed path* from $V_i$ to $V_j$ or $V_i = V_j$. The (possible) ancestor set is denoted by $\text{An}(V_j, G)$ ($\text{PossAn}(V_j, G)$). Similar definitions are for *(possible) descendant*, denoted by $\text{De}(V_j, G)$ ($\text{PossDe}(V_j, G)$). If $V_i \in \text{An}(V_j, G)$ and $V_i \leftrightarrow V_j$, it forms an *almost directed cycle*. There is a *collider path* between $V_i$ and $V_j$ if they are adjacent or there is a path between them where all passing nodes are colliders in the path. A path from $V_i$ to $V_j$ is a *minimal path* if there is not a path with order preserving subsequence from $V_i$ to $V_j$.

A mixed graph is an *ancestral graph* if there is no directed or almost directed cycle (since we assume no selection bias, we do not consider undirected edges in this paper). *m-separation* (*m-connecting* or *active*) is proposed by Richardson et al. [22] to describe the conditional independence (or not) in ancestral graph. An ancestral graph is a *maximal ancestral graph (MAG)* if for any two non-adjacent vertices, there is a set of vertices that m-separates them.

In an MAG, a path $p = \langle X, \cdots, W, V, Y \rangle$ is a *discriminating path for* $V$ if (1) $X$ and $Y$ are not adjacent, (2) every vertice between $X$ and $V$ in the path is a collider on $p$ and a parent of $Y$. Two MAGs are *Markov equivalent* if they share the same m-separations. A class comprised of all Markov equivalent MAGs is a *Markov equivalence class (MEC)*. We use a *partial ancestral graph (PAG)* to denote an MEC, where circle occurs if the marks here are not the same in all Markov equivalent MAGs, while tail (arrowhead) occurs if all the marks here are tails (arrowheads).

Given an MAG $\mathcal{M}$ or a PAG $\mathcal{P}$, A directed edge $A \rightarrow B$ is *visible* if there is a vertex $C$ not adjacent to $B$, such that either there is an edge between $C$ and $A$ that is into $A$, or there is a collider path between $C$ and $A$ that is into $A$ and every vertex on the path is a parent of $B$. Otherwise $A \rightarrow B$ is said to be *invisible*. As shown in Zhang [11], Maathuis et al. [12], the intention to introduce "visible" is to imply the situation that there cannot be a latent confounder influencing the two variables of the edge. Let $G$ denote an MAG $\mathcal{M}$ or a PAG $\mathcal{P}$. $G_{\underline{X}}$ is the graph obtaining from $G$ by removing all directed edges out of $X$ that are visible in $G$. *Positivity* requires that any one combination of the values of all the variables is with a positive probability. *Selection bias* says that when we collect the observational data, some latent variables are given, which are influenced by more than one observed variables.

For a partial mixed graph, we say it is a *partial mixed ancestral graph (PMAG)* if there is no directed or almost directed cycle and denote it by $\mathcal{P}$. The relation between PMAG and PAG is like that between partially directed acyclic graph (PDAG) and completed partial directed acyclic graph (CPDAG) when there are no latent confounders [10]. Both MAG and PAG are seen as special cases of PMAG. Next, we define the Markov Equivalent Class of PMAG $\mathcal{P}$. Before that, we present the definition of a triples *with order* and a necessary and sufficient condition about Markov equivalence by Ali et al. [14].

**Definition 6** (with order). Let $\mathfrak{D}_i (i \geq 0)$ be the set of triples of *with order $i$* in an MAG $\mathcal{M}$, defined recursively as follows:

| | |
|---|---|
| Order 0 | A triple $\langle a, b, c \rangle \in \mathfrak{D}_0$ if $a$ and $C$ are not adjacent |
| Order $i + 1$ | A triple $\langle a, b, c \rangle \in \mathfrak{D}_{i+1}$ if |
| | (1) for all $j < i + 1$, $\langle a, b, c \rangle \notin \mathfrak{D}_j$, and, |
| | (2) there is a discriminating path $\langle x, q_1, \cdots, q_p, b, y \rangle$ for $b$ with |
| | either $\langle a, b, c \rangle = \langle q_p, b, y \rangle$ or $\langle a, b, c \rangle = \langle y, b, q_p \rangle$ and the $p$ colliders |
| | $\langle x, q_1, q_2 \rangle, \cdots, \langle q_{p-1}, q_p, b \rangle \in \bigcup_{j \leq i} \mathfrak{D}_j$. |

The Markov Equivalent Class of PMAG $\mathcal{P}$ is comprised of all the MAGs $\mathcal{M}$ such that if (1) $\mathcal{M}$ and $\mathcal{P}$ have the same adjacencies, (2) all the colliders with order in $\mathcal{M}$ are colliders in $\mathcal{P}$, (3) all the non-circle marks in $\mathcal{P}$ are also in $\mathcal{M}$. In another word, $\mathcal{P}$ could be seen as a partial graph oriented by PAG of $\mathcal{M}$ with all non-circle marks consistent to $\mathcal{M}$. By classical causal discovery method, FCI for example, we could only obtain a PAG. In our

method, we need to keep identifying some circles at PAG until achieving target effect identification. We thus introduce $\mathcal{P}$ to denote the continuously updated graph "between" the initial PAG and the truth MAG $\mathcal{M}$.

## A.2  The relation between pure and visible

Here, we illustrate that visibility is sufficient but not necessary for purity. It is trivially concluded by Zhang [11]. By Lemma 10 of Zhang [11], we can see that for each invisible edge $V_i \to V_j$ in an MAG $\mathcal{M}$, there could be a DAG whose MAG is $\mathcal{M}$ as well as $V_i$ and $V_j$ are influenced by a common latent variable. It also evidently holds that there is a DAG whose MAG is $\mathcal{M}$ and the two variables of any an invisible edge are not influenced by common latent variables. Hence invisibility is not sufficient for the existence of latent confounders. That is, visibility is not necessary for purity. Also, By the converse negative proposition of Lemma 9 of Zhang [11], we know if a directed edge is visible, there cannot be latent confounding effects for this edge. It thus holds visibility is sufficient for purity. Hence we see that visibility is sufficient but not necessary for purity.

## A.3  Generalized back-door criterion and the graphical condition for the existence of generalized back-door set proposed by Maathuis et al. [12]

In this part, we first show the definition of back-door paths. Then we present some related concepts, followed by the necessary and sufficient graphical criteria for the existence of a set of variables that satisfies generalized back-door criterion. All of these results are presented by Zhang [11], Maathuis et al. [12].

**Definition** (Back-door path). Let $(X, Y)$ be an ordered pair of vertices in $G$, where $G$ is a DAG, CPDAG, MAG or PAG. We say that a path between $X$ and $Y$ is a back-door path from $X$ to $Y$ if it does not have a visible edge out of $X$.

**Definition** (Definite noncollider; Zhang [11]). A nonendpoint vertex $X_j$ on a path $\langle \cdots, X_i, X_j, X_k, \cdots \rangle$ in a partial mixed graph $G$ is a *definite non-collider* on the path if (1) there is a tail mark at $X_j$, that is $X_i \ast\!\!-\, X_j$ or $X_j \,-\!\!\ast X_k$, or (2) $\langle X_i, X_j, X_k \rangle$ is unshielded and has circle marks at $X_j$, that is, $X_i \ast\!\!-\!\circ X_j \circ\!-\!\ast X_k$ and $X_i$ and $X_k$ are not adjacent in $G$.

**Definition** (Definite status path; Zhang [11], Maathuis et al. [12]). A nonendpoint vertex $X$ on a path $p$ in a partial mixed graph is said to be of a definite status if it is either a collider or a definite noncollider on $p$. The path $p$ is said to be of a definite status if all nonendpoint vertices on the path are of a definite status.

**Definition** (Generalized back-door criterion; Maathuis et al. [12]). Let $\mathbf{X}$, $\mathbf{Y}$ and $\mathbf{Z}$ be pairwise disjoint sets of vertices in $G$, where $G$ represents a DAG, CPDAG, MAG or PAG. Then $\mathbf{Z}$ satisfies the *generalized back-door criterion* relative to $(\mathbf{X}, \mathbf{Y})$ and $G$ if (1) $\mathbf{Z}$ does not contain possible descendants of $\mathbf{X}$ in $G$; (2) for every $X \in \mathbf{X}$, the set $\mathbf{Z} \cup \mathbf{X} \backslash \{X\}$ blocks every definite status back-door path from $X$ to any member of $\mathbf{Y}$, if any, in $G$. A set $\mathbf{Z}$ that satisfies the generalized back-door criterion relative to $(\mathbf{X}, \mathbf{Y})$ and $G$ is called a *generalized back-door set* relative to $(\mathbf{X}, \mathbf{Y})$ and $G$.

Given a graph $G$, Maathuis et al. [12] proposed the necessary and sufficient graphical criteria for the existence of a set of variables that satisfies generalized back-door criterion with *only observational data*. Before that, the definition of D-SEP$(X, Y, G)$ is presented in Def. 1. Then we show one of the main result by Maathuis et al. [12] in Prop. 11.

**Definition 1** (D-SEP$(X, Y, G)$ [12]). Let $X$ and $Y$ be two distinct vertices in a mixed graph $G$. We say that $V \in$ D-SEP$(X, Y, G)$ if $V \neq X$, and there is a collider path between $X$ and $V$ in $G$, such that every vertex on this path (including $V$) is an ancestor of $X$ or $Y$ in $G$.

**Proposition 11** (Maathuis et al. [12]). *Let $X$ and $Y$ be two distinct vertices in $G$. There exists a generalized back-door set relative to $(X, Y)$ and $G$ if and only if $Y \notin$ Adj$(X, G_{\underline{X}})$ and D-SEP$(X, Y, G_{\underline{X}}) \cap$ PossDe$(X, G) = \emptyset$. Moreover, if such a generalized back-door set exists, then D-SEP$(X, Y, G_{\underline{X}})$ is such a set. Denote D-SEP$(X, Y, G_{\underline{X}})$ by $\mathbf{D}$, the causal effect of $X$ on $Y$ is*

$$P(Y|do(X = x)) = \int_{\mathbf{D}} P(\mathbf{D})P(Y|\mathbf{D}, X = x)\,\mathrm{d}\mathbf{D}. \tag{4}$$

## A.4  The improved result in Proposition 1

In this part, we first present our improved definition about back-door paths. With that we present the improved generalized back-door criterion. The justification for such modification is that our method continuously identifies some new marks with the actively interventional data until achieving target effect identification. The updated graph among the process is a PMAG, but not necessarily a PAG or MAG. In this case, whether a directed edge is with latent variables could be not only learned by the graphical criterion of visibility, but also learned by interventional data, which is reflected by the purity matrix. Hence we modify the related conditions to make it adapt to PMAG with purity matrix. This idea is simple and direct. But rigorously, the proof is very lengthy since it refers to too many results of Maathuis et al. [12], Zhang [23, 11]. Hence we suggest readers skipping the statements below and just reading the result in Prop. 1 without too much attention on the detail.

Considering the back-door paths in PAG or MAG from $X$ to $Y$ according to the definition of Maathuis et al. [12], they are the paths from $X$ to $Y$ that do not have a visible edges out of $X$. In fact, the reason that we need to consider the paths that have invisible edge out of $X$ is that there is one possibility that there are latent confounders behind the invisible edge out of $X$ (i.e. there is at least one latent confounder that influences the two variables of the invisible edge). In the circumstances, there is a back-door path from $X$ to $Y$ through the latent confounder. Hence we need to block such paths to make all the possible back-door paths blocked. As shown by Zhang [11] and Appendix A.2, by mere observational data, only the visible edges can be confirmed that have no latent confounders. However, if we could learn more knowledge about the existence of the latent confounders by interventional data, e.g., if we could identify that some possible back-door paths are not back-door paths in the DAGs, we do not consider blocking such paths. For example, for an MAG $X \to Y$, if the edge is invisible, $X \to Y$ is a back-door path. By original generalized back-door criterion, we cannot find a set that blocks the back-door path $X \to Y$, thus no set satisfies generalized back-door criterion. However, if we could learn that there is no latent confounder behind the edge $X \to Y$, then we could be sure that there is no need to block $X \to Y$, and $\emptyset$ satisfies back-door criterion in the causal graph, because no matter what the true causal graph that the MAG represents is, there are no back-door paths from $X$ to $Y$. Thus we can see $P(Y|do(X)) = P(Y|X)$. With such considerations, we take the purity matrix $W$ into account and define back-door path relative to $W$ as follows. Note we restrict that $W$ must be initialized by setting $W[i, j] = 1$ if the directed edge $i \to j$ is visible in the PAG learned by FCI algorithm. That is, $W$ is able to indicate that the visible edges in PAG are pure.

Next, we present the definition of $G_{\underline{X}}$. If $G$ is an MAG, then $G_{\underline{X}}$ denotes the graph by removing the directed edges out of $X$ in $G$ that are labeled to be pure in purity matrix. If $G$ is a PMAG, we obtain $G_{\underline{X}}$ by two steps. In the first step, let $R$ be any one MAG from the subclass of MAGs consistent to $G$ that have the same number of edges into $X$ in $G$. In the second step, we obtain $G_{\underline{X}}$ by removing the directed edges out of $X$ in $R$ if the edge is labeled to be pure in purity matrix or is visible in $R$.

**Definition 7** (Back-door path relative to $W$). Let $(X, Y)$ be an ordered pair of vertices in $G$, where $G$ is a PMAG. Denote the purity matrix in graph $G$ by $W$. We say that the path between $X$ and $Y$ is a back-door path from $X$ to $Y$ if it does not start with a directed edge $X \to Z$ with $W[X, Z] = 1$, where $Z$ is any a variable except for $X$.

With the back-door path relative to $W$, the generalized back-door criterion relative to $W$ is presented in the following. In contrast to the original definition, we also add the statement of $W$.

**Definition 8** (Generalized back-door criterion relative to $(G, W)$). Let $\mathbf{X}$, $\mathbf{Y}$ and $\mathbf{Z}$ be pairwise disjoint sets of vertices in $G$, where $G$ represents a PMAG. Denote the purity matrix in graph $G$ by $W$. Then $\mathbf{Z}$ satisfies the *generalized back-door criterion* relative to $(\mathbf{X}, \mathbf{Y})$ and $(G, W)$ if (1) $\mathbf{Z}$ does not contain possible descendants of $\mathbf{X}$ in $G$; (2) for every $X \in \mathbf{X}$, the set $\mathbf{Z} \cup \mathbf{X}\backslash\{X\}$ blocks every definite status back-door path relative to $W$ from $X$ to any member of $\mathbf{Y}$, if any, in $G$. A set $\mathbf{Z}$ that satisfies the generalized back-door criterion relative to $(\mathbf{X}, \mathbf{Y})$ and $(G, W)$ is called a *generalized back-door set* relative to $(\mathbf{X}, \mathbf{Y})$ and $(G, W)$.

Then, we present the adjustment criterion in PMAG and show that the set satisfies GBC in a PMAG satisfies adjustment criterion in the PMAG.

**Definition 9** (Adjustment criterion). Let $\mathbf{X}$, $\mathbf{Y}$, and $\mathbf{Z}$ be pairwise disjoint sets of vertices in a PMAG $G$ and $W$ be a purity matrix of $G$. Then we say that $\mathbf{Z}$ satisfies the adjustment criterion relative to $(X, Y)$ and $(G, W)$ if for any density $f$ compatible with $G$, we have

$$f(\mathbf{Y}|do(\mathbf{X})) = \begin{cases} f(\mathbf{Y}|\mathbf{X}), & \text{if } \mathbf{Z} = \varnothing, \\ \int_{\mathbf{Z}} f(\mathbf{Y}|\mathbf{Z}, \mathbf{X}) f(\mathbf{Z}) \, \mathrm{d}\mathbf{Z} = \mathbb{E}_{\mathbf{Z}}\{f(\mathbf{Y}|\mathbf{Z}, \mathbf{X})\}, & \text{otherwise.} \end{cases}$$

**Proposition 12.** *Let $\mathbf{X}$, $\mathbf{Y}$, $\mathbf{Z}$ be pairwise disjoint sets of variables in a PMAG $G$ and $W$ be a purity matrix of $G$. If $\mathbf{Z}$ satisfies the generalized back-door criterion relative to $(\mathbf{X}, \mathbf{Y})$ and $(G, W)$, then it satisfies the adjacent criterion relative to $(\mathbf{X}, \mathbf{Y})$ and $(G, W)$.*

*Proof.* It could be proved trivially by the results in Maathuis et al. [12]. Here we just show a proof sketch. If a set $\mathbf{Z}$ satisfies GBC relative to $(X, Y)$ and $(G, W)$, then it satisfies invariance criterion (see Definition 7.1 of Maathuis et al. [12]). The proof process is as one direction in Theorem 7.3 of Maathuis et al. [12]. Note here the result in converse direction does not necessarily hold, which is a difference between PMAG and PAG. Then by Theorem 7.1 of Maathuis et al. [12] we could directly prove the set satisfies the adjacent criterion relative to $(\mathbf{X}, \mathbf{Y})$. □

Based on these, we present Prop. 1.

**Proposition 1.** *Let $G$ be a PMAG and $W$ be a purity matrix of $G$. Suppose $X \in \text{An}(Y, G)$ and $Y$ are two distinct vertices in $G$. There exists a generalized back-door set relative to $(X, Y)$ and $(G, W)$ if and only if $\text{D-SEP}(X, Y, G_{\underline{X}}) \cap \text{PossDe}(X, G) = \emptyset$. Moreover, if the set exists, $\text{D-SEP}(X, Y, G_{\underline{X}})$ is such a set. Denote*

D-SEP$(X, Y, G_{\underline{X}})$ by $\mathbf{D}$, *the causal effect is*

$$P(Y|do(X = x)) = \int_{\mathbf{D}} P(\mathbf{D}) P(Y|\mathbf{D}, X = x) \, d\mathbf{D}. \tag{1}$$

*Proof.* There are two modifications between Prop. 11 and Prop. 1. One is that we additionally take the purity matrix $W$ into account in Prop. 1. Thus we need to replace the *visibility* in their proof by *purity*, and extend the *back-door path* and *generalized back-door criterion relative to $G$* to *back-door path relative to $W$* and *generalized back-door criterion relative to $(G, W)$*. The other is that we consider PMAG rather than only PAG. The rigorous proof is very lengthy and completely follows the proof of Maathuis et al. [12], Zhang [23], we thus just show the different part and leave the details of the proof for the readers. For the "if" statement, we need to prove $Y \notin \text{Adj}(X, G_{\underline{X}})$. If not, there is a directed edge $X \to Y$ where the two variables are influenced by a latent confounder. In such case we could see that $Y \in$ D-SEP$(X, Y, G_{\underline{X}}) \cap$ PossDe$(X, G)$, which contradicts D-SEP$(X, Y, G_{\underline{X}}) \cap$ PossDe$(X, G) = \emptyset$. Hence we have $Y \notin \text{Adj}(X, G_{\underline{X}})$ and D-SEP$(X, Y, G_{\underline{X}}) \cap$ PossDe$(X, G) = \emptyset$. The other proof process of Prop. 1 is totally same as that of Maathuis et al. [12]. Note that $\mathcal{R}$ in Theorem 4.1 of Maathuis et al. [12] denotes any an MAG in the subclass of MAGs in the MEC described by $G$ that have the same number of edges into $X$ as $G$. Hence $\mathcal{R}_{\underline{X}}$ denotes the MAG obtained from $\mathcal{R}$ by removing the pure edges out of $X$ in $G$, which is exactly the $G_{\underline{X}}$ in our paper.

For the "only if" statement, Lemma 7.7 of Maathuis et al. [12] does not necessarily hold in PMAG. We thus cannot follow the proof directly. However, since we restrict that $X \in \text{An}(Y, G)$, and $V \in$ D-SEP$(X, Y, G_{\underline{X}})$, we can also conclude that there must be a directed path from $V$ to $Y$ in $G_{\underline{X}}$. The other parts are the same as those of Theorem 4.1 of Maathuis et al. [12]. $\qquad \square$

## Appendix B  Minimal Conditional Set

In this part, we first define *minimal conditional set regarding* D-SEP$(X, Y, \mathcal{M}_{\underline{X}})$. Then we show the uniqueness of MCS by Lemma 14 in Appendix B.1. Next, we propose Alg. 0.1 to find the MCS in Appendix B.2, along with the theoretical guarantee for the soundness of Alg. 0.1.

**Definition 2** (Minimal conditional set regarding D-SEP$(X, Y, \mathcal{M}_{\underline{X}})$ (MCS)). Let $\mathcal{M}$ be an MAG. $X$, $Y$, and $\mathbf{D}$ denote the intervened variable, response variable, and the set D-SEP$(X, Y, \mathcal{M}_{\underline{X}})$ in $\mathcal{M}$, respectively. $\overline{\mathbf{D}}$ is a subset of $\mathbf{D}$. $\overline{\mathbf{D}}$ is a minimal conditional set regarding $\mathbf{D}$ in $\mathcal{M}$ if

(1) $(Y \perp \mathbf{D}\backslash\overline{\mathbf{D}} \,|\, \overline{\mathbf{D}} \cup \{X\})_{\mathcal{M}}$,

(2) $(Y \not\perp \mathbf{D}\backslash\overline{\mathbf{D}}' |\overline{\mathbf{D}}' \cup \{X\})_{\mathcal{M}}$, for any $\overline{\mathbf{D}}' \subset \overline{\mathbf{D}}$.

### B.1  The uniqueness of MCS

Here, we prove the uniqueness of MCS. Before that, we propose Lemma 13 for supporting the main proof of the uniqueness of MCS. Note that positivity is assumed in this paper. With an abuse of notation, we do not distinguish random variable from the value of the random variable to make the proof concise.

**Lemma 13.** *Let* $\mathbf{A}, \mathbf{B}, \mathbf{C}, \mathbf{D}$ *be four pairwise disjoint variable sets. If* $\Pr(\mathbf{D}|\mathbf{C}, \mathbf{A}) = \Pr(\mathbf{D}|\mathbf{C}, \mathbf{B})$, *then* $\Pr(\mathbf{D}|\mathbf{C}, \mathbf{A}) = \Pr(\mathbf{D}|\mathbf{C}, \mathbf{B}) = \Pr(\mathbf{D}|\mathbf{C})$.

In fact, the condition of this lemma provides an intuition that conditional distribution is irrelevant to $\mathbf{A}$ and $\mathbf{B}$, which concludes the result directly. Nevertheless, we give a rigorous proof as follows.

*Proof.* Multiply both sides of the condition by $\Pr(\mathbf{A}|\mathbf{C})$, it holds that

$$\Pr(\mathbf{A}|\mathbf{C}) \Pr(\mathbf{D}|\mathbf{C}, \mathbf{A}) = \Pr(\mathbf{A}|\mathbf{C}) \Pr(\mathbf{D}|\mathbf{C}, \mathbf{B})$$
$$\sum_{\mathbf{A}} \Pr(\mathbf{A}|\mathbf{C}) \Pr(\mathbf{D}|\mathbf{C}, \mathbf{A}) = \sum_{\mathbf{A}} \Pr(\mathbf{A}|\mathbf{C}) \Pr(\mathbf{D}|\mathbf{C}, \mathbf{B})$$
$$\Pr(\mathbf{D}|\mathbf{C}) = \Pr(\mathbf{D}|\mathbf{C}, \mathbf{B}).$$

Hence, we have $\Pr(\mathbf{D}|\mathbf{C}, \mathbf{B}) = \Pr(\mathbf{D}|\mathbf{C})$. Similarly, it also holds that $\Pr(\mathbf{D}|\mathbf{C}, \mathbf{A}) = \Pr(\mathbf{D}|\mathbf{C})$. We thus get the desired conclusion. $\qquad \square$

**Lemma 14** (Uniqueness). *If both* $\mathbf{F}_1$ *and* $\mathbf{F}_2$ *are minimal conditional sets regarding* $\mathbf{D}$ *in* $\mathcal{M}$, *then* $\mathbf{F}_1 = \mathbf{F}_2$.

*Proof.* If $\mathbf{F}_1 \neq \mathbf{F}_2$, according to the definition of MCS, it evidently concludes $\mathbf{F}_1 \backslash \mathbf{F}_2 \neq \emptyset$ and $\mathbf{F}_2 \backslash \mathbf{F}_1 \neq \emptyset$. We denote $\mathbf{F} = \mathbf{F}_1 \cap \mathbf{F}_2$, $\mathbf{A} = \mathbf{F}_1 \backslash \mathbf{F}$, $\mathbf{B} = \mathbf{F}_2 \backslash \mathbf{F}$. Evidently $\mathbf{A}, \mathbf{B}, \mathbf{F}, X$ are disjoint.

According to the definition, we have

$$Y \perp \mathbf{D} \backslash (\mathbf{F} \cup \mathbf{A}) | \mathbf{F} \cup \mathbf{A} \cup X, \tag{5}$$

$$Y \perp \mathbf{D} \backslash (\mathbf{F} \cup \mathbf{B}) | \mathbf{F} \cup \mathbf{B} \cup X. \tag{6}$$

By (5), it holds

$$\Pr(Y | \mathbf{F}, \mathbf{A}, X) \Pr(\mathbf{D} \backslash (\mathbf{F} \cup \mathbf{A}) | \mathbf{F}, \mathbf{A}, X)$$
$$= \Pr(Y, \mathbf{D} \backslash (\mathbf{F} \cup \mathbf{A}) | \mathbf{F}, \mathbf{A}, X).$$

Multiply $\Pr(\mathbf{A} | \mathbf{F}, X)$ on both sides,

$$\Pr(Y | \mathbf{F}, \mathbf{A}, X) \Pr(\mathbf{D} \backslash \mathbf{F} | \mathbf{F}, X) = \Pr(Y, \mathbf{D} \backslash \mathbf{F} | \mathbf{F}, X). \tag{7}$$

$$\Pr(Y | \mathbf{F}, \mathbf{A}, X) = \Pr(Y | \mathbf{D} \backslash \mathbf{F}, \mathbf{F}, X) = \Pr(Y | \mathbf{D}, X). \tag{8}$$

The conversion from (7) to (8) depends on $\Pr(\mathbf{D} \backslash \mathbf{F} | \mathbf{F}, X) \neq 0$ for any attainable value of the variables, which is due to the positivity assumption. Similarly, we have

$$\Pr(Y | \mathbf{F}, \mathbf{B}, X) = \Pr(Y | \mathbf{D}, X). \tag{9}$$

Combine (8) and (9), $\Pr(Y | \mathbf{F}, X, \mathbf{A}) = \Pr(Y | \mathbf{F}, X, \mathbf{B}) = \Pr(Y | \mathbf{D}, X)$. By Lemma 13, we know $\Pr(Y | \mathbf{F}, X, \mathbf{A}) = \Pr(Y | \mathbf{F}, X, \mathbf{B}) = \Pr(Y | \mathbf{F}, X)$. We rewrite (7) as

$$\Pr(Y | \mathbf{F}, X) \Pr(\mathbf{D} \backslash \mathbf{F} | \mathbf{F}, X) = \Pr(Y, \mathbf{D} \backslash \mathbf{F} | \mathbf{F}, X).$$

It concludes $Y \perp \mathbf{D} \backslash \mathbf{F} | \mathbf{F}, X$. It conflicts with the conditions that $\mathbf{F} \subsetneq \mathbf{F}_1$ and $\mathbf{F}_1$ is a minimal conditional set regarding $\mathbf{D}$. □

## B.2 The algorithm to find MCS and the theoretical guarantee

Till now, we have proved the uniqueness of MCS, which plays an important role in the following proofs. Next, we propose Alg. 0.1 to find the MCS regarding $\mathbf{D}$ in $\mathcal{M}$. Lemma 15 and Lemma 16 are two important lemmas used widely in the proofs below. With these results, we prove the soundness of Alg. 0.1 to find the MCS in Lemma 17.

---

**Algorithm 0.1** Find the MCS regarding $\mathbf{D}$ in $\mathcal{M}$

1: $\mathbf{S} = \mathbf{D}$          // Record the MCS
2: **for** $V$ **in S do**
3:      **if** $V \perp Y | \{X, \mathbf{S} \backslash \{V\}\}$ in $\mathcal{M}$ **then**
4:          $\mathbf{S} = \mathbf{S} \backslash \{V\}$
5:      **end if**
6: **end for**
**output:** $\mathbf{S}$.

---

**Lemma 15.** *Let $\mathbf{A}, \mathbf{B}, \mathbf{C}$ be three pairwise disjoint variables sets. If $\mathbf{E} \subset \mathbf{B}$, $\mathbf{E} \neq \emptyset$ and $\mathbf{A} \perp \mathbf{B} | \mathbf{C}$, then $\mathbf{A} \perp \mathbf{B} \backslash \mathbf{E} | \{\mathbf{C} \cup \mathbf{E}\}$.*

*Proof.* Since $\mathbf{A} \perp \mathbf{B} | \mathbf{C}$ and $\mathbf{E} \subset \mathbf{B}$, it concludes that $\mathbf{A} \perp \mathbf{E} | \mathbf{C}$. We have

$$\Pr(\mathbf{A} | \mathbf{C}, \mathbf{E}) \Pr(\mathbf{B} \backslash \mathbf{E} | \mathbf{C}, \mathbf{E})$$
$$= \Pr(\mathbf{A} | \mathbf{C}) \Pr(\mathbf{B} \backslash \mathbf{E} | \mathbf{C}, \mathbf{E})$$
$$= \Pr(\mathbf{A} | \mathbf{C}, \mathbf{B}) \Pr(\mathbf{B} \backslash \mathbf{E} | \mathbf{C}, \mathbf{E})$$
$$= \Pr(\mathbf{A} | \mathbf{C}, \mathbf{B} \backslash \mathbf{E}, \mathbf{E}) \Pr(\mathbf{B} \backslash \mathbf{E} | \mathbf{C}, \mathbf{E})$$
$$= \Pr(\mathbf{A}, \mathbf{B} \backslash \mathbf{E} | \mathbf{C}, \mathbf{E}).$$

We get the desired conclusion. □

**Lemma 16.** *For two pairwise disjoint sets $\mathbf{A}_1, \mathbf{A}_2$ such that $(\mathbf{A}_1 \cup \mathbf{A}_2) \cap \{X, Y\} = \emptyset$, denote $\mathbf{A} = \mathbf{A}_1 \cup \mathbf{A}_2$. Let $\mathbf{E}$ be a set such that $\mathbf{E} \supseteq \mathbf{A}$ and $\mathbf{E} \cap \{X, Y\} = \emptyset$. If $\mathbf{A}_2 \perp Y | X, \mathbf{A}_1$ and $\mathbf{E} \backslash \mathbf{A} \perp Y | X, \mathbf{A}$, then $(\mathbf{E} \backslash \mathbf{A}, \mathbf{A}_2) \perp Y | X, \mathbf{A}_1$.*

*Proof.* Denote $\mathbf{E}\backslash\mathbf{A}$ by $\mathbf{B}$.

$$\Pr(\mathbf{B}, Y|X, \mathbf{A}_1)$$
$$= \sum_{\mathbf{A}_2} \Pr(\mathbf{B}, \mathbf{A}_2, Y|X, \mathbf{A}_1),$$
$$= \sum_{\mathbf{A}_2} \Pr(\mathbf{A}_2, Y|X, \mathbf{A}_1) \Pr(\mathbf{B}|X, \mathbf{A}_1, \mathbf{A}_2, Y),$$
$$= \sum_{\mathbf{A}_2} \Pr(\mathbf{A}_2|X, \mathbf{A}_1) \Pr(Y|X, \mathbf{A}_1) \Pr(\mathbf{B}|X, \mathbf{A}_1, \mathbf{A}_2)$$
$$\because (\mathbf{A}_2 \perp Y|X, \mathbf{A}_1) \text{ and } (\mathbf{E}\backslash\mathbf{A} \perp Y|X, \mathbf{A}),$$
$$= \Pr(Y|X, \mathbf{A}_1) \Pr(\mathbf{B}|X, \mathbf{A}_1).$$

It completes the proof. $\qquad\square$

**Lemma 17.** *Algorithm 0.1 is sound in finding the MCS regarding $\mathbf{D}$ in $G$.*

*Proof.* Denote the MCS by $\overline{\mathbf{D}}$. Without loss of generality, suppose the whole enumerating sequence in Line 2 is $T_1, T_2, \cdots, T_k$. For any $T_i$, we will prove if $T_i \in \mathbf{D}\backslash\overline{\mathbf{D}}$, the condition in Line 3 is satisfied. While if $T_i \in \overline{\mathbf{D}}$, the condition in Line 3 is not satisfied. In this case we could delete the variables which do not belong to $\overline{\mathbf{D}}$ and keep the variables belonging to $\overline{\mathbf{D}}$ in $\mathbf{D}$. We prove it by mathematical induction. $T_1$ is considered at first. If $T_1 \in \mathbf{D}\backslash\overline{\mathbf{D}}$, by the definition of $\overline{\mathbf{D}}$, it holds that $\mathbf{D}\backslash\overline{\mathbf{D}} \perp Y|\overline{\mathbf{D}}, X$. Combining $\mathbf{D}\backslash\overline{\mathbf{D}} \perp Y|\overline{\mathbf{D}}, X$ with lemma 15, we have $T_1 \perp Y|\mathbf{D}\backslash\{T_1\}, X$, which satisfies the condition in Line 3. While if $T_1 \in \overline{\mathbf{D}}$, we aim to prove $T_1 \not\perp Y|\mathbf{D}\backslash\{T_1\}, X$. For the sake of contradiction, we assume $T_1 \perp Y|\mathbf{D}\backslash\{T_1\}, X$. By the definition of MCS, we know that one set $\mathbf{D}' \subseteq \mathbf{D}\backslash\{T_1\}$ could be as the MCS. However, we know $\overline{\mathbf{D}}$ an MCS which contains $T_1$. That means there are two distinct MCSs, which contradicts Lemma 14. Hence $T_1 \not\perp Y|\mathbf{D}\backslash\{T_1\}, X$.

We suppose the result above holds for the sequence $T_1, T_2, \cdots, T_{i-1}$ and now the algorithm judges $T_i$ in Line 3. Since some variables have been deleted by Line 4, we denote the remained variable set $\mathbf{T}$. The only difference between $\mathbf{D}$ and $\mathbf{T}$ is the variables from $T_1, T_2, \cdots, T_{i-1}$ which satisfies Line 3, i.e. the variable does not belong to $\overline{\mathbf{D}}$ according to the inductive hypothesis. Hence it is evident that $\overline{\mathbf{D}} \subseteq \mathbf{T} \subseteq \mathbf{D}$. If $T_i \in \mathbf{D}\backslash\overline{\mathbf{D}}$, using $Y \perp \mathbf{D}\backslash\overline{\mathbf{D}}|\overline{\mathbf{D}}, X$ and Lemma 15 (setting $\mathbf{B} = \mathbf{D}\backslash\overline{\mathbf{D}}$ and $\mathbf{D} = (\mathbf{T}\backslash\{T_i\})\backslash\overline{\mathbf{D}}$ in Lemma 15), we have $Y \perp \mathbf{D}\backslash(\mathbf{T}\backslash\{T_i\})|\mathbf{T}\backslash\{T_i\}, X$. Hence $Y \perp T_i|\mathbf{T}\backslash\{T_i\}, X$, which implies that the condition in Line 3 is satisfied. If $T_i \in \overline{\mathbf{D}}$, the proof is also similar to that for $T_1$. For the sake of contradiction, we assume $T_i \perp Y|\mathbf{T}\backslash\{T_i\}, X$. Without loss of generality, we denote the last deleted variable from $T_1 \cdots, T_{i-1}$ by $T_{l_1}$. Since $T_{l_1}$ is deleted in Line 4, it implies that $T_{l_1} \perp Y|\mathbf{T}, X$. Combining $T_i \perp Y|\mathbf{T}\backslash\{T_i\}, X$ and $T_{l_1} \perp Y|\mathbf{T}, X$ by Lemma 16, we obtain that $(T_{l_1}, T_i) \perp Y|\mathbf{T}\backslash\{T_i\}, X$. We continue finding the second-to-last deleted variable $T_{l_2}$ from $T_1 \cdots, T_{i-1}$ and we can obtain $(T_{l_1}, T_{l_2}, T_i) \perp Y|\mathbf{T}\backslash\{T_i\}, X$. Repeat this process until we find all deleted variables from $T_1 \cdots, T_{i-1}$, we obtain that $(T_{l_1}, T_{l_2}, \cdots, T_{l_s}, T_i) \perp Y|\mathbf{T}\backslash\{T_i\}, X$. Note $T_{l_1}, T_{l_2}, \cdots, T_{l_s} = \mathbf{D}\backslash\mathbf{T}$, it thus holds $(\mathbf{D}\backslash\mathbf{T}, T_i) \perp Y|\mathbf{T}\backslash\{T_i\}, X$. By the definition of MCS, we can obtain a subset $\mathbf{D}' \subseteq \mathbf{T}\backslash\{T_i\}$ as the MCS regarding $\mathbf{D}$. But according to the condition, $T_i$ should be one variable from the MCS, which contradicts Lemma 14 that implies the MCS is unique. Hence $T_i \not\perp Y|\mathbf{T}\backslash\{T_i\}, X$. We prove the result for $T_i$. The mathematical induction completes.

By mathematical induction, we prove the set of the remained variables by Algorithm 0.1 is MCS. $\qquad\square$

## Appendix C   Proofs for the Results in Section 3.3

There are three parts in this section. The first part is about the proof for Theorem 2. We first present Lemma 18, which plays an important role in the proof of Theorem 2. By the lemma we prove Theorem 2 trivially. The second part is about the proof for Theorem 3. We define critical variable in Definition 5. Then we provide the proof for Theorem 3. In the last part, we prove Lemma 4.

### C.1   Proofs for Theorem 2

**Definition 3** (Local MAG of $X$ based on $\mathcal{P}$). Given a PMAG $\mathcal{P}$ and a variable $X$, a PMAG $M$ is a *local MAG of $X$ based on $\mathcal{P}$* if (1) $M$ is with definite marks (arrowheads or tails) at $X$; (2) $M$ is obtained from $\mathcal{P}$ by marking some circles without generating new unshielded colliders or directed or almost directed cycles. We call it *local MAG* for short if there is no ambiguity and denote it by $M$, which is different from calligraphic $\mathcal{M}$ that denotes MAG.

**Lemma 18.** *Let $\mathcal{P}$ be a PMAG which MAG $\mathcal{M}$ is consistent to. If $V \in$ D-SEP$(X, Y, \mathcal{M})$, then there is at least one collider path between $X$ and $V$ in $\mathcal{P}$ and each variable in this path is a possible ancestor of $X$ or $Y$ in $\mathcal{P}$.*

*Proof.* Suppose $V \in$ D-SEP$(X, Y, \mathcal{M})$. According to the definition of D-SEP$(X, Y, \mathcal{M})$, there is a collider path between $V$ and $X$, and each variable in this path is an ancestor of $X$ or $Y$ in $\mathcal{M}$. Since $\mathcal{M}$ and $\mathcal{P}$ have the same skeleton, we analyze the corresponding path in PMAG $\mathcal{P}$. It is evident that each variable in this path is a possible ancestor of $X$ or $Y$ in $\mathcal{P}$. We then just prove that there is a collider path where each variable is the possible ancestor of $X$ or $Y$ in $\mathcal{P}$.

Since there is a collider path between $X$ and $V$, where each variable is an ancestor of $X$ or $Y$ in $\mathcal{M}$, there must be a minimal collider path between $X$ and $V$, where each variable is an ancestor of $X$ or $Y$. For the sake of clarity, we replace $V$ by $F_n$ and denote the minimal collider path by $X \ast\!\!\rightarrow F_1 \leftrightarrow \cdots \leftrightarrow F_{n-1} \leftarrow\!\ast F_n$. We assume the collider $F_{i-1} \leftrightarrow F_i \leftrightarrow F_{i+1}$ is a collider without order. There must be an edge between $F_{i-1}$ and $F_{i+1}$. The edge is evidently not bi-directed, otherwise it contradicts the minimal collider path. Without loss of generality, we suppose the edge is $F_{i-1} \rightarrow F_{i+1}$. The reasonableness of this assumption is that collider path between $X$ and $F_n$ is symmetric (here the condition that each variable is an ancestor of $X$ or $Y$ is not considered).

Next we consider the collider $F_{i-2} \leftrightarrow F_{i-1} \leftrightarrow F_i$. There are two possible situations. One is that the collider is with some constant order. In this case, we prove there is an edge $F_{i-1} \rightarrow F_{i+1}$ in the following. If they are not adjacent, then there is a discriminating path $F_{i-2}, F_{i-1}, F_i, F_{i+1}$ for $F_i$, which implies that the collider $F_{i-1} \leftrightarrow F_i \leftrightarrow F_{i+1}$ is with order, contradicting the condition. Due to the ancestral property, the edge could only be as $F_{i-2} \ast\!\!\rightarrow F_{i+1}$. And it thus be $F_{i-2} \rightarrow F_{i+1}$ due to the minimal collider path condition. The other situation is that $F_{i-2} \leftrightarrow F_{i-1} \leftrightarrow F_i$ is also without order.

In the following we consider the collider at $F_{i-2}, F_{i-3}, \cdots, F_1$ recursively. During the iteration, suppose the first variable $F_k$ where the collider is without order. That is, the colliders at the variables $F_{k+1}, F_{k+2}, \cdots, F_{i-2}, F_{i-1}$ are with order. In this case, we first prove that for all the variable $F_m, k \leq m \leq i-1$, there is an edge $F_m \rightarrow F_{i+1}$. We could prove it by mathematical induction. We have proven that $F_{i-1} \rightarrow F_{i+1}$ before. We suppose that for all the variables among $F_{i-t}, F_{i-t+1}, \cdots, F_{i-1}, i-t \geq k+1$, there is a directed edge from the variable to $F_{i+1}$. For the variable $F_{i-t-1}$, if it is not adjacent to $F_{i+1}$, then there is a discriminating path $F_{i-t-1}, F_{i-t}, F_{i-t+1}, \cdots, F_{i-2}, F_{i-1}, F_i, F_{i+1}$ for $F_i$, which implies that $F_{i-1}, F_i, F_{i+1}$ is a collider with order, contradicting the condition. Hence there must be an edge between $F_{i-t-1}$ and $F_{i+1}$. Since $F_{i-t-1} \leftrightarrow F_{i-t} \rightarrow F_{i-t+1}$, the edge is as $F_{i-t-1} \ast\!\!\rightarrow F_{i+1}$. In addition, the edge cannot be bi-directed, otherwise contradicting the minimal collider path condition. Hence it can only be as $F_{i-t-1} \rightarrow F_{i+1}$. The mathematical induction completes.

Hence, for all the variables among $F_k, F_{k+1}, \cdots, F_{i-1}$, there is a directed edge from the variable to $F_{i+1}$. Since $F_{k-1} \leftrightarrow F_k \leftrightarrow F_{k+1}$ is a collider without order, there must be an edge between $F_{k-1}$ and $F_{k+1}$. Evidently the edge cannot be bi-directed due to the minimal collider path condition. We consider the situation that the edge is $F_{k-1} \leftarrow F_{k+1}$. In this case, we could prove that there is an edge $F_{k-1} \leftarrow F_{k+2}$. If they are not adjacent, then there is a discriminating path $F_{k-1}, F_k, F_{k+1}, F_{k+2}$ for $F_{k+1}$ and the collider $F_k \leftrightarrow F_{k+1} \leftrightarrow F_{k+2}$ is with a constant order, thus $F_{k-1} \leftrightarrow F_k \leftrightarrow F_{k+1}$ is with a constant order, which contradicts the condition that the collider at $F_k$ is without order. Due to the ancestral property, the edge is as $F_{k-1} \leftarrow\!\ast F_{k+1}$. Due to the minimal collider path condition, the edge could only be $F_{k-1} \leftarrow F_{k+1}$. Similar to the process above, we could prove that for all variables between $F_{k+1}$ and $F_i$, there is a directed edge from the variable to $F_{k-1}$. In this case, there is a collider path $F_{k-1} \leftrightarrow F_k \leftrightarrow \cdots F_{i+1}$ with edges $F_j \rightarrow F_{i+1}, \forall k \leq j \leq i-1$, and $F_i \rightarrow F_{k-1}$. It is an inducing path thus there is a bi-directed edge $F_{k-1} \leftrightarrow F_{i+1}$, otherwise the maximal property is violated. However, the bi-directed edge $F_{k-1} \leftrightarrow F_{i+1}$ makes the path not minimal, which contradicts the minimal collider path condition.

Thus, the only possible condition is that there is an edge between $F_{k-1}$ and $F_{k+1}$ and the edge is as $F_{k-1} \rightarrow F_{k+1}$, and the collider $F_{k-1} \leftrightarrow F_k \leftrightarrow F_{k+1}$ is without order. Now we consider the collider without order $F_{k-1} \leftrightarrow F_k \leftrightarrow F_{k+1}$ instead of $F_{i-1} \leftrightarrow F_i \leftrightarrow F_{i+1}$. In another word, by such exchange we consider a collider without order that is nearer to $X$ in the collider path. By such exchange we could find the collider at $F_m$ without order which is nearest to $X$ and there is an edge $F_{m-1} \rightarrow F_{m+1}$. Note during the iteration from $F_{i-2}$ to $F_1$ before, if there is not a collider without order, then $m = i$.

If $F_m$ is $F_1$, i.e., $X \ast\!\!\rightarrow F_1 \leftrightarrow F_2$ is such a collider without order and there is an edge $X \rightarrow F_2$, then it contradicts the minimal collider path condition since there is a collider path $X \rightarrow F_2 \leftrightarrow F_3 \leftrightarrow \cdots \leftrightarrow F_{n-1} \leftarrow\!\ast F_n$. If $F_m$ is not $F_1$, all the colliders between $X$ to $F_m$ are with orders. Similar to the proof above, there is edge $F_j \rightarrow F_{m+1}, \forall 0 \leq j \leq m$, where $F_0$ is $X$. In this condition there is a collider path $X \rightarrow F_{m+1} \leftrightarrow F_{m+2} \leftrightarrow \cdots \leftrightarrow V$, which also contradicts the minimal collider path condition. Hence we conclude that there cannot be a collider without order in the minimal collider path between $X$ and $V$. By Thm. 3.7 of Ali et al. [14], all MAGs in a Markov equivalent class have the same colliders with order. Since FCI is complete [13], all the colliders with order are identified in the graph learned by FCI, thus are identified in $M$. Hence, there is a collider path between $X$ and $V$ in $M$. $\square$

**Lemma 19.** *Let $M$ be a local MAG of $X$. Suppose an MAG $\mathcal{M}$ is consistent to $M$. For any variable $A$ in $\mathcal{M}$, $A$ is a descendant of $X$ if and only if there is at least one minimal possible directed path from $X$ to $A$ in $M$.*

*Proof.* $\Leftarrow$ Consider there is a minimal possible directed path from $X$ to $A$ in $M$. Since the marks at $X$ are certain in $M$, the corresponding path can only start with $X \rightarrow$. To avoid the generation of unshielded colliders, the path could only be directed from $X$ to $A$. Hence in any an MAG $\mathcal{M}$ consistent to $M$, $A$ is the descendant of $X$. Hence in $\mathcal{M}$, $A$ is the descendant of $X$.

$\Rightarrow$ Since $\mathcal{M}$ is an MAG consistent to $M$, there must be a possible directed path from $X$ to $A$ in $M$. Then by Lemma B.1 of Zhang [13], there is a minimal possible directed path from $X$ to $A$ in $M$. $\qquad\square$

**Theorem 2.** Let $M$ be a local MAG of $X$ and $\mathcal{M}$ be an MAG consistent to $M$. Suppose $V$ ($V \neq X, Y$) is a variable in $\mathcal{M}$. If D-SEP$(X, Y, \mathcal{M}_{\underline{X}}) \cap \mathrm{De}(X, \mathcal{M}) = \emptyset$, then $V \in$ D-SEP$(X, Y, \mathcal{M}_{\underline{X}})$ holds if and only if there is at least one collider path from $X$ to $V$ starting by an arrowhead at $X$ in $M$ such that each variable except for $X$ on the path is an ancestor of $X$ or $Y$ in $\mathcal{M}$.

*Proof.* The proof is easy by Lemma 18. Combining the definition of D-SEP$(X, Y, \mathcal{M})$ and $\mathcal{M}_{\underline{X}}$, we could trivially conclude that $V \in$ D-SEP$(X, Y, \mathcal{M}_{\underline{X}})$ holds if and only if there is at least one collider path from $V$ to $X$ in $M$ such that in $\mathcal{M}$ each variable except for $X$ on the path is an ancestor of $X$ or $Y$, and the path *does not* contain $X \xrightarrow{p}$. Hence the "if" statement is evident. For the "only if" statement, considering the condition D-SEP$(X, Y, \mathcal{M}_{\underline{X}}) \cap \mathrm{De}(X, \mathcal{M}) = \emptyset$, if there is a collider path from $X$ to $V$ starting by an impure edge in $M$ such that in $\mathcal{M}$ each variable except for $X$ on the path is an ancestor of $X$ or $Y$, we could conclude the variable adjacent to $X$ on the path belongs to D-SEP$(X, Y, \mathcal{M}_{\underline{X}})$. According to the path we also know such a variable is the descendant of $X$, which contradicts the condition D-SEP$(X, Y, \mathcal{M}_{\underline{X}}) \cap \mathrm{De}(X, \mathcal{M}) = \emptyset$. Hence the path cannot start by an impure edge from $X$. Combining with the fact that all the marks at $X$ are not circle in the local MAG $M$, the mark at $X$ can only be arrowhead. We get the desired conclusion. $\qquad\square$

## C.2 Proofs for Theorem 3

**Definition 4** (PD-SEP$(X, Y, M)$)**.** Let $M$ be a local MAG of $X$ and $\mathcal{M}$ be an MAG consistent to $M$. Variable $V \in$ PD-SEP$(X, Y, M)$ if and only if $V \in \mathrm{PossAn}(Y, M) \backslash \mathrm{De}(X, \mathcal{M})$[5] and there exists a collider path between $X$ and $V$ in $M$, where each non-endpoint variable is an ancestor of $X$ or $Y$ in $M$ but not a descendant of $X$ in $\mathcal{M}$.

Note that PD-SEP$(X, Y, M)$ is not necessarily a set that contains $\mathbf{D}$.

**Definition 5** (Critical variable for $(X, Y)$)**.** In a local MAG $M$ with a path $X \leftrightarrow F_1 \leftrightarrow \cdots \leftrightarrow F_{t-1} \leftrightarrow F_t$ or $X \leftrightarrow F_1 \leftrightarrow \cdots \leftrightarrow F_{t-1} \leftarrow\!\!\circ F_t$, $t \geq 1$, where $F_1, \cdots, F_{t-1} \in$ PD-SEP$(X, Y, M)$, $F_t$ is called a *critical variable* for $(X, Y)$ if there is a non-empty variable set $\mathbf{S}$ relative to $F_t$ defined as follows: $S \in \mathbf{S}$ if and only if in $M$ (1) $S$ is a child of $X, F_1, \cdots, F_{t-1}$, (2) there is $F_t \circ\!\!-\!\!* S$, (3) $S$ is at one minimal possible directed path from $F_t$ to $Y$, and no variable on the path belongs to PD-SEP$(X, Y, M)$. Each circle at $F_t$ on the edge with $F_{t-1}$ or $S \in \mathbf{S}$ is called a *critical mark* of $F_t$.

**Theorem 3.** Let $M$ be a local MAG of $X$ based on a PMAG $\mathcal{P}$. Then condition (1) below is sufficient for condition (2):

(1) there is no critical variable for $(X, Y)$ in $M$.

(2) for any an MAG $\mathcal{M}$ consistent to $M$ such that ($a$) D-SEP$(X, Y, \mathcal{M}_{\underline{X}}) \cap \mathrm{De}(X, \mathcal{M}) = \emptyset$, ($b$) $X \in \mathrm{An}(Y, \mathcal{M})$, it holds that $\mathbf{A}_{\mathcal{M}} = \mathbf{A}'_{\mathcal{M}}$, where $\mathbf{A}_{\mathcal{M}}$ denotes the MCS regarding D-SEP$(X, Y, \mathcal{M}_{\underline{X}})$ in $\mathcal{M}$ and $\mathbf{A}'_{\mathcal{M}}$ denotes the MCS regarding PD-SEP$(X, Y, M)$ in $\mathcal{M}$;

*Proof.* Let $\mathcal{M}$ be any an MAG consistent to $M$ such that ($a$) D-SEP$(X, Y, \mathcal{M}_{\underline{X}}) \cap \mathrm{De}(X, \mathcal{M}) = \emptyset$, ($b$) $X \in \mathrm{An}(Y, \mathcal{M})$. Denote D-SEP$(X, Y, \mathcal{M}_{\underline{X}})$ by $\mathbf{D}$ and PD-SEP$(X, Y, M) \backslash \mathbf{D}$ by $\mathbf{K}$. We first prove that if there does not exist a critical variable for $(X, Y)$ in $M$, then $\mathbf{K} \perp Y | \mathbf{D}, X$ in $\mathcal{M}$. Note that here we aim to prove this result holds for *each* MAG satisfying the conditions ($a$) and ($b$) above. Our main proof strategy here is to attempt to construct an MAG $\mathcal{M}$ in which $\mathbf{K} \not\perp Y | \mathbf{D}, X$. By considering all possible MAG $\mathcal{M}$, we conclude the desired result. Then, we prove that if $\mathbf{K} \perp Y | \mathbf{D}, X$, then $\mathbf{D} \subseteq$ PD-SEP$(X, Y, M)$. Combining the two results, we finally conclude that if there does not exist a critical variable for $(X, Y)$ in $M$, then all the MAGs consistent to $M$ induce the same MCS. For brevity, below we omit the conditions ($a$) and ($b$), but in fact, when we say an MAG $\mathcal{M}$ consistent to $M$, we restrict that $\mathcal{M}$ satisfies the conditions ($a$) and ($b$).

As shown in the outline, we prove in the first section that if there does not exist a critical variable for $(X, Y)$ in $M$, then given any an MAG $\mathcal{M}$ consistent to $M$, for any a variable $F_t \in$ PD-SEP$(X, Y, M) \backslash \mathbf{D}$, it holds that $F_t \perp Y | \mathbf{D}, X$.

---

[5]Note all the descendants of $X$ in $\mathcal{M}$ consistent to $M$ are knowable in $M$, which is detailed by Lemma 19 in Appendix C.1. Hence PD-SEP$(X, Y, M)$ can be obtained based on $M$ without the further knowledge about $\mathcal{M}$.

According to the definition of $\mathbf{D}$ and PD-SEP$(X, Y, M)$, we list the condition as follows: (1) there is a collider path $X \leftrightarrow F_1 \leftrightarrow \cdots \leftrightarrow F_{t-1} \leftrightarrow ($ or $\leftarrow\!\circ)F_t$ in $M$, where $X, F_1, \cdots, F_{t-1}$ are ancestors of $Y$ in $M$, while $F_t$ is a possible ancestor but not ancestor of $Y$ in $M$, is not an ancestor of $Y$ in $\mathcal{M}$. It trivially concludes that the edge between $F_{t-1}$ and $F_t$ is bi-directed. And by Prop. 1, we see that $\mathbf{D}$ m-separates all generalized back-door paths from $X$ to $Y$ relative to $W$, where $W$ is the true purity matrix. Since $F_t$ is a possible ancestor of $Y$ in $M$, there must be a minimal possible directed path from $F_t$ to $Y$ in $M$. Without loss of generality, we suppose the minimal possible directed path $F_t \circ\!\!-\!\!* S \cdots Y$ ($S$ could be $Y$).

**A. If there does not exist a critical variable for $(X, Y)$ in $M$, then $\mathbf{K} \perp Y|\mathbf{D}, X$ in each $\mathcal{M}$ consistent to $M$.**

Suppose a variable $F_t \in \mathbf{K}$ such that $F_t \not\perp Y|\mathbf{D}, X$ in $\mathcal{M}$. Given the variable $F_t$ and local MAG $M$, we say an MAG $\mathcal{M}$ is legal if $\mathcal{M}$ is consistent to $M$ and $F_t \in \mathbf{K}$ in $\mathcal{M}$.

At first, we give two supporting results in A.1 and A.2.

**A.1. If there is an active path relative to $(\mathbf{D}, X)$ from $F_t$ to $U$ without colliders in $\mathcal{M}$, where $U$ is an ancestor of $Y$ in $\mathcal{M}$, then there is a minimal active path without colliders relative to $(\mathbf{D}, X)$ from $F_t$ to $U$ in $\mathcal{M}$.**

Evidently that there exists a minimal path from $F_t$ to $U$ in $\mathcal{M}$. The main part is to prove that path is active. For the sake of contradiction, we suppose for an active path $\mathcal{L}$ from $F_t$ to $U$ relative to $(\mathbf{D}, X)$ in $\mathcal{M}$, the path is not minimal and there is a minimal path $\mathcal{L}_1$ of $\mathcal{L}$ that is m-separated by $(\mathbf{D}, X)$ in $\mathcal{M}$. Since there do not exist colliders in the path $\mathcal{L}$, it is evident that the path cannot go through the variables in $(\mathbf{D}, X)$, otherwise the path will be m-separated by such variables. Considering $\mathcal{L}_1$ is a minimal path of $\mathcal{L}$, both $\mathcal{L}$ and $\mathcal{L}_1$ do not go through $(\mathbf{D}, X)$. If $\mathcal{L}_1$ is directed, then evidently $\mathcal{L}$ is directed due to no colliders in it. There are no colliders in $\mathcal{L}$ so that the path $\mathcal{L}$ is m-separated by $(\mathbf{D}, X)$ if the minimal path $\mathcal{L}_1$ is m-separated by $(\mathbf{D}, X)$. Next we mainly consider the situation that $\mathcal{L}_1$ is not directed. If $\mathcal{L}_1$ is m-separated by $(\mathbf{D}, X)$, $\mathcal{L}$ and $\mathcal{L}_1$ could only be like $F_t *\!\!-\!\!* \cdots *\!\!-\!\!* s_k \to s_{k+1} *\!\!-\!\!* \cdots *\!\!-\!\!* s_l *\!\!-\!\!* \cdots *\!\!-\!\!* U$ and $F_t *\!\!-\!\!* \cdots *\!\!-\!\!* s_k \leftarrow\!\!* s_l *\!\!-\!\!* \cdots *\!\!-\!\!* U$ (here $F_t$ and $U$ can be swapped), where $l \geq k + 2$ and $s_k$ is a collider in $\mathcal{L}_1$ but not a collider in $\mathcal{L}$. Since there are no colliders in $\mathcal{L}$, the sub-path from $s_k$ to $s_l$ in $\mathcal{L}_1$ could only be $s_k \to s_{k+1} \to \cdots \to s_l$. In such a case, no matter the edge between $s_k$ and $s_l$ is either $s_k \leftarrow s_l$ or $s_k \leftrightarrow s_l$, it is against the ancestral property of $\mathcal{M}$. Hence if there is an active path from $F_t$ to $U$ relative to $(\mathbf{D}, X)$, there is at least one minimal active path from $F_t$ to $U$ relative to $(\mathbf{D}, X)$.

**A.2. Some properties about the minimal paths from $F_t$ to $U$ without colliders in $\mathcal{M}$ whose corresponding path in $M$ begins with $F_t \circ\!\!-\!\!* S_1$, where $U$ is an ancestor of $Y$ in $\mathcal{M}$.**

For the corresponding paths in $\mathcal{M}$ of the minimal paths from $F_t$ to $U$ without colliders beginning with $F_t \circ\!\!-\!\!* S_1$ in $M$, there are five types $\mathcal{L}_1, \mathcal{L}_2, \mathcal{L}_3, \mathcal{L}_4, \mathcal{L}_5$. For $\mathcal{L}_1$, the path in $\mathcal{M}$ is as $F_t \leftrightarrow S_1 \to \cdots U$. For $\mathcal{L}_2$, the path in $\mathcal{M}$ is as $F_t \leftarrow S_1 \leftarrow \cdots \leftarrow S_k \to S_{k+1} \cdots \to U$, where $k \geq 1$. For $\mathcal{L}_3$, the path in $\mathcal{M}$ is as $F_t \leftarrow S_1 \leftarrow \cdots \leftarrow S_k \leftrightarrow S_{k+1} \cdots \to U$, where $k \geq 1$. For $\mathcal{L}_4$, the path in $\mathcal{M}$ is as $F_t \to S_1 \to \cdots \to U$. For $\mathcal{L}_5$, the path in $\mathcal{M}$ is as $F_t \leftarrow S_1 \leftarrow \cdots \leftarrow U$. $\mathcal{L}_4$ and $\mathcal{L}_5$ are evidently impossible. If there is a path as $\mathcal{L}_4$, $F_t \in \text{Anc}(Y, \mathcal{M})$, thus $F_t \in (\mathbf{D}, X)$, which contradicts the condition $F_t \in \mathbf{K}$. If there is a path as $\mathcal{L}_5$, $F_t$ is a descendant of $Y$, thus a descendant of $X$. According to Lemma 19, we could identify all the variables in $\text{De}(X, \mathcal{M})$ in $M$. Hence $F_t$ could be identified to be a descendant of $X$ in $M$, which contradicts the condition $F_t \in \text{PD-SEP}(X, Y, M)$. In the following, we first show in A.2.1 that the sets of paths as $\mathcal{L}_2$ and $\mathcal{L}_3$ are also empty if the path does not go through the variables in $(\mathbf{D}, X)$.

**A.2.1 Both the sets of paths as $\mathcal{L}_2$ and $\mathcal{L}_3$ are empty if the path does not go through the variables belonging to $(\mathbf{D}, X)$.**

Consider the paths as $\mathcal{L}_2$ at first. If $S_1$ and $F_{t-1}$ are not adjacent, since there is an unshielded collider $F_{t-1} \leftrightarrow F_t \leftarrow\!\!* S_1$, we could identify the two arrowheads here by FCI algorithm in $M$, which contradicts the circle at $F_t$ in $M$.

Since $S_1 \notin (\mathbf{D}, X)$, there cannot be an edge $S_1 \to F_{t-1}$. Hence the edge between $S_1$ and $F_{t-1}$ is as $F_{t-1} *\!\!\to S_1$ in $\mathcal{M}$. And due to the ancestral property, the edge could only be $F_{t-1} \leftrightarrow S_1$ given the fact $F_{t-1} \leftrightarrow F_t$ and $S_1 \to F_t$. Then we consider the variable $S_2$. If $S_2$ and $F_{t-1}$ are not adjacent, then $S_2, S_1, F_{t-1}$ forms an unshielded collider, so that $S_2 *\!\!\to S_1 \leftarrow\!\!* F_{t-1}$ could be identified in $M$. Due to the minimal path, $S_1 \to F_t$ could also be identified. According to the ancestral property, we could identify $S *\!\!\to F_t$ in $M$ since $S_1 \to F_t$ and $S *\!\!\to S_1$, which contradicts the circle at $F_t$ in minimal possible directed path from $F_t$ to $Y$ across $S$ in $M$. Hence $S_2$ and $F_{t-1}$ are adjacent. Considering $S_2 \to S_1$ and $S_1 \leftrightarrow F_{t-1}$ in $\mathcal{M}$, the edge between $S_2$ and $F_{t-1}$ is as $F_{t-1} \leftarrow\!\!* S_2$. Since the path does not go through $(\mathbf{D}, X)$, $S_2 \notin (\mathbf{D}, X)$. Hence the edge can only be $F_{t-1} \leftrightarrow S_2$.

We could repeat the process above for $S_3, S_4 \cdots$ until for $S_k$. Similar to the proof above, we see that the edge between $S_t$ and $F_{t-1}$ is $S_t \leftrightarrow F_{t-1}$. In this case, there is a collider path $X \leftrightarrow F_1 \cdots F_{t-1} \leftrightarrow S_k$, and we also know $S_k$ is an ancestor of $U$, thus is an ancestor of $Y$. Hence $S_k \in (\mathbf{D}, X)$ in $\mathcal{M}$, which contradicts the

condition that the path from $F_t$ to $U$ does not go through the variables in $(\mathbf{D}, X)$. Hence we conclude that the set of paths as $\mathcal{L}_2$ is empty if the path does not go through the variable in $(\mathbf{D}, X)$.

Then we consider the paths as $\mathcal{L}_3$. The proof of this part is quite similar to that for the paths as $\mathcal{L}_2$. To prevent from discovering the arrowheads at $F_t$ on the edge between $F_t$ and $S$, there must be bi-directed edges between $F_{t-1}$ and $S_1, S_2, \cdots, S_k$. If $S_{k+1}$ is not adjacent to $F_{t-1}$, a v-structure forms so that we could identify $S_{k+1} \ast\!\!\rightarrow S_k$. Due to the fact that the path is minimal and without colliders in $M$, we could learn the arrowhead at $F_t$, which contradicts with the circle at $F_t$. If $S_{k+1} \ast\!\!\rightarrow F_{t-1}$, $S_{k+1} \in \mathbf{D}$ since $S_{k+1}$ is an ancestor of $U$, thus is an ancestor of $Y$. This contradicts with the fact that path does not go through the variable in $(\mathbf{D}, X)$ in $\mathcal{M}$. Hence the edge can only be $S_{k+1} \leftarrow F_{t-1}$. However, there is an inducing path comprised of $S_{k+1} \leftrightarrow S_k \leftrightarrow F_{t-1} \leftrightarrow S_{k-1}$, $F_{t-1} \rightarrow S_{k+1}$, and $S_k \rightarrow S_{k-1}$. To satisfy the maximal property, there is a bi-directed edge between $S_{k+1}$ and $S_{k-1}$, which contradicts with the condition that the path $\mathcal{L}_3$ is minimal. Hence we conclude that the set of paths as $\mathcal{L}_3$ is empty if the path does not go through the variable in $(\mathbf{D}, X)$.

**A.2.2 If the edge between $F_{t-1}$ and $S_1$ is as $F_{t-1} \ast\!\!\rightarrow S_1$ in $\mathcal{M}$, $U$ is an ancestor of $Y$ in $\mathcal{M}$ and $S_1$ is the variable adjacent to $F_t$ in one minimal possible directed path from $F_t$ to $U$ in $M$, then there exists at least one variable among $X, F_1, \cdots, F_{t-1}$ that has a bi-directed edge with $S_1$ in $\mathcal{M}$.**

If the edge between $F_{t-1}$ and $S_1$ is bi-directed, we get the desired conclusion directly. If the edge between them are $F_{t-1} \rightarrow S_1$, since $F_{t-1} \leftrightarrow F_t$ and $F_t$ is not an ancestor of $Y$, the edge between $F_t$ and $S_1$ is bi-directed. And we consider the edge between $S_1$ and $F_{t-2}$. Since $F_{t-2} \leftrightarrow F_{t-1}$ and $F_{t-1} \rightarrow S_1$, the edge is as $F_{t-2} \ast\!\!\rightarrow S_1$ in $\mathcal{M}$. If it is bi-directed, the desired conclusion is directly obtained. If it is $F_{t-2} \rightarrow S_1$, we consider the edge between $F_{t-3}$ and $S_1$. We repeat the process above. If there is no variable in $F_1, \cdots, F_{t-1}$ with a bi-directed edge with $S_1$, there must be an edge $X \rightarrow S_1$ in $\mathcal{M}$. However, since the edge $F_t \leftrightarrow S_1$ and $S_1$ is located at one minimal possible directed path from $F_t$ to $U$, the corresponding path in $\mathcal{M}$ of this minimal possible directed path can only be $F_t \leftrightarrow S_1 \rightarrow \cdots \rightarrow U$ in order to avoid the generation of v-structure. Since $X \rightarrow S_1$, all the variables from $S_1$ to $Y$ are descendants of $X$. Since the mark at $X$ is known in $M$, we could identify the tail at $X$ on the edge between $X$ and $S_1$ in $M$. In this case, there exists at least one critical variable among $F_1, F_2, \cdots, F_t$, which contradicts the condition that there does not exist a critical variable in $M$. Thus, it is impossible that $V \rightarrow S_1$ for $\forall V \in \{X, F_1, \cdots, F_{t-1}\}$ in $\mathcal{M}$. Hence there must be a bi-directed edge with $S_1$ among the variables in $X, F_1, \cdots, F_{t-1}$.

With the results in A.1 and A.2, we prove the main results in the following. In the beginning, we prove the desired results by showing that it is impossible to construct a legal MAG with an active path from $F_t$ to $Y$. We divide all possible paths from $F_t$ to $Y$ in $\mathcal{M}$ into two classes. The first class is comprised of all the paths without colliders in $\mathcal{M}$. And we prove in A.3 that all such paths cannot be active relative to $(\mathbf{D}, X)$ in $\mathcal{M}$. The second class is comprised of all the paths with colliders in $\mathcal{M}$. We prove in A.4 that all such paths cannot be active relative to $(\mathbf{D}, X)$ in $\mathcal{M}$.

**A.3. There do not exist active paths relative to $(\mathbf{D}, X)$ from $F_t$ to $Y$ without colliders in any legal MAG $\mathcal{M}$.**

Suppose an active path from $F_t$ to $Y$ without colliders in a legal MAG $\mathcal{M}$. By A.1, there is an active minimal path without colliders from $F_t$ to $Y$ in $\mathcal{M}$. Since there are no colliders in this path, the active path cannot go through the variables in $(\mathbf{D}, X)$, otherwise the path is m-separated by $(\mathbf{D}, X)$. By A.2.1, we see that the active minimal paths without colliders cannot be as $\mathcal{L}_2$ or $\mathcal{L}_3$. The only possible paths are like $\mathcal{L}_1$. However, in this case by A.2.2 there exists at least one variable $F_s \in \{X, F_1, \cdots, F_{t-1}\}$ with $F_s \leftrightarrow S_1$. And because $S_1$ is an ancestor of $U$ in $\mathcal{L}_1$, thus is an ancestor of $Y$, thus $S_1 \in (\mathbf{D}, X)$, in which case the path without colliders $F_t, S_1, \cdots, Y$ is m-separated by $(\mathbf{D}, X)$, which contradicts the active path. Thus there is not a path as $\mathcal{L}_1$. Hence we get the desired conclusion that there do not exist active paths from $F_t$ to $Y$ without colliders in any a legal MAG $\mathcal{M}$.

**A.4. There do not exist active paths relative to $(\mathbf{D}, X)$ from $F_t$ to $Y$ with colliders in any legal MAG $\mathcal{M}$.**

This part is a bit complex. Before proposing the proof, we define the *distance between $F_t$ and $(\mathbf{D}, X)$*. For any variable $F_t \in \mathbf{K}$, there must be some minimal possible directed paths from $F_t$ to $Y$ in $M$ according to the definition of $\mathbf{K}$. We say that the distance between $F_t$ and $(\mathbf{D}, X)$ is $k$ if:

(1) There is one minimal possible directed path from $F_t$ to $Y$, where the nearest variable to $F_t$ that belongs to $(\mathbf{D}, X)$ is with distance $k$ to $F_t$. In another word, the minimal possible directed path is such as $F_t, C_1, C_2, \cdots, C_k, \cdots, Y$, where $C_1, \cdots, C_{k-1} \notin (\mathbf{D}, X)$ and $C_k \in (\mathbf{D}, X)$.

(2) There do not exist minimal possible direct paths from $F_t$ to $Y$, where the nearest variable to $F_t$ that belongs to $(\mathbf{D}, X)$ is with distance less than $k$ to $F_t$.

We prove the desired conclusion by mathematical induction. In A.4.a, we prove that for any $F_t \in \mathbf{K}$ that has distance 1 to $(\mathbf{D}, X)$, all the paths from $F_t$ to $Y$ with colliders are m-separated by $(\mathbf{D}, X)$ in any one legal $\mathcal{M}$. Combining this result with A.3, we conclude $F_t \perp Y | \mathbf{D}, X$ for $F_t \in \mathbf{K}$ that has distance 1 to $(\mathbf{D}, X)$

in $\mathcal{M}$. Then in A.4.b, we prove that if for any $F_t \in \mathbf{K}$ that has distance $k-1$ to $(\mathbf{D}, X)$ in $\mathcal{M}$ it holds that $F_t \perp Y | \mathbf{D}, X$, then for any $F_t \in \mathbf{K}$ that has distance $k$ to $(\mathbf{D}, X)$ in $\mathcal{M}$, all the paths from $F_t$ to $Y$ with colliders are m-separated by $(\mathbf{D}, X)$. Also combining this result with A.3, we conclude $F_t \perp Y | \mathbf{D}, X$ for $F_t \in \mathbf{K}$ that has distance $k$ to $(\mathbf{D}, X)$ in $\mathcal{M}$. We thus prove the desired result that for any variable $F_t \in \mathbf{K}$, it holds that $F_t \perp Y | \mathbf{D}, X$.

### A.4.a. For any $F_t \in \mathbf{K}$ that has distance $1$ to $(\mathbf{D}, X)$, all the paths from $F_t$ to $Y$ with colliders are m-separated by $(\mathbf{D}, X)$ in any one legal $\mathcal{M}$.

Since the distance between $F_t$ and $(\mathbf{D}, X)$ is 1, there is at least one minimal possible directed path $F_t, S, \cdots, Y$, where $S \in (\mathbf{D}, X)$ in $\mathcal{M}$. For the sake of contradiction, we suppose there is an active path relative to $(\mathbf{D}, X)$ from $F_t$ to $Y$ with colliders in $\mathcal{M}$. We denote the collider closest to $F_t$ in this path by $C$. Since the path is active relative to $(\mathbf{D}, X)$, the sub-path from $F_t$ to $C$ that contains no colliders is also active relative to $(\mathbf{D}, X)$. And it is evident that this sub-path cannot go through the variables in $(\mathbf{D}, X)$, otherwise the path is m-separated by $(\mathbf{D}, X)$.

Since $F_t$ is not a variable in $(\mathbf{D}, X)$, the edge between $F_t$ and $F_{t-1}$ is bi-directed, and the mark at $F_t$ on the edge between $F_t$ and $S$ is arrowhead, otherwise there is a directed path from $F_t$ to $Y$ across $S$ in $\mathcal{M}$, which concludes that $F_t \in (\mathbf{D}, X)$. Since there is a circle at $F_t$ on the edge between $F_t$ and $S$ in $M$, $F_{t-1}$ and $S$ are adjacent in $\mathcal{M}$.

By the result of A.1, since $C \in \text{Anc}(Y, \mathcal{M})$, there exist some active minimal paths without colliders from $F_t$ to $C$ relative to $(\mathbf{D}, X)$ in $\mathcal{M}$. Note in the following we may omit "relative to $(\mathbf{D}, X)$". That is, if we do not speak intentionally, "active path" refers to "active path relative to $(\mathbf{D}, X)$" . We will consider such active minimal paths and construct the contradiction. We denote such an active minimal path by $\mathcal{L}$. It is easy to see that the mark at $F_t$ is arrowhead, otherwise $F_t$ is an ancestor of $C$ thus an ancestor of $Y$, which contradicts the condition that $F_t \notin \mathbf{D}$. We separate all possible situations for $\mathcal{L}$ into three classes. In the first class, $C$ and $F_t$ are adjacent, i.e. the path is $F_t \leftrightarrow C$. We denote it by $\mathcal{L}_1$. Note $F_t \to C$ is impossible otherwise $F_t \in \mathbf{D}$, which contradicts the condition $F_t \in \mathbf{K}$. In the second class, we suppose the minimal active path is as $F_t \leftarrow S_1 \leftarrow \cdots \leftarrow S_k \to S_{k+1} \to \cdots \to C, k \geq 1$. We denote it by $\mathcal{L}_2$. In the third class, we suppose the minimal active path is as $F_t \leftarrow S_1 \leftarrow \cdots \leftarrow S_k \leftrightarrow S_{k+1} \to \cdots \to C, k \geq 1$. We denote it by $\mathcal{L}_3$.

### A.4.a.1 There does not exist an active path as $\mathcal{L}_1$ in $\mathcal{M}$.

If there is such an active path as $\mathcal{L}_1$, there exists an active path from $F_t$ to $Y$ where there is a sub minimal collider path beginning with $F_t$, that is the path is as $F_t \leftrightarrow C_1 \leftrightarrow C_2 \leftrightarrow \cdots \leftrightarrow C_{t-1} \leftarrow\!* C_t \cdots Y$, where $C_t$ is the first variable that is not collider in the path and there are no bi-directed edges between $C_i$ and $C_j$ for $|j - i| \geq 2, 1 \leq i, j \leq t$.

Evidently $F_{t-1}$ and $S$ are adjacent and $C_1$ and $S$ are also adjacent, otherwise the arrowhead on $F_t \leftarrow\!* S$ could be identified due to the unshielded collider.

### A.4.a.1.1 If an active path as $\mathcal{L}_1$ exists, then $C_1$ and $F_{t-1}$ are adjacent in $\mathcal{M}$.

At first, we consider the situation that $C_1$ and $F_{t-1}$ are not adjacent. We will construct a contradiction by proving $C_t *\!\to C_{t-1} \leftrightarrow \cdots \leftrightarrow C_1 \leftrightarrow F_t \leftrightarrow S$ is a minimal collider path.

Since there is an unshielded collider $F_{t-1} \leftrightarrow F_t \leftarrow\!* C_1$, we could identify the two arrowheads here by FCI algorithm in $M$. And we could see that there must be a sub-structure $F_{t-1} *\!\to S \leftarrow\!* C_1$ in $\mathcal{M}$, otherwise the mark at $F_t$ on the edge between $F_t$ and $S$ could be identified to be arrowhead by Rule 3 of Zhang [13]. Next, We consider the edge between $S$ and $F_{t-1}$. By A.2.2, there exists an bi-directed edge between $S$ and some variable among $X, F_1 \cdots, F_{t-1}$ in $\mathcal{M}$. Without loss of generality, we suppose the variable with a bi-directed edge with $S$ that is closest to $F_t$ by $F_s$. In this case, there is a collider path $X \leftrightarrow F_1 \cdots F_s \leftrightarrow S \leftarrow\!* C_1$. Here the edge between $S$ and $C_1$ could only be $S \leftarrow C_1$, otherwise if $F_t \leftrightarrow C_1 \cdots Y$ is active relative to $\mathbf{D}$ in $\mathcal{M}$, the path $X \leftrightarrow F_1 \leftrightarrow \cdots \leftrightarrow F_s \leftrightarrow S \leftrightarrow C_1 \cdots Y$ is also active relative to $\mathbf{D}$ in $\mathcal{M}$, which contradicts the condition that $\mathbf{D}$ m-separates all generalized back-door paths from $X$ to $Y$.

Similarly, it is easy to prove that for all variables $C_i, 1 \leq i \leq t-1$ there is not an edge $C_i \leftrightarrow S$, otherwise the path $X \leftrightarrow F_1 \leftrightarrow \cdots \leftrightarrow F_s \leftrightarrow S \leftrightarrow C_i \cdots Y$ is active relative to $\mathbf{D}$ in $\mathcal{M}$, which contradicts the condition that $\mathbf{D}$ m-separates all generalized back-door paths. Hence we conclude that the path $C_t *\!\to C_{t-1} \leftrightarrow \cdots \leftrightarrow C_1 \leftrightarrow F_t \leftrightarrow S$ is a minimal collider path. By Lemma 18 (see the detailed proof process), we could identify all the colliders in the minimal collider path by FCI algorithm, that is we could identify the arrowhead at $F_t$ in $M$, which contradicts the possible directed path condition.

### A.4.a.1.2. If an active path as $\mathcal{L}_1$ exists, then $C_1$ and $F_{t-1}$ cannot be adjacent in $\mathcal{M}$.

Then we consider the condition that $C_1$ and $F_{t-1}$ are adjacent. Evidently the edge cannot be bi-directed, otherwise given the active path $F_t \leftrightarrow C_1 \leftrightarrow \cdots \leftrightarrow C_t \cdots Y$ relative to $\mathbf{D}$, the path $X \leftrightarrow F_1 \leftrightarrow \cdots \leftrightarrow F_{t-1} \leftrightarrow C_1 \leftrightarrow \cdots \leftrightarrow C_t \cdots Y$ is active relative to $\mathbf{D}$, which contradicts the condition that $\mathbf{D}$ m-separates all the generalized back-door paths.

We first consider the condition that $F_{t-1} \leftarrow C_1$. We discuss the relation between $C_2$ and $F_{t-1}$. If $C_2$ is adjacent to $F_{t-1}$, then by ancestral property it holds $C_2 *\to F_{t-1}$ in $\mathcal{M}$. And similar to the proof for that there is not bi-directed edge between $F_{t-1}$ and $C_1$, if there is $C_2 \leftrightarrow F_{t-1}$, there is an active path $X \leftrightarrow F_1 \leftrightarrow \cdots \leftrightarrow F_{t-1} \leftrightarrow C_2 \leftrightarrow \cdots \leftrightarrow C_t \cdots Y$ relative to $\mathcal{D}$ in $\mathcal{M}$, contradicting the condition that $\mathbf{D}$ m-separates all the generalized back-door paths. Hence the edge could only be $C_2 \to F_{t-1}$ in $\mathcal{M}$. In addition, for the variable $C_t$, it is impossible that there is an edge $C_t \to F_{t-1}$, otherwise $C_t \in \mathbf{D}$, which contradicts the condition that $\mathcal{L}_1$ is active relative to $\mathbf{D}$ and $C_t$ is not a collider in the path. Thus if all the variables among $C_2, C_3, \cdots, C_{t-1}$ are adjacent to $F_{t-1}$, then $C_t$ is not adjacent to $F_{t-1}$.

Hence, suppose the first variable $C_s$ from $F_t$ to $C_t$ that is no adjacent to $F_{t-1}$. Then we consider the minimal collider path $C_s *\to C_{s-1} \leftrightarrow \cdots \leftrightarrow C_1 \leftrightarrow F_t$ from $C_s$ to $F_t$. Since $F_{t-1} \leftrightarrow F_t$ and $C_i \to F_{t-1}, 1 \le i \le s-1$, the path $C_s *\to C_{s-1} \leftrightarrow \cdots \leftrightarrow C_1 \leftrightarrow F_t \leftrightarrow F_{t-1}$ is a minimal collider path from $C_s$ to $F_{t-1}$. By Lemma 18 (see the detailed proof process), all the colliders could be identified in $M$. And all directed edges $C_i \to F_{t-1}, 1 \le i \le s-1$ could be identified by Rule 4 of Zhang [13].

In the following we consider the edge between $S$ and $C_i, 1 \le i \le s$. Here we only construct the contradiction when all the variables between $F_t$ and $C_s$ are adjacent to $S$. It is easy to construct a contradiction if there are some variables not adjacent to $S$, we thus leave them to readers. We discuss the edge between $F_{t-1}$ and $S$.

If the edge is as $F_{t-1} *\to S$, we see the edge between $S$ and $C_1$ is as $S \leftarrow *C_1$ in $\mathcal{M}$. Then similar to the last part A.4.a.1.1 we could construct a contradiction. Hence we only consider $S \to F_{t-1}$ in $\mathcal{M}$ in the following.

We first prove that there is some variable $C_j, 2 \le j \le s-1$ such that $C_i \to F_t, 1 \le i \le j-1$ and $C_j$ is not adjacent to $F_t$. If $F_t$ and $C_2$ are not adjacent, then $C_2$ is such a variable. If they are adjacent, since the minimal collider path between $C_s$ and $F_t$ and $C_3 \to F_{t-1} \leftrightarrow F_t$, the edge between $F_t$ and $C_3$ could only be $F_t \leftarrow C_3$. Repeat the process for $C_4, C_5, \cdots, C_s$. If there is $C_i \to F_t, 2 \le i \le s-1$ and there is also an edge $C_s \to F_t$, it contradicts the minimal collider path condition. Hence we get the desired conclusion.

We consider the sub-structure comprised of $S, F_t, C_{j-1}, C_j$. The edge $C_{j-1} \to F_t$ is identifiable in $M$, similar to the proof process for that $C_{j-1} \to F_{t-1}$ is identifiable so that we skip this part. Hence we discuss whether $F_t, S, C_j$ form an unshielded collider. If $F_t, S, C_j$ is not an unshielded collider, we consider the sub-structure comprised of $F_t, S, C_j, C_{j-1}$. We have proved before that the edge $C_j *\to C_{j-1}$ and $C_{j-1} \to F_t$ could be identified in $M$. We prove $S *\to F_t$ could be identified in $M$ in the following. The reason is, if there is an edge $S \leftarrow F_t$, there is an edge $C_{j-1} \to S$ in $\mathcal{M}$ due to the ancestral property and thus there is an edge $C_{j-1} *\to S$ in $\mathcal{M}$, which contradicts the condition that $F_t, S, C_j$ is not an unshielded collider. Hence we see that there cannot be an edge as $F_t *\to S$ in $\mathcal{M}$. Due to the completeness of FCI [13], the mark at $F_t$ on the edge between $F_t$ and $S$ is identifiable in the PAG. Thus the arrowhead at $F_t$ is known in $M$, which contradicts the possible directed path from $F_t$ to $Y$ across $S$. If they form an unshielded collider, we identify $S \leftarrow *F_t$ in $M$. If $C_s$ is not adjacent to $F_t$, we further discuss the edge between $S$ and $C_s$. If $C_s, S, F_t$ does not form an unshielded collider, $S \to C_s$ is identified in $M$ by Rule 1 of Zhang [13], thus $S *\to C_{s-1}$ is identified by Rule 2 of Zhang [13], and $S \to F_{t-1}$ is identified by Rule 4 of Zhang [13], and $S *\to F_t$ is identified by Rule 2 of Zhang [13], which contradicts the possible directed path from $F_t$ to $Y$ across $S$. If they form an unshielded collider, we could identify $C_s \to S$, thus identify $S \to F_{t-1}$ and $S *\to F_t$ in $M$, which also contradicts the possible directed path from $F_t$ to $Y$ across $S$. If $C_s$ is adjacent to $F_t$, the edge cannot be as $F_t \to C_s$, in which case $F_t \in \mathbf{D}$ since $C_s$ is an ancestor of $Y$. It is also not bi-directed, which contradicts the minimal collider path condition. Thus it is as $F_t \leftarrow C_s$. Consider the sub-structure comprised of $C_s, F_t, F_{t-1}, S$. Since $S \to F_{t-1}$ in $\mathcal{M}$, the edge between $S$ and $F_t$ could only be as $S *\to F_t$ in $\mathcal{M}$, and $C_s, S, F_{t-1}$ does not form an unshielded collider. Hence $S *\to F_t$ could be identified in $M$ by Rule 3 of Zhang [13], which contradicts the possible directed path from $F_t$ to $Y$ across $S$. Hence we construct contradictions when there is an edge $C_1 \to F_{t-1}$.

If the edge between $F_{t-1}$ and $C_1$ is as $F_{t-1} \to C_1$, we note that $C_1$ and $F_{t-1}$ are symmetrical on $F_t$ as well as $X$ and $C_s$ are symmetrical on $F_t$. Hence similar to the process above we could conclude $S \to C_1$ in $\mathcal{M}$. And it evidently hold that there cannot be a bi-directed edge $F_s \leftrightarrow C_1, 0 \le s \le t-1$, otherwise there is an active path $X \leftrightarrow F_1 \leftrightarrow \cdots \leftrightarrow F_2 \leftrightarrow C_1, \cdots, Y$ relative to $(\mathbf{D}, X)$, which contradicts the fact that $(\mathbf{D}, X)$ m-separates all the generalized back-door paths. By the ancestral property, if $F_{t-2}$ is adjacent to $C_1$, then the edge is as $F_{t-2} *\to C_1$, thus the edge is $F_{t-2} \to C_1$ since it cannot be bi-directed. Similarly, we could prove that there must be a variable between $X$ and $F_t$ that is not adjacent to $C_1$. If there is $X \to C_1$, it contradicts $\mathbf{D} \cap \mathrm{De}(X, \mathcal{M}) = \emptyset$. Suppose the first variable $F_k$ from $X$ to $F_t$ that is not adjacent to $C_1$. By considering the minimal collider path $F_k *\to F_{k+1} \leftrightarrow \cdots \leftrightarrow F_t \leftrightarrow C$ as the proof process above, where $F_k$ is not adjacent to $C_1, S \to C_1$ and $C_1 \leftrightarrow F_t$ is always identified in $M$, thus there is $S *\to F_t$ in $M$, which contradicts the possible directed path. Combining the conditions that there is an edge $F_{t-1} \to C_1$ and there is an edge $F_{t-1} \leftarrow C_1$, we conclude that $F_{t-1}$ cannot be adjacent to $C_1$ in $\mathcal{M}$.

**A.4.a.2. There does not exist an active path as $\mathcal{L}_2$ in $\mathcal{M}$.**

**A.4.a.2.1 If an active path as $\mathcal{L}_2$ exists, then $S_1$ and $F_{t-1}$ are adjacent in $\mathcal{M}$.**

At first, we consider the situation that $S_1$ and $F_{t-1}$ are not adjacent. Since there is an unshielded collider $F_{t-1} \leftrightarrow F_t \leftarrow *S_1$, we could identify the two arrowheads here. And we could see that there must be a sub-

structure $F_{t-1}*\to S \leftarrow\!*S_1$ in $\mathcal{M}$, otherwise the mark at $F_t$ on the edge between $F_t$ and $S$ could be identified to be arrowhead by Rule 3 of Zhang [13]. Next, we consider the edge between $S$ and $F_{t-1}$. Since $S$ is located at one minimal possible directed path from $F_t$ to $Y$ in $M$, and the edge between $F_{t-1}$ and $S$ is as $F_{t-1}*\to S$, by A.2 we see that there exists a bi-directed edge between $S$ and some variable among $X, F_1\cdots, F_{t-1}$ in $\mathcal{M}$. Without loss of generality, we suppose such variable by $F_s$.

Next we consider the edge $S \leftarrow\!*S_1$. If the edge is directed, then it holds that there is a collider path $X \leftrightarrow F_1\cdots F_s \leftrightarrow S \leftarrow S_1$, which concludes that $S_1 \in (\mathbf{D}, X)$, which contradicts the condition that the sub-path from $F_t$ to $C$ does not go through the variables in $(\mathbf{D}, X)$. Hence the edge between $S$ and $S_1$ could only be $S \leftrightarrow S_1$. If $S_2$ and $S$ are not adjacent, then $S, S_1, S_2$ forms an unshielded collider, so that $S*\to S_1 \leftarrow\!*S_2$ could be identified in $M$. Since the minimal path, $S_1 \to F_t$ could also be identified. According to the ancestral property, we could identify $S*\to F_t$ in $M$, which contradicts the condition that $S$ is located at the minimal possible directed path from $F_t$ to $Y$ in $M$. Hence $S_2$ and $S$ are adjacent. And there is $S_2*\to S$ by ancestral property.

We could repeat the process above for $S_3, S_4\cdots$ until for $S_k$. Similar to the proof above, we see that the edge between $S_k$ and $S$ is $S \leftarrow\!*S_k$. In this case, there is a collider path $X \leftrightarrow F_1\cdots F_s \leftrightarrow S \leftarrow\!*S_k$, and we also know $S_k$ is an ancestor of $Y$ since it is an ancestor of $C$ that is a variable from $(\mathbf{D}, X)$. Hence we conclude that $S_k \in (\mathbf{D}, X)$, which contradicts the condition that the sub-path from $F_t$ to $C$ does not go through the variables in $(\mathbf{D}, X)$.

Hence we see that if a path such as $\mathcal{L}_2$ exists, it is impossible that $S_1$ and $F_{t-1}$ are not adjacent in $\mathcal{M}$.

**A.4.a.2.2 If an active path as $\mathcal{L}_2$ exists, then $S_1$ and $F_{t-1}$ cannot be adjacent in $\mathcal{M}$.**

Here we consider the situation that $S_1$ and $F_{t-1}$ are adjacent. We discuss all possible situations of this edge.

If the edge between them is $S_1 \to F_{t-1}$, the path $X \leftrightarrow F_1 \leftrightarrow \cdots \leftrightarrow F_{t-1} \leftarrow S_1 \leftarrow S_2\cdots Y$ is active relative to $(\mathbf{D}, X)$, which contradicts with the condition that $(\mathbf{D}, X)$ could m-separate all the generalized back-door paths relative to $(G, W)$.

If the edge between them is $S_1 \leftarrow F_{t-1}$, the ancestral property is violated since $F_{t-1} \to S_1 \to F_t \leftrightarrow F_{t-1}$, which constructs a contradiction.

If the edge between them is $S_1 \leftrightarrow F_{t-1}$ in $\mathcal{M}$, there is a collider path $X \leftrightarrow F_1 \leftrightarrow \cdots \leftrightarrow F_{t-1} \leftrightarrow S_1$. We consider the edge between $S$ and $S_1$. Since $S \in (\mathbf{D}, X)$, the edge between $S$ and $S_1$ cannot be as $S_1 \to S$, otherwise $S_1$ is an ancestor of $Y$ thus $S_1 \in (\mathbf{D}, X)$. Hence the edge between $S_1$ and $S$ is as $S*\to S_1$. If $S_2$ and $S$ are not adjacent, then $S, S_1, S_2$ forms an unshielded collider, so that $S*\to S_1 \leftarrow\!*S_2$ could be identified in $M$. Since the minimal path, $S_1 \to F_t$ could also be identified. According to the ancestral property, we could identify $S*\to F_t$ in $M$, which contradicts the condition that $S$ is located at the minimal possible directed path from $F_t$ to $Y$ in $M$. Also, if $F_{t-1}$ and $S_2$ are not adjacent, then $S_2*\to S_1 \leftarrow\!*F_{t-1}$ could be identified in $M$ since they form an unshielded collider. We could also identify $S_2*\to S_1$ thus identify $S_1 \to F_t$. We note that in $M$ there is a sub-structure comprised of $F_{t-1}, S, S_1, S_2$ where $F_{t-1}*\to S_1 \leftarrow\!*S_2$ and $F_{t-1}$ is not adjacent to $S_2$. We discuss the marks at $S$ in this sub-structure. If it is not as $F_{t-1}*\to S \leftarrow\!*S_2$ in $M$, then by Rule 3 of Zhang [13] there is $S*\to S_1$ in $M$. Then by Rule 2 of Zhang [13] there is $S*\to F_t$ in $M$, which contradicts the condition. If it is as $F_{t-1}*\to S \leftarrow\!*S_2$ in $M$, by A.2.2 there must be a variable $F_s, 1 \le s \le t-1$ such that $F_s \leftrightarrow S$. Since $S_2$ is not adjacent to $F_t$ in $M$, $S_2, S, F_t$ is thus unshielded. And because $S \leftarrow\!*S_2$ in $M$, the mark at $S$ on the edge between $F_t$ and $S$ can be identified. To guarantee that $S$ is located at one minimal directed path from $F_t$ to $Y$, the mark at $S$ could only be arrowhead in $M$. We consider such $\mathcal{M}$ consistent to $M$. The edge between $S$ and $S_1$ could only be $S \leftrightarrow S_1$. The reason is, if it is $S \leftarrow\!*S_1$, there is a path $X \leftrightarrow F_1 \leftrightarrow \cdots \leftrightarrow F_s \leftrightarrow S \leftarrow S_1 \leftarrow S_2\cdots Y$ active relative to $\mathbf{D}$, which contradicts the fact that $\mathbf{D}$ m-separates all the generalized back-door path. Since $S_2$ is adjacent to $S$, similarly we could prove that $S_2 \leftrightarrow S$. Repeat the similar process for $S_3, S_4, \cdots, S_k$. We conclude $S_k \in \mathbf{D}$, which contradicts the condition.

Hence $S_2$ and $F_{t-1}$ are adjacent. Due to the ancestral property, the edge between $S_2$ and $F_{t-1}$ must be $F_{t-1} \leftarrow\!*S_2$. If the edge is $F_{t-1} \leftarrow S_2$, similar to the previous proof, the path $X \leftrightarrow F_1 \leftrightarrow \cdots \leftrightarrow F_{t-1} \leftarrow S_2\cdots Y$ are active relative to $(\mathbf{D}, X)$, which contradicts with the condition that $(\mathbf{D}, X)$ could m-separate all the generalized back-door paths. Hence the edge between $F_{t-1}$ and $S_2$ could only be $F_{t-1} \leftrightarrow S_2$. Also similar to the previous proof, the edge between $S$ and $S_2$ must be as $S*\to S_2$, otherwise $S_2 \in \mathbf{D}$ if $S \leftarrow S_2$ since there is a collider path $X \leftrightarrow F_1 \leftrightarrow \cdots \leftrightarrow F_{t-1} \leftrightarrow S$.

We could repeat the above process for $S_3, S_4\cdots$ until for $S_k$. Similar to the proof above, we see $S_k$ and $F_{t-1}$ must be adjacent in $\mathcal{M}$, and the edge between them is $S_k \leftrightarrow F_{t-1}$. However, we know that $S_k$ is an ancestor of $C$, where $C \in (\mathbf{D}, X)$. Hence $S_k$ is an ancestor of $Y$, so that $S_k \in (\mathbf{D}, X)$, which contradicts the condition that the sub-path from $F_t$ to $C$ does not go through the variables in $(\mathbf{D}, X)$.

Hence we see that if a path such as $\mathcal{L}_2$ exists, $S_1$ and $F_{t-1}$ cannot be adjacent in $\mathcal{M}$. Combining A.4.a.2.1 and A.4.a.2.2, we conclude that there does not exist a path like $\mathcal{L}_2$ in $\mathcal{M}$.

**A.4.a.3. There does not exist an active path like $\mathcal{L}_3$ in $\mathcal{M}$.**

**A.4.a.3.1 If an active path as $\mathcal{L}_3$ exists, then $S_1$ and $F_{t-1}$ are adjacent in $\mathcal{M}$.** Similar to the part A.4.a.2.1, we could prove that there exists an bi-directed edge between $S$ and some variable among $X, F_1 \cdots, F_{t-1}$ in $\mathcal{M}$. Without loss of generality, we suppose such variable by $F_s$. And similarly we also conclude that there must be bi-directed edges between $S$ and $S_i, 1 \le i \le S_k$. We discuss the edge between $S$ and $S_{k+1}$ next. If the edge is like $S \leftarrow\!\ast S_{k+1}$, there is a collider path $X \leftrightarrow F_1 \cdots F_s \leftrightarrow S \leftarrow S_{k+1}$, and we also know $S_{k+1}$ is an ancestor of $Y$ since it is an ancestor of $C$ that is a variable from $\mathbf{D}$. Hence we conclude that $S_{k+1} \in \mathbf{D}$, which contradicts the condition that the sub-path from $F_t$ to $C$ does not go through the variables in $(\mathbf{D}, X)$.

Hence, the edge between $S$ and $S_{k+1}$ could only be $S \to S_{k+1}$. In this case, there forms a sub-structure $S_{k+1} \leftrightarrow S_k \leftrightarrow S \leftrightarrow S_{k-1}$, $S_k \to S_{k-1}$, and $S \to S_{k+1}$, which is an inducing path. To satisfy the maximal property, there is a bi-directed edge between $S_{k-1}$ and $S_{k+1}$, which contradicts the condition that $F_t \leftarrow S_1 \leftarrow \cdots \leftarrow S_k \leftrightarrow S_{k+1} \to \cdots \to C$ is a minimal path.

Hence we see that if a path such as $\mathcal{L}_3$ exists, it is impossible that $S_1$ and $F_{t-1}$ are not adjacent in $\mathcal{M}$.

**A.4.a.3.2 If an active path as $\mathcal{L}_3$ exists, then $S_1$ and $F_{t-1}$ cannot be adjacent in $\mathcal{M}$.**

If $F_{t-1}$ and $S_1$ are adjacent, similar to the proof of A.4.a.2.2, we could prove that the edge between them are $F_{t-1} \leftrightarrow S_1$. And we could prove that the edge between $F_{t-1}$ and $S_i, 1 \le i \le S_k$ are bi-directed, and $F_{t-1}$ and $S_{k+1}$ are adjacent. We discuss the edge between $F_{t-1}$ and $S_{k+1}$ next. If edge is $S_{k+1} \to F_{t-1}$, the path $X \leftrightarrow F_1 \leftrightarrow \cdots \leftrightarrow F_{t-1} \leftarrow S_{k+1} \cdots Y$ are active relative to $(\mathbf{D}, X)$, which contradicts with the condition that $(\mathcal{D}, X)$ could m-separate all the generalized back-door paths. If the edge is $S_{k+1} \leftrightarrow F_{t-1}$, $S_{k+1} \in \mathbf{D}$ since it is an ancestor of the variable in $\mathbf{D}$, which contradicts the condition that the sub-path from $F_t$ to $C$ does not go through the variables in $(\mathbf{D}, X)$. If the edge is $S_{k+1} \to F_{t-1}$, we notice that there forms a sub-structure comprised of $S_{k+1} \leftrightarrow S_k \leftrightarrow F_{t-1} \leftrightarrow S_{k-1}$, $S_k \to S_{k-1}$, and $F_{t-1} \to S_{k+1}$. To satisfy the maximal property, there is a bi-directed edge between $S_{k-1}$ and $S_{k+1}$, which contradicts the condition that $F_t \leftarrow S_1 \leftarrow \cdots \leftarrow S_k \leftrightarrow S_{k+1} \to \cdots \to C$ is a minimal path.

Hence we see that if a path such as $\mathcal{L}_3$ exists, $S_1$ and $F_{t-1}$ cannot be adjacent in $\mathcal{M}$. Combining A.4.a.3.1 and A.4.a.3.2, we conclude that there does not exist a path as $\mathcal{L}_3$ in $\mathcal{M}$.

Till now, we have concluded that for any $F_t \in \mathbf{K}$ that has distance 1 to $(\mathbf{D}, X)$, all the paths from $F_t$ to $Y$ with colliders are m-separated by $(\mathbf{D}, X)$ in any one legal $\mathcal{M}$. Then we prove the induction.

**A.4.b. If for any $F_t \in \mathbf{K}$ that has distance $k - 1 \ge 1$ to $(\mathbf{D}, X)$ in $\mathcal{M}$ it holds that $F_t \perp Y | \mathbf{D}, X$, then for any $F_t \in \mathbf{K}$ that has distance $k$ to $(\mathbf{D}, X)$ in $\mathcal{M}$, all the paths from $F_t$ to $Y$ with colliders are m-separated by $(\mathbf{D}, X)$.**

Denote the minimal possible directed path from $F_t$ to $Y$ where the distance between $F_t$ and $(\mathbf{D}, X)$ is $k$ by $F_t, S, \cdots, Y$. It is trivial to prove that $S$ and $F_{t-1}$ are adjacent. We first prove that $S \in \mathbf{K}$. If there is an edge as $F_{t-1} \leftarrow\!\ast S$ in $\mathcal{M}$, then $S \in \mathbf{K}$. If the edge is as $F_{t-1} \to S$, by A.2.2 there is at least one variable $F_s, 0 \le s \le t - 2$ such that $F_s \leftrightarrow S$, thus we also conclude $S \in \mathbf{K}$. The distance between $S$ and $(\mathbf{D}, X)$ is $k - 1$. By the inductive hypothesis, it holds that $S \perp Y | \mathbf{D}, X$ in $\mathcal{M}$.

It is easy to see that $S \to F_t$, otherwise $S$ must be an ancestor of $Y$ due the that fact that $S$ is located at the minimal possible directed path from $F_t$ to $Y$ in $M$. And because $S \in \mathbf{K}$, it holds that $S \in (\mathbf{D}, X)$. However, the distance between $S$ and $(\mathbf{D}, X)$ is $k - 1 \ge 1$, there is a contradiction.

Suppose an active path from $F_t$ to $Y$ with colliders $F_t, S_1, \cdots, S_m, S_{m+1}, \cdots, Y$ in $\mathcal{M}$. Considering the corresponding augmented path with $S \to F_t$, i.e. $S \to F_t, S_1, \cdots, S_m, S_{m+1}, \cdots, Y$. Since $S \perp Y | \mathbf{D}, X$ in $\mathcal{M}$, and the sub-path from $F_t$ to $Y$ is active relative to $(\mathbf{D}, X)$ in $\mathcal{M}$, the edge between $S_1$ and $F_t$ can only be $S_1 \ast\!\to F_t$ in $\mathcal{M}$, i.e., there is a collider at $F_t$.

We first present a supporting result in A.4.b.1.

**A.4.b.1. If $S_m$ is adjacent to $S$ in $\mathcal{M}$, then the marks at $S_m$ on the edge between $S_m$ and $S$ and the edge between $S_m$ and $S_{m-1}$ are distinct, and the edge between $S_m$ and $S_{m+1}$ is as $S_m \leftarrow\!\ast S_{m+1}$.**

Since $S \perp Y | \mathbf{D}, X$ in $\mathcal{M}$, the path $S, S_m, S_{m+1}, \cdots Y$ is m-separated by $(\mathbf{D}, X)$ in $\mathcal{M}$. Denote the path $F_t, S_1, \cdots, S_m, S_{m+1}, \cdots, Y$ by $\mathcal{L}_1$ and $S, S_m, S_{m+1}, \cdots Y$ by $\mathcal{L}_2$. If the edge between $S_m$ and $S_{m+1}$ is as $S_m \to S_{m+1}$, then $S_m$ is not a collider in both paths $\mathcal{L}_1$ and $\mathcal{L}_2$. In this case if $S_m \in (\mathbf{D}, X)$ in $\mathcal{M}$, then $\mathcal{L}_1$ is m-separated by $(\mathbf{D}, X)$ since $S_m$ is a non-collider in the path, which contradicts the active path. If $S_m \notin (\mathbf{D}, X)$ in $\mathcal{M}$, then the sub-path between $S_m$ and $Y$ is m-separated by $(\mathbf{D}, X)$ since $\mathcal{L}_2$ is m-separated by $(\mathbf{D}, X)$ in $\mathcal{M}$. Thus $\mathcal{L}_1$ is m-separated by $(\mathbf{D}, X)$, which contradicts the active path.

Similarly, if the marks at $S_m$ on the edge between $S_m$ and $S$ and the edge between $S_m$ and $S_{m-1}$ are the same, we could conclude $\mathcal{L}_1$ is m-separated by $(\mathbf{D}, X)$ in $M$ by discussing the conditions $S_m \in (\mathbf{D}, X)$ and $S_m \notin (\mathbf{D}, X)$ in $\mathcal{M}$, which contradicts the condition.

**A.4.b.2. For any an active path $F_t, S_1, \cdots, Y$ relative to $(\mathbf{D}, X)$ with colliders in $\mathcal{M}$, the edge between $S$ and $S_1$ in $\mathcal{M}$ is as $S \ast\!\to S_1$.**

Suppose $S_1 \rightarrow S$ in $\mathcal{M}$. Since $S \rightarrow F_t$, the edge between $F_t$ and $S_1$ is as $S_1 \rightarrow F_t$. In this case, $S_1$ is adjacent to $S$ and the marks at $S_1$ on the edge between $S_1$ and $S$ and the edge between $S_1$ and $F_t$ are the same, which contradicts the result in A.4.b.1. Hence the edge is as $S\ast\rightarrow S_1$. By A.4.b.1 again, there is an edge $S_1 \rightarrow F_t$ in $\mathcal{M}$.

**A.4.b.3. For any $F_t \in \mathbf{K}$ that has distance $k$ to $(\mathbf{D}, X)$ in $\mathcal{M}$, all the paths from $F_t$ to $Y$ with colliders are m-separated by $(\mathbf{D}, X)$.**

Similar to A.4.a, we consider the collider closest to $F_t$ in the active path from $F_t$ to $Y$ with colliders. By A.1, since the collider is an ancestor of $Y$ in $\mathcal{M}$ (this collider belongs to $(\mathbf{D}, X)$ so that the path with this collider could be active relative to $(\mathbf{D}, X)$), there exist some active minimal paths without colliders from $F_t$ to this collider relative to $(\mathbf{D}, X)$ in $\mathcal{M}$. Without loss of generality, we suppose the active minimal path $F_t \leftarrow S_1 \cdots S_{m-1}\ast\rightarrow S_m \leftarrow\ast S_{m+1} \cdots Y, m \geq 2$ ($S_{m+1}$ could be $Y$), where $S_m$ is the collider that is nearest to $F_t$ in the path. For simplification, we denote the path by $\mathcal{L}_1$. By A.4.b.1, the mark at $S_1$ on the edge between $S_1$ and $S_2$ is an arrowhead.

If $S_2$ and $S$ are not adjacent, then there is an unshielded collider $S\ast\rightarrow S_1 \leftarrow\ast S_2$, hence the two arrowheads at $S_1$ could be identified in $M$. And because the path is minimal, $F_t \leftarrow S_1$ could be identified in $M$. In this case we could identify there is an arrowhead at $F_t$ on the edge between $F_t$ and $S$ by Rule 2 of Zhang [13], which contradicts the condition that the path $F_t, S, \cdots, Y$ is a minimal possible directed path from $F_t$ to $Y$. Hence $S_2$ and $S$ are adjacent.

By A.4.b.1, the edge between $S_3$ and $S_2$ is as $S_3\ast\rightarrow S_2$. Since $S_2$ is not a collider in $\mathcal{L}_1$, the edge between $S_2$ and $S_1$ could only be $S_2 \rightarrow S_1$. By A.4.b.1 again, the edge between $S$ and $S_2$ is as $S\ast\rightarrow S_2$. For the sake of satisfying ancestral property, the edge could only be $S \leftrightarrow S_2$ considering $S \leftrightarrow S_1 \leftarrow S_2$. In addition, $S_3$ is adjacent to $S$, otherwise there is an unshielded collider $S_3, S_2, S$ thus the two arrowheads at $S_2$ could be identified in $M$. Thus $S_2 \rightarrow S_1$ is identified by Rule 1 of Zhang [13] and $S_2 \leftrightarrow S$ is identified by Rule 4 of Zhang [13]. Thus $S \rightarrow F_t$ is further identified by Rule 1 of Zhang [13], which contradicts the condition that the path $F_t, S, \cdots, Y$ is a minimal possible directed path from $F_t$ to $Y$. Hence $S_3$ and $S$ are adjacent. We repeat the process above and for all variable $S_i, 0 \leq i \leq m - 2$ ($S_0 = F_t$), there are edges $S_{i+1} \rightarrow S_i$ and $S \leftrightarrow S_i$ in $\mathcal{M}$, and $S_{i+2}$ is adjacent to $S$.

By A.4.b.1, the edge between $S_m$ and $S_{m-1}$ is $S_m\ast\rightarrow S_{m-1}$. Since $S_m$ is a collider, the edge is $S_m \leftrightarrow S_{m-1}$. By A.4.b.1 again, the edge between $S_m$ and $S$ is $S_m \rightarrow S$.

We discuss the distinct value attained by $m$. If $m > 2$, it is easy to construct a contradiction. Since $S_m$ is not adjacent to $S_{m-2}$ and there are edges $S_m \rightarrow S \leftrightarrow S_{m-2}$. They form an unshielded collider so that $S_m\ast\rightarrow S$ could be identified in $M$, thus $S \rightarrow F_t$ is identified by Rule 1 of Zhang [13], which contradicts the condition that the path $F_t, S, \cdots, Y$ is a minimal possible directed path from $F_t$ to $Y$.

The proof for $m = 2$ is a bit complex. The reason is that the subpath $S_{m-1}, S_m, S_{m+1}, \cdots, Y$ (here $m = 2$, we still use $m$ for generality) is not necessarily minimal. In the following we present the proof in this case.

We first prove $S_{m+1}$ and $S$ are adjacent. Suppose it does not hold. If $S_{m-1}$ and $S_{m+1}$ are not adjacent, $S_{m-1}\ast\rightarrow S_m \leftarrow\ast S_{m+1}$ form an unshielded collider, thus the two arrowheads at $S_m$ could be identified in $M$. In this case we could identify $S_m \rightarrow S$ in $M$ by Rule 1 of Zhang [13], since $S_m$ is not adjacent to $F_t$ since the path $F_t \leftarrow S_1 \cdots \ast\rightarrow S_m$ is minimal, we could identify $S \rightarrow F_t$ in $M$, which contradicts the condition that the path $F_t, S, \cdots, Y$ is a minimal possible directed path from $F_t$ to $Y$. Then consider the situation that $S_{m-1}$ and $S_{m+1}$ are adjacent. Note that $S_{m-1} \notin (\mathbf{D}, X)$ since the path $F_t \leftarrow S_1 \cdots \leftarrow S_{m-1} \leftrightarrow S_m \leftarrow\ast S_{m+1} \cdots Y$ is active relative to $(\mathbf{D}, X)$ in $\mathcal{M}$. If the edge between $S_{m-1}$ and $S_{m+1}$ is as $S_{m-1} \rightarrow S_{m+1}$, there is an edge $S_m \leftrightarrow S_{m+1}$ and a path $S \leftrightarrow S_{m-1} \rightarrow S_{m+1}, \cdots, Y$. Since $F_t \leftarrow S_1 \cdots \ast\rightarrow S_m \leftrightarrow S_{m+1} \cdots Y$ is active relative to $(\mathbf{D}, X)$ in $\mathcal{M}$, the sub-path $S_{m+1}, \cdots, Y$ is active relative to $(\mathbf{D}, X)$ in $\mathcal{M}$. And because $S_{m-1} \notin (\mathbf{D}, X)$, the path $S \leftrightarrow S_{m-1} \rightarrow S_{m+1}, \cdots, Y$ is active relative to $(\mathbf{D}, X)$ in $\mathcal{M}$, which implies $S \not\perp Y | \mathbf{D}, X$ and contradicts with the condition. Hence the edge between $S_{m-1}$ and $S_{m+1}$ is as $S_{m-1} \leftarrow\ast S_{m+1}$. In this case, there is an unshielded collider $S \leftrightarrow S_{m-1} \leftarrow\ast S_{m+1}$, thus could identify the two arrowheads at $S_{m-1}$ in $M$. And since there is no unshielded collider $S_{m+1}\ast\rightarrow S_m \leftarrow\ast S$ in $M$ since the edge between $S$ and $S_m$ is as $S \leftarrow S_m$ in $M$, we could identify $S_m\ast\rightarrow S_{m-1}$ in $M$ by Rule 3 of Zhang [13], thus identify $S_{m-1} \rightarrow S_{m-2} \rightarrow \cdots \rightarrow F_t$ by Rule 1 of Zhang [13] and $S_{m-1} \leftrightarrow S \leftrightarrow S_{m-2}$ by Rule 4 of Zhang [13]. If $m = 2$, then we identify $S\ast\rightarrow F_t$ by Rule 2 of Zhang [13]. If $m > 2$, then we identify $S \rightarrow F_t$ by Rule 1 of Zhang [13] since $S_{m-1} \rightarrow S$ and $S_{m-1}$ is not adjacent to $F_t$. Both of them contradict the condition that the path $F_t, S, \cdots, Y$ is a minimal possible directed path from $F_t$ to $Y$. Hence there is always a contradiction if $S$ is not adjacent to $S_{m+1}$.

We consider the situation that $S$ is adjacent to $S_{m+1}$. If the edge is as $S\ast\rightarrow S_{m+1}$, there is an edge $S_{m+1} \rightarrow S_m$ by A.4.b.1. This is against the ancestral property since $S_m \rightarrow S$. Hence the edge between $S$ and $S_{m+1}$ could only be as $S \leftarrow S_{m+1}$. And by A.4.b.1 there is an edge $S_{m+1} \leftrightarrow S_m$. And the edge between $S_{m+2}$ and $S_{m+1}$ is as $S_{m+2}\ast\rightarrow S_{m+1}$. Similar to this process, we could prove if $S_j, j \geq m + 1$ is adjacent to $S$, then $S_j \rightarrow S, S_j \leftrightarrow S_{j-1}$, and $S_{j+1}\ast\rightarrow S_j$. If all the variables between $S_{m+1}$ and $Y$ are adjacent to $S$, it holds that $S \not\perp Y | \mathbf{D}, X$, which contradicts the condition. Hence there is at least one variable between $S_{m+1}$ and $Y$ that

is not adjacent to $S$. Suppose the variable that is nearest to $S_{m+1}$ and not adjacent to $S$ in the path $S_n$. That is, there is a path $S \leftrightarrow S_m \leftrightarrow S_{m+1} \leftrightarrow \cdots, S_n, n \geq m + 2$, where $S_{m+1}, S_{m+2}, \cdots, S_{n-1}$ is a parent of $S$ and $S_n$ is not adjacent to $S$. That is, $S_n, S_{n-1}, \cdots, S_m, S$ is a discriminating path for $S_m$ in $\mathcal{M}$.

Note again that there is a collider path from $S_n$ to $S$ in $\mathcal{M}$. And all the variables between $S_{n-1}$ and $S_1$ are parents of $S$ in $\mathcal{M}$. Here evidently $S_1$ is located at the minimal collider path between $S$ and $S_n$. By Lemma 18 (see the detailed proof process), we could identify all the colliders in the minimal collider path from $S_n$ to $S$ by FCI algorithm. Hence there is $S_2 \leftrightarrow S_1 \leftrightarrow S$ in $M$. Since $F_t$ and $S_1$ are not adjacent, we could identify $S_1 \rightarrow F_t$ by Rule 1 of Zhang [13] and identify $S\ast\!\!\rightarrow F_t$ by Rule 2 of Zhang [13], which contradicts the condition that the path $F_t, S, \cdots, Y$ is a minimal possible directed path from $F_t$ to $Y$. Hence we conclude that for all the paths from $F_t$ to $Y$, they are m-separated by $(\mathbf{D}, X)$.

Hence, we conclude that for any $F_t \in \mathbf{K}$ that has distance $k$ to $(\mathbf{D}, X)$ in $\mathcal{M}$, all the paths from $F_t$ to $Y$ with colliders are m-separated by $(\mathbf{D}, X)$. The mathematical induction completes. Hence we prove that there do not exist active paths relative to $(\mathbf{D}, X)$ from $F_t$ to $Y$ with colliders in any legal MAG $\mathcal{M}$. The part A.4 completes.

Combining A.3 and A.4, we conclude that there is no an active path relative to $(\mathbf{D}, X)$ from $F_t$ to $Y$ in any legal MAG $\mathcal{M}$. That is, there is not a legal MAG $\mathcal{M}$ such that $F_t \not\perp Y | \mathbf{D}, X$ in $\mathcal{M}$. Hence we conclude that $\mathbf{K} \perp Y | \mathbf{D}, X$.

**2. In any an MAG $\mathcal{M}$ consistent to $M$, if $\mathbf{K} \perp Y | \mathbf{D}, X$, then $\mathbf{D} \subseteq \text{PD-SEP}(X, Y, M)$.**

For the sake of contradiction, we suppose there is a minimal collider path $X \leftrightarrow F_1 \leftrightarrow \cdots \leftrightarrow F_t \leftarrow\ast F_{t+1}$, where $F_t \in \mathbf{K}$, $F_{t+1} \notin \text{PD-SEP}(X, Y, M)$ and $F_{t+1}$ belongs to $\mathbf{D}$. If it never happens, it is concluded trivially that $\mathbf{D} \subseteq \text{PD-SEP}(X, Y, M)$, which we leave for the readers. There is no need to worry that there is a minimal collider path between $X$ and $F_{t+1}$ which is not across a variable $F_t \in \text{PD-SEP}(X, Y, M) \backslash \text{Anc}(Y, M)$, because in this case it concludes that $F_{t+1} \in \mathbf{K}$, which contradicts the conditions.

By the definition of $\mathbf{D}$, $F_t$ and $F_{t+1}$ are ancestors of $Y$ in $\mathcal{M}$. Suppose $F_t \rightarrow S_1 \rightarrow \cdots \rightarrow S_p \rightarrow Y$ the directed path from $F_t$ to $Y$. Since the collider path $X \leftrightarrow F_1 \leftrightarrow \cdots \leftrightarrow F_t \leftarrow\ast F_{t+1}$ is minimal, by Lemma 18 (see the detailed proof process) we could identify all the colliders in this path. Hence $F_{t-1}$ is adjacent to $S_1$, otherwise the tail at $F_t$ on the edge between $F_t$ and $S$ is identified in $M$, which contradicts the condition $F_t \in \mathbf{K}$. Similarly, $F_{t+1}$ and $S_1$ are adjacent. By ancestral property, the edge between $F_{t-1}$ and $S_1$ is as $F_{t-1}\ast\!\!\rightarrow S_1$ in $\mathcal{M}$. Since $F_{t+1}\ast\!\!\rightarrow F_t$ and $F_t \rightarrow S_1$, there is $F_{t+1}\ast\!\!\rightarrow S_1$ in $\mathcal{M}$.

Then we prove that there exist some variable $F_s, 0 \leq s \leq t - 1$ where $F_0 = X$ such that $F_s \leftrightarrow S_1$. Otherwise, for the edge between $F_{t-1}$ and $S_1$, the edge could only be $F_{t-1} \rightarrow S_1$. And we consider the edge between $F_{t-2}$ and $S_1$ further. If they are not adjacent, there is a discriminating path $F_{t-2} \leftrightarrow F_{t-1} \leftrightarrow F_t \rightarrow S_1$ with $F_{t-1} \rightarrow S_1$ in $M$, thus we could identify $F_t \rightarrow S_1$ in $M$. Hence there is an edge between $F_{t-2}$ and $S_1$. By the ancestral property, the edge could only be $F_{t-2}\ast\!\!\rightarrow S_1$. Since we suppose no bi-directed edges between any $F_s$ and $S_1$, the edge could only be $F_{t-2} \rightarrow S_1$. Repeat the process and we have $X \rightarrow S_1$. However, in this case $F_t$ is a critical variable, contradicting the condition.

Suppose $F_{s_1}$ is the variable with a bi-directed edge with $S_1$ that is nearest to $F_t$ in the collider path. It is easy to see that the collider path $X \leftrightarrow F_1 \leftrightarrow \cdots \leftrightarrow F_{s_1} \leftrightarrow S_1 \leftarrow\ast F_{t+1}$ is also a minimal collider path. Thus $S_1$ belongs to $\mathbf{D}$. If $S_1 \in \text{Anc}(Y, M)$, there is a collider path $X \leftrightarrow F_1 \leftrightarrow \cdots \leftrightarrow F_{s_1} \leftrightarrow S_1 \leftarrow\ast F_{t+1}$ where each variable between $X$ and $F_{t+1}$ are ancestors of $Y$ in $M$, hence we identify $F_{t+1} \in \mathbf{D}$ in $M$ or $F_{t+1} \in \text{PD-SEP}(X, Y, M)$, neither contradicts the condition. If $S_1 \notin \text{Anc}(Y, M)$, we see $S_1 \in \text{PD-SEP}(X, Y, M)$. In this case, there is a collider path $X \leftrightarrow F_1 \leftrightarrow \cdots \leftrightarrow F_{s_1} \leftrightarrow S_1 \leftarrow\ast F_{t+1}$ in $M$ where $S_1 \in \text{PD-SEP}(X, Y, M)$. Note here $S_1$ and $F_t$ are symmetrical in the sense that they have the same property but $S_1$ is closer to $Y$ in the directed path to $Y$. In this case, we see $S_1$ as original $F_t$ and discuss $S_2$. Similarly, if there is not a contradiction, there must be some variable $F_{s_2}$ with a bi-directed edge with $S_2$, and $S_2 \notin \text{Anc}(Y, M)$. Repeat this process for $S_3, S_4, \cdots, Y$, we could identify that there is some variable $F_{s_t}$ with a bi-directed edge with $Y$. In this case, there is a path $X \leftrightarrow F_1 \leftrightarrow \cdots \leftrightarrow F_{s_t} \leftrightarrow Y$, which is active relative to $\mathbf{D}$, contradicting the condition that $\mathbf{D}$ m-separates all generalized back-door paths from $X$ to $Y$. Hence, we conclude that there cannot be a variable $F_{t+1} \in \mathbf{D} \backslash \text{PD-SEP}(X, Y, M)$. Hence we conclude if $\mathbf{K} \perp Y | \mathbf{D}, X$, then $\mathbf{D} \subseteq \text{PD-SEP}(X, Y, M)$ in any an MAG $\mathcal{M}$ consistent to $M$.

Hence, if there is no critical variable, it holds that $\mathbf{K} \perp Y | \mathbf{D}, X$ and $\mathbf{D} \subseteq \text{PD-SEP}(X, Y, M)$ in any a legal MAG. That is $\text{PD-SEP}(X, Y, M) = \mathbf{D} \cup \mathbf{K}$. Thus the MCS regarding $\text{PD-SEP}(X, Y, M)$ equals to the MCS regarding $\mathbf{D}$. $\qquad\square$

### C.3 Proofs for Lemma 4

**Lemma 4.** *Let $\mathcal{P}$ be a PMAG of MAG $\mathcal{M}$. If $X \in \text{An}(Y, \mathcal{M})$ and $\text{D-SEP}(X, Y, \mathcal{M}_{\underline{X}}) \cap \text{De}(X, \mathcal{M}) = \emptyset$, then the MCS regarding $\text{D-SEP}(X, Y, \mathcal{M}_{\underline{X}})$ in $\mathcal{M}$ and the corresponding local MAG are contained in the output of Algorithm 1.*

*Proof.* In line 2 of Alg. 1, the algorithm enumerates local MAGs with different marks at $X$. Evidently $\mathcal{M}$ is consistent to one local MAG among them. Without loss of generality, suppose $\mathcal{M}$ is consistent to $M^j$. If there is no critical variable (Line 5, Line 6) in $M^j$, as implied by Theorem 3, all the MAGs $\mathcal{M}'$ consistent to $M^j$ have the same MCS regarding D-SEP$(X, Y, \mathcal{M}')$ in respective graph $\mathcal{M}'$, and equal to the MCS regarding PD-SEP$(X, Y, M^j)$. Hence no matter what $\mathcal{M}$ is in the class of MAGs consistent to $M^j$, the MCS regarding D-SEP$(X, Y, \mathcal{M}_{\underline{X}})$ in $\mathcal{M}$ is equal to MCS regarding PD-SEP$(X, Y, M^j)$, which is returned in the output.

If there is a critical variable set $\mathbf{C}^j$ in $M^j$ (Line 3, 4), denote the non-empty set $\mathbf{S}$ defined in Definition 5 of the main paper by $\mathbf{S}$ also, we firstly consider the situation that $\mathbf{C}^j$ contains only one variable $C$. We see by the definition of critical variable that there is a collider path $X \leftrightarrow F_1 \leftrightarrow \cdots \leftrightarrow F_{t-1} \leftrightarrow C$ or $X \leftrightarrow F_1 \leftrightarrow \cdots \leftrightarrow F_{t-1} \leftarrow\!\circ C$. We notice that the edge between $C$ and $S$ belonging to $\mathbf{S}$ is as $C *\!\!\to S$ due to the ancestral property. Hence, we just consider two further local MAGs $M^{j_1}$ and $M^{j_2}$, where $M^{j_1}$ is with the edges $F_{t-1} \leftrightarrow C \leftrightarrow S$ for $\forall S \in \mathbf{S}$, and $M^{j_2}$ is with the edges $C \to S$ for $\forall S \in \mathbf{S}$ and $F_{t-1} \leftrightarrow C$ or $F_{t-1} \leftarrow\!\circ C$ (Line 11). The edge between $F_{t-1}$ and $C$ in $M^{j_2}$ follows the edge in local MAG $M^j$. Note that here we rule out the local MAGs with both $C \to S$ for some $S \in \mathbf{S}$ and $C \leftrightarrow S$ for the other $S \in \mathbf{S}$. The reason is if there is some $S_1$ belonging to $\mathbf{S}$ such that $C \to S_1$ and another $S_2$ belonging to $\mathbf{S}$ such that $C \leftrightarrow S_2$ in some MAG $\mathcal{M}'$, it holds that $S_2 \in$ D-SEP$(X, Y, \mathcal{M}'_{\underline{X}})$. We prove $S_2 \in$ D-SEP$(X, Y, \mathcal{M}'_{\underline{X}})$ in the following. (1) It is evident that there is a collider path $X \leftrightarrow F_1 \leftrightarrow \cdots \leftrightarrow F_{t-1} \leftrightarrow C \leftrightarrow S_2$ and each variable except for $S_2$ on it is ancestor of $X$ or $Y$ in $\mathcal{M}'_{\underline{X}}$; (2) $S_2$ is also an ancestor of $Y$ due to the fact that $S_2$ is localed at a minimal possible directed path from $C$ and $Y$. When there is an arrowhead at $S_2$ on the edge between $C$ and $S_2$, $S_2$ must be an ancestor of $Y$ in the path to prevent from generating unshielded colliders. Hence, we conclude $S_2 \in$ D-SEP$(X, Y, \mathcal{M}'_{\underline{X}})$. In addition, $S_2 \in$ De$(X, \mathcal{M}')$ according to the condition (2) of the definition of critical variables. That is $S_2 \in$ De$(X, \mathcal{M}') \cap$ D-SEP$(X, Y, \mathcal{M}'_{\underline{X}})$, which contradicts the condition.

Therefore, we only consider local MAG $M^{j_1}$ and $M^{j_2}$. Similarly, when there are more than one variable in $\mathbf{C}^j$, for each $C \in \mathbf{C}^j$, since we only consider two situations $C \leftrightarrow S$ for $\forall S \in \mathbf{S}$ and $C \to S$ for $\forall S \in \mathbf{S}$, we consider $2^{|\mathbf{C}^j|}$ situations in total, where $|\mathbf{C}^j|$ denotes the set size of $\mathbf{C}^j$. Given a subset $\mathbf{C} \subseteq \mathbf{C}^j$, there possibly exists a local MAG $M^{j_1}$ in which $C_1 \to S$ and $C_2 \leftrightarrow S$ for $\forall C_1 \in \mathbf{C}, C_2 \in \mathbf{C}^j \backslash \mathbf{C}$, and $\forall S \in \mathbf{S}$. That is reflected by Line 11. For all the MAGs $\mathcal{M}'$ consistent to $M^{j_1}$, it holds that $\mathbf{C} \subseteq$ D-SEP$(X, Y, \mathcal{M}_{\underline{X}})$ and $\mathbf{C}^j \backslash \mathbf{C} \cap$ D-SEP$(X, Y, \mathcal{M}_{\underline{X}}) = \emptyset$. Combining with the algorithm to find MCS, it is easy to see that all the variables in $\mathbf{C}$ belong to MCS, while all the variables in $\mathbf{C}^j \backslash \mathbf{C}$ do not belong to MCS. Since there are $2^{|\mathbf{C}^j|}$ possible $\mathbf{C} \subseteq \mathbf{C}^j$, we could obtain $2^{|\mathbf{C}^j|}$ new local MAGs based on $M^j$ in which the causal effects (or MCSs) are different from the others. If there is a critical variable in a new local MAG $M^{j_k}$ for example, the remaining part is same as before, which is in Line 12-15.

Hence for each local MAG $M^j$ of $X$, all the considered local MAGs have covered the whole space of MAGs where GBC does not fail to identify $P(Y|do(X))$. Hence no matter which MAG consistent to $M^j$ is the true MAG, we could find a local MAG consistent to it and the corresponding MCS regarding D-SEP$(X, Y, \mathcal{M}_{\underline{X}})$ as long as GBC does not fail to identify $P(Y|do(X))$ in $\mathcal{M}$. By enumerating all $M^j$, we can find all MCSs regarding D-SEP$(X, Y, \mathcal{M}_{\underline{X}})$ in all MAG $\mathcal{M}$ consistent to $\mathcal{P}$. $\qquad \square$

*Remark.* A pity here is that there is no guarantee for the existence of MAGs consistent to each considered local MAG. However, this does not impact the soundness of the proposed method that the MCS in the true MAG could be returned.

# Appendix D Proofs for the Results in Section 3.4

In this section, we present the proof for the propositions in "Learning marks and purity matrix by interventional data" in order.

**Proposition 5.** *If $P(Y|do(X)) = P(Y)$, the marks at $X$ are arrowheads in all the minimal possible directed paths from $X$ to $Y$ in a partially mixed ancestral graph.*

*Proof.* If there is a tail at $X$ on a minimal possible directed path, the path begins from a directed edge out of $X$. Such a path must be a directed path from $X$ to $Y$ in order to avoid the generation of new unshielded colliders, otherwise there will be an arrowhead pointing to $X$ on the minimal possible directed path in $\mathcal{P}$, which contradicts the fact that the path is a minimal possible directed path from $X$ to $Y$. While such a directed path implies that $X$ is an ancestor of $Y$, contradicting $P(Y|do(X)) = P(Y)$. $\qquad \square$

**Proposition 6.** *In situation* (3)*, let $\mathbf{T}$ denote all variables adjacent to $X$ in the minimal possible directed paths from $X$ to $Y$. For $T \in \mathbf{T}$, if for $\forall V \in \mathbf{T} \backslash T$, it holds either $T \notin$ Adj$(V, \mathcal{P})$ or there is a variable $S \notin$ Adj$(V, \mathcal{P})$*

*such that there is a collider path $X \circ\!\!\rightarrow T \leftrightarrow \cdots \leftarrow\!\!*S$ and every vertex except $S$ on the path is a parent of $V$, then $X \rightarrow T$.*

*Proof.* For the sake of contradiction we assume a variable $T$ meets the condition in the proposition and $X \in \mathrm{An}(Y, \mathcal{M})$ but the edge is $X \leftarrow\!\!*T$ in $\mathcal{M}$. For $V \in \mathbf{T} \backslash T$, if $V$ is not adjacent to $T$, the mark at $X$ in the edge of $V$ and $X$ is evidently tail otherwise there is an unshielded collider $V *\!\!\rightarrow X \leftarrow\!\!*T$ in $\mathcal{P}$ and contradicts the premise that $T$ is in the minimal possible directed path from $X$ to $Y$. And we know the edge $X \rightarrow V$ is visible thus pure by definition 8 of Zhang [11]. If $T$ is adjacent to $V$ but there is a variable $S$ not adjacent to $V$ such that there is a collider path $X \circ\!\!\rightarrow T \leftrightarrow \cdots \leftarrow\!\!*S$ and every vertex except for $S$ on the path is a parent of $V$, if the mark at $X$ in the edge of $T$ and $X$ is arrowhead, it concludes that the edge $X \rightarrow V$ is visible thus pure by Definition 8 of Zhang [11]. Hence, we see that all directed edge out of $X$ in the minimal possible directed path from $X$ to $Y$ are visible. In this case GBC does not fail to identify $P(Y|do(X))$, which contradicts the condition. $\square$

**Proposition 7.** *Rule 11 is sound.*

*Proof.* If it is $d \leftarrow c$, the edge between $b$ and $d$ is $b \rightarrow d$. Hence the edge between $a$ and $d$ is as $a *\!\!\rightarrow d$ by Lemma A.1 of Zhang [13]. Hence there is an unshielded collider $a *\!\!\rightarrow d \leftarrow c$, which contradicts the condition.

If the edge is $d \leftrightarrow c$, we could prove that the only possible structure where $a, d, c$ do not form an unshielded collider is comprised of $a \leftrightarrow b \leftrightarrow d \leftrightarrow c$, $b \rightarrow c$, and $d \rightarrow a$. In this case there is an inducing path $a \leftrightarrow b \leftrightarrow d \leftrightarrow c$, which contradicts the maximal property. $\square$

**Proposition 8.** *In a PMAG $\mathcal{P}$, if there is no minimal possible directed path from $X$ to $Y$, then $X$ cannot be ancestor of $Y$ in any MAG consistent to $\mathcal{P}$. And it holds that $X \notin \mathrm{PossAn}(Y, \mathcal{P})$.*

*Proof.* It is a direct conclusion by the inverse negative proposition of Lemma B.1 of Zhang [13]. Although they do not consider PMAG, it will not influence the result. $\square$

# Appendix E  Proofs for the Results in Section 4

In this section, we present the proofs for the results in Section 4 in order. Then we give a detailed analysis about the computational complexity, which is omitted in the main paper due to the limit of the space.

## E.1  Proofs

**Theorem 9.** Given the observational distribution of the observed variables, if there exists a valid generalized back-door set for $(X, Y)$ in the true MAG with the knowledge of the purity of each directed edge, then we can identify this set by only additional data of $Y$ under intervention on $X$.

*Proof.* When GBC does not fail to identify $P(Y|do(X))$, the interventional data accords with situation (1) or situation (2). The result for situation (1) is obvious since the distribution of $Y$ remains unchanged under intervention on $X$, which implies $X$ has no causal effect on $Y$. The result for situation (2) is trivially guaranteed by Prop. 1 and Lemma 4. According to Prop. 1, there exist generalized back-door sets and D-SEP$(X, Y, \mathcal{M}_{\underline{X}})$ is one among those. By Lemma 4, we could find the MCS regarding D-SEP$(X, Y, \mathcal{M}_{\underline{X}})$. Hence we could identify this set which could result in a consistent estimation of the causal effect with interventional data by Eq. 3. $\square$

**Proposition 10.** *Let $\mathcal{M}$ be a complete MAG with $p + 1$ variables $X_1, \cdots, X_p, Y$, where the causal order of the variables except $Y$ is completely random and $Y$ is at the last. Denote the graph obtained by FCI with observational data by $\mathcal{P}$ and intervention variable by $X_i$. And let $M$ be a local MAG of $X_i$ with $p - 1 - k$ tails and $k$ arrowheads at $X_i$. The computational complexity of finding all possible causal effects $P(Y|do(X_i))$ in all the MAGs consistent to $M$ is $O(2^k)$. Further, the computational complexity of finding all causal effects $P(Y|do(X_i))$ in all the MAGs consistent to $\mathcal{P}$ is $O(3^p)$.*

*Proof.* In $M$, there are $p - 1 - k$ tails and $k$ arrowheads at $X_i$. Without loss of generality, we assume the $k$ variables $X_1, \cdots, X_k$ have edges with $X_i$ in which there is an arrowhead at $X_i$ and $p - 1 - k$ variables $X_1', \cdots, X_{p-1-k}'$ have edges with $X_i$ in which there is a tail at $X_i$. At first we find all of $X_1, \cdots, X_k$ are critical variables. This part takes $k * (p - 1 - k)$ complexity (For each variable $X \in X_1, \cdots, X_k$, we judge whether there is some variable $X' \in X_1', \cdots, X_{p-1-k}'$ which is a child of $X$). According to Line 10 in Algorithm 1, we obtain $2^k$ new local MAGs, and in each new local MAG there is no new critical variable, thus we only have 1 calculation in each new MAG. Hence the computation complexity in these new local MAGs is

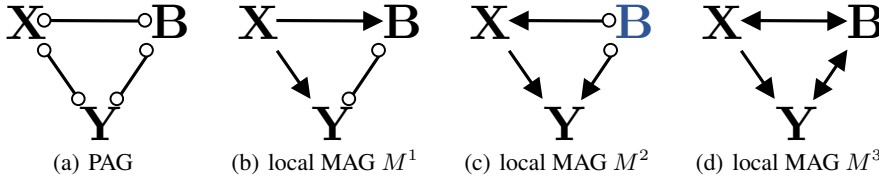

| (a) PAG | (b) local MAG $M^1$ | (c) local MAG $M^2$ | (d) local MAG $M^3$ |

Figure 4: Fig. 4(a) is the PAG. If we intervene on the variable $X$, we can obtain some local MAGs of $X$. Two of them are shown in Fig. 4(b) and Fig. 4(c). Note there is a critical variable $B$ in Fig. 4(c), hence we need to consider the critical marks at $B$ and we can obtain some new local MAGs and Fig. 4(d) is one of them. We can see although Fig. 4(b) and Fig. 4(d) have different marks at $X$, the causal effects of $X$ on $Y$ are the same in the two local MAGs.

$2^k$. The total computation times are $2^k + k * (p - 1 - k) \sim O(2^k)$. When considering the whole process in Algorithm 1, denote the total computation complexity by $T(p)$, it holds

$$
\begin{aligned}
T(p) &= \sum_{k=0}^{p-1} C_{p-1}^k (2^k + k * (p - 1 - k)), \\
&= \sum_{k=0}^{p-1} C_{p-1}^k (2^k) + \sum_{k=0}^{p-1} C_{p-1}^k (k * (p - 1 - k)), \\
&= 3^{p-1} + p(p-1)2^{p-2} \sim O(3^p).
\end{aligned}
$$

$\square$

### E.2 Computational complexity analysis

In the setting of Prop. 10, let $\mathbf{S}$ denote the set of variable that has an edge with an arrowhead at $X_i$. Since the graph is complete, for any subset $\mathbf{S}_1$ of $\mathbf{S}$, we could construct an MAG $\mathcal{M}$ based on $M$ such that MCS in $\mathcal{M}$ is $\mathbf{S}_1$. We show the construction process in the following. Note that the graph is complete, if the constructed graph is ancestral, then it is an MAG since any two variables are adjacent. And for the same reason there are no discriminating paths in the graph. Thus there is no need to worry that there forms a new discriminating path in the constructed graph, which makes the graph entail the conditional independence that is not entailed in the original PAG $\mathcal{P}$. We divide the variables in the local MAG $M$ into three classes. The first class is $\mathbf{S}_1$. The second class is $\mathbf{S}_2 = \mathbf{S} \backslash \mathbf{S}_1$. The third class is comprised of the variables that have an edge with a tail at $X_i$, denoted by $\mathbf{S}_3$. $Y$ belongs to $\mathbf{S}_3$ since it is the descendant of $X_i$. Evidently $\mathbf{S}_1$, $\mathbf{S}_2$, and $\mathbf{S}_3$ cover all the variables in $\mathcal{P}$ except for $X_i$. We orient the subgraph of each class into a DAG, respectively. Note when we orient the subgraph of the third classes, we set $Y$ as the variable at the last of causal order. It could be achieved since the subgraph is complete. For the edges between the variables $A$ and $B$ from two classes, we orient $A \to B$, if the order of the class of $A$ is less than that of $B$, for example $A$ is in the first class $\mathbf{S}_1$ and $B$ is in the third class $\mathbf{S}_3$. For the edge between $A$ and $X_i$, we mark the circle at $A$ by arrowhead if $A \in \mathbf{S}_2 \cup \mathbf{S}_3$, by tail if $A \in \mathbf{S}_1$. Till now we have marked all the circles at $M$ and obtain a graph $\mathcal{M}$, and it is easy to prove this graph is ancestral. According to our discussions before, we know $\mathcal{M}$ is an MAG. And it is not hard to prove that MCS in $\mathcal{M}$ is $\mathbf{S}_1$, because for $\forall S_1 \in \mathbf{S}_1$, there is an edge between $S_1$ and $Y$, which cannot be m-separated by other variables, and for $\forall S_2 \in \mathbf{S}_2$, $S_2$ is not a variable in D-SEP$(X, Y, \mathcal{M}_{\underline{X}})$. Since there are $2^k$ subsets of $\mathbf{S}_1$, there are $2^k$ MCSs, in other word, $2^k$ causal effects in all the MAGs consistent to $M$.

When finding all possible causal effects in $\mathcal{P}$, the complexity gap between our method and the lower bound (i.e. $2^{p-1}$, the number of possible causal effects in all MAGs consistent to $\mathcal{P}$) is from two parts. One is in the search of the critical variables, which results in that we calculate $2^k + k * (p - 1 - k)$ times, which are $k * (p - 1 - k)$ larger than the number of causal effects. But this term do not influence the magnitude of the complexity. The other is caused by that different local MAGs may lead to the same causal effect. An example in Fig. 4 is given to illustrate it. There are different marks at $X$ in Fig. 4(b) and Fig. 4(d), but the causal effects of $X$ on $Y$ in the two graphs are the same. While such a causal effect is calculated by our method for twice when we consider the local MAG $M^1$ and local MAG $M^3$ separately. Such kind of repeated calculations leads to that the complexity of our method is $O(3^{p-1})$ but not $O(2^{p-1})$.

Next, we present a rough estimate about the complexity of the local algorithm (Algorithm 4.2) of Malinsky and Spirtes [16]. In the complete graph, all variables are possible-D-SEP , which is as Definition 4.4 of Malinsky and Spirtes [16]. Hence the search space of MAG is $3^{\frac{p(p+1)}{2}}$, where $\frac{p(p+1)}{2}$ is the number of the edges and $3$ implies there are three kinds for each edge $\to, \leftarrow, \leftrightarrow$. If we enumerate the MAG by brute force, i.e. we consider each combination of circle, then for each searched MAG, we need an extra $p^2 * \frac{p(p-1)}{2}$ complexity [15]

to verify whether it is consistent to $\mathcal{P}$. Since $\mathcal{M}$ is obtained from $\mathcal{P}$, all the non-circle marks at $\mathcal{P}$ are still in $\mathcal{M}$. The only remaining part is to verify whether the enumerated MAG entails the same conditional independence with $\mathcal{P}$. A direct method is to generate an MAG $\mathcal{M}'$ based on PAG by Theorem 2 of Zhang [13] as a representative of $\mathcal{P}$. And then we judge whether $\mathcal{M}$ and $\mathcal{M}'$ are Markov equivalent by the proposed criterion for judging Markov equivalence [15]. If as the process proposed by Malinsky and Spirtes [16], we adopt the method of Zhang and Spirtes [24] to evaluate the Markov equivalence, it is hard to estimate the complexity accurately. However, note that the three rules in Lemma 1 of Zhang and Spirtes [24] are needed to be judged in each step of transformation. This step needs a complexity of $p$ at least (it is a very rough estimate. $p$ is not necessarily enough but just a loose lower bound). Hence, we need a complexity $L_0(p)$ satisfying $p * 3^{\frac{p(p+1)}{2}} \leq L_0(p) \leq \frac{p^3(p-1)}{2} 3^{\frac{p(p+1)}{2}}$ to find all MAGs in total. Next, for each searched MAG $\mathcal{M}$, we will judge whether D-SEP$(X, \mathcal{M}_{\underline{X}}) \cap \text{De}(X, \mathcal{M}) = \emptyset$, which also takes at least a complexity of $p$. Hence the total complexity $L(p)$ is $p^2 * 3^{\frac{p(p+1)}{2}} \leq L(p) \leq \frac{p^4(p-1)}{2} * 3^{\frac{p(p+1)}{2}} = O(3^{p^2})$. Hence the complexity is $p^2 * 3^{\frac{p(p+1)}{2}}$ at least.

## Appendix F   Running Example

In this part, we provide a running example to show the procedure of the proposed method in Fig. 5. The true MAG and the PAG obtained by FCI algorithm are in Fig. 5(a) and Fig. 5(b). No causal effect on $Y$ is identified in $\mathcal{P}$. To identify these causal effects, we introduce interventions. Since there is the largest number of circles at $X$, our algorithm selects $X$ to intervene and collects the interventional data of $Y$, i.e., $P(Y|do(X))$. We could see $X \in \text{An}(Y, \mathcal{M})$ by $P(Y|do(X)) \neq P(Y)$, we thus need to further find all MCSs in the MAGs consistent to $\mathcal{P}$. According to Line 2 of Alg. 1 we obtain the local MAGs with different marks combinations at $X$. Fig. 5(c) to Fig. 5(g) show five of them in which $X \in \text{An}(Y, \mathcal{M})$. There are critical variables on the last four local MAGs, thus the first condition in Theorem 3 is violated for them and we need to call Function CRITICAL. Here we show the detailed further process for local MAG in Fig. 5(g). $T$ is the critical variable so that there are two elements $\emptyset$ and $\{T\}$ in the power set of $T$. According to Line 11 of Alg. 1, we obtain Fig. 5(h) and Fig. 5(i) according to the element of the power set, respectively. And it is easy to see that there is no critical variable in these two graphs. In other words, we know that the MCSs in each graph of Fig. 5(h) and Fig. 5(i) are the same, respectively. And we obtain that the MCSs in them are $\emptyset$ and $\{T\}$. By Line 15 of Alg. 1, we add (Fig. 5(h), $\emptyset$) and (Fig. 5(i), $\{T\}$) to $L$. Similar, when we consider Fig. 5(c), we add (Fig. 5(c), $\emptyset$) to $L$. When we consider Fig. 5(d), we add (Fig. 5(j), $\emptyset$) and (Fig. 5(k), $\{S\}$) to $L$. When we consider Fig. 5(e), we add (Fig. 5(l), $\emptyset$) and (Fig. 5(m), $\{A\}$) to $L$. When we consider Fig. 5(f), we add (Fig. 5(n), $\emptyset$) and (Fig. 5(o), $\{S\}$) to $L$. According to the discussions before, we have

$$L = \{(\text{Fig. 5(h)}, \emptyset),$$
$$(\text{Fig. 5(i)}, \{T\}),$$
$$(\text{Fig. 5(c)}, \emptyset),$$
$$(\text{Fig. 5(j)}, \emptyset),$$
$$(\text{Fig. 5(k)}, \{S\}),$$
$$(\text{Fig. 5(l)}, \emptyset),$$
$$(\text{Fig. 5(m)}, \{A\}),$$
$$(\text{Fig. 5(n)}, \emptyset),$$
$$(\text{Fig. 5(o)}, \{S\})\}.$$

By Eq. 3 in the main paper, we infer MCS$^\star = \{T\}$, hence we learn the graph as Fig. 5(i). By applying the eleven rules to update the PMAG by background knowledge, we learn the graph and purity matrix as Fig. 5(n). $T$ is the only variable whose causal effect is not identified. We thus intervene on $T$ and identify Fig. 5(o). In this task, even if we have the interventional data of full variables, it is hard to discover the structure with once intervention on $X$, for instance by the method of Kocaoglu et al. [25] in Fig. 5(p).

## Appendix G   Related Work

In Pearl's causality framework, many criteria are proposed to identify causal effects with the prior knowledge of causal graph [26, 27, 28, 29]. Considering sometimes such causal knowledge is not available, some work [11, 9, 30, 31, 12, 32, 16, 10, 33, 34] identifies the causal effects based on different kinds of partial graphs learned according to the conditional independence of the variables. For example, a sufficient and necessary graphical characterization for the identifiability of causal effect by *generalized back-door criterion* is given by Maathuis et al. [12]. And Jaber et al. [33] proposed a complete algorithm to identify causal effects in PAGs. However, due to the insufficient information contained in PAG, some causal effects cannot be identified, which could have been identified if we have more structure knowledge.

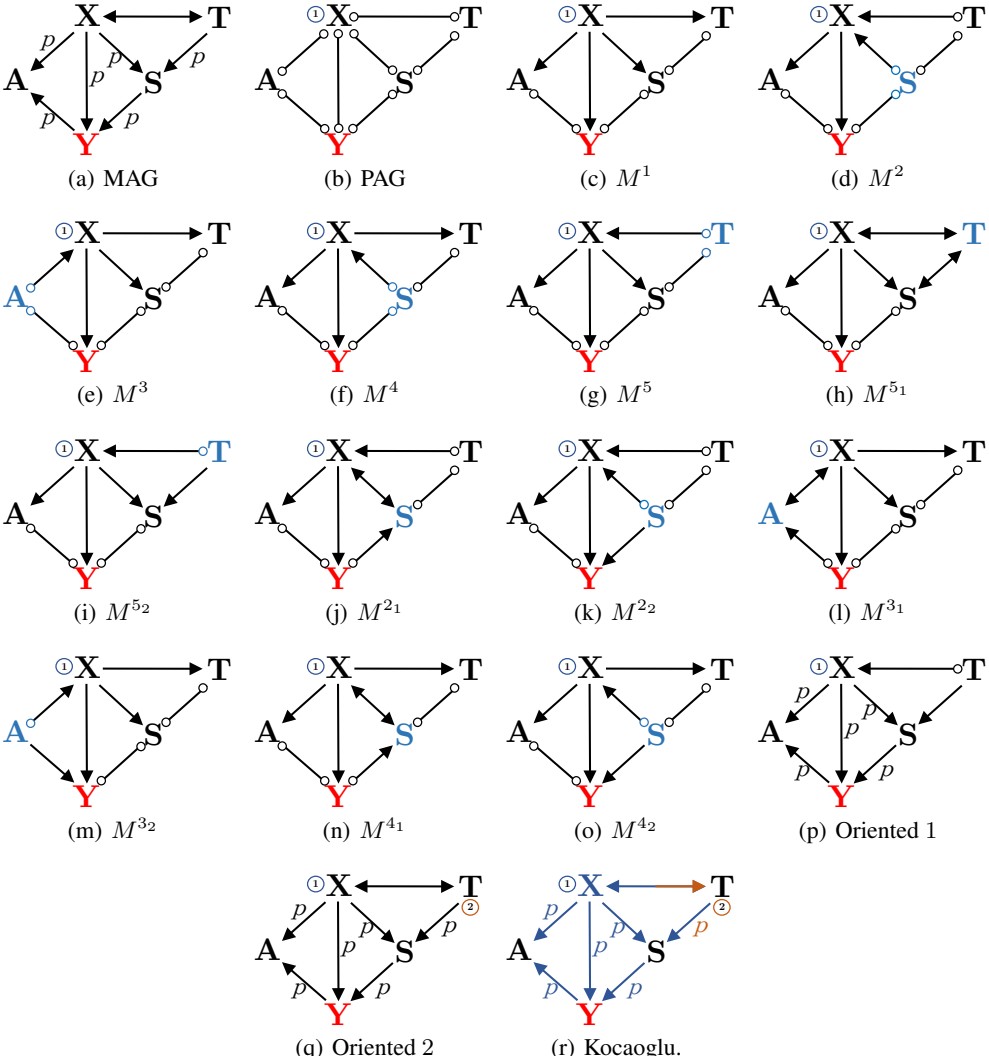

Figure 5: Fig. 5(a) is the true MAG and Fig. 5(b) is the PAG by FCI algorithm. Fig. 5(c)∼ Fig. 5(g) denote five local MAGs. Critical variables are colored by blue. By the interventional data, we learn the graph in Fig. 5(p), followed by the learned graph by intervening on $S$ in Fig. 5(q). Fig. 5(r) depicts that of Kocaoglu et al. [25].

Towards learning the causal structure, there are many methods in the literature [8, 35, 5, 36, 6, 37]. However, if there are no further functional assumptions [38, 39], it is hard to discover the whole graph. Some methods achieve it by introducing active interventions [40, 2, 3, 41, 42, 43], and some by utilizing the changing distributions [44, 45, 46]. The proposed method in this paper is different from these classical methods in two aspects. One is although we have the data under active intervention, we could only observe the response variable. The other is that our goal is just target effect identification, where discovering the whole causal structure is not necessary.

When there are latent confounders, ancestral graph is introduced to describe the relation between the observed variables [22]. In this paper, when we adopt the trivial method to enumerate each MAGs consistent to $\mathcal{P}$ by considering all combinations of marks, we need to judge the Markov equivalence of two MAGs. The graphical characterization of the Markov equivalence condition of two maximal ancestral graphs is a fundamental problem. And many methods are proposed towards this problem with less computational complexity [47, 14, 15]. Zhang and Spirtes [24] proposed a method to enumerate the MAGs more efficiently, which could be used to prevent judging the Markov equivalence. Based on the ancestral graph, Kocaoglu et al. [25], Jaber et al. [6] proposed solid causal discovery methods with the fully observed interventional data by utilizing do-calculus reversely.