# OpenReview forum: "Actively Identifying Causal Effects with Latent Variables Given Only Response Variable Observable"
_NeurIPS.cc/2021/Conference — NeurIPS 2021 Poster_

### Official Review · Reviewer_sQJg · 2021-06-28

**Rating:** 5
**Confidence:** 3

**Summary:**

This paper extends a previous work where a method was presented for learning causal effects when some interverventions can be conducted, but such that only the distribution of the response variable is observable after the intervention. This paper extends the mentioned previous work by allowing for possible latent confounders, which requires listing possible alternative structures in an active manner.

**Limitations And Societal Impact:**

There is little on limitations and no discussion on negative impacts. I don't think there are serious ethical impacts, but more discussion could be provided on the limitations.

**Main Review:**

PROS:
- Extending a previously published idea to a situation where there are latent confounders.
- The derivations seem correct.

CONS:
- The motivation for the paper is not very well elaborated. The paper mentions an application in healthcare, where patients are not fully checked after treatment and only their illness is observable by doctors, but this seems overly simplistic, especially because there is no corresponding real data experiment.
- The presentation is a bit hasty. First of all, the main text is not self-standing, but it's necessary to read at least ref [8] and the supplementary to grasp the main idea. Also, some abbreviations are used in the text without defining them first (at least ACI, MCS). Obviously the page limitation restricts what can be put to the main text, but I think it would improve the paper if the main text focused more on motivation, key points, and intuition.
- Novelty compared to ref [8] seems a bit limited.
- Based on experiments the difference of the presented active learning approach to a "random" approach seems small (Fig. 3). Also, the experiments are somewhat limited in general.

**Time Spent Reviewing:**

3

---

> ### Author Response · Authors · 2021-08-10
> **Response to Reviewer sQJg**
>
> Thank you for your reviews. We will improve the paper accordingly.
>
> Motivation: In this paper, we restrict that only response variable is observable under active intervention. However, our method is feasible not only in this scenario. Our techniques could be easily generalized to any setting with partial observation under intervention. As far as we know about the medical data, for example, it is quite common that the collected data is partial observed. Also, our method could be generalized to tackle the causal discovery problem with both observational data and passive interventional data (as opposed to active intervention). Hence, we believe that our setting is reasonable and the proposed method is helpful in many real tasks.
>
> Novelty compared to ref [8] of the main paper: Our method follows [8] in framework. However, we want to point out that handling latent confounders is essentially a very hard problem. Many well-known results for DAGs still remain to be explored for MAGs. In this paper, we develop new related theoretical results for MAG. In addition, we propose the graphical characterization to efficiently estimate possible causal effects in a PAG by GBC, which contribute to not only our problem, but also some other problems related to causal effect identification in a PAG.
>
> Experiments: Thank you for your feedback. The name of the comparison method is possibly a bit misleading. We will change the name. In fact, the *only* difference between ''ACIC-random'' approach and ''ACIC’’ approach is that we randomly select the intervention variable instead of selecting the variable with the largest number of circles. The other parts such as learning structure by interventional data and identifying causal effects are totally the same between these two approaches. We will change the name ''ACIC-Random’’ to ``ACIC-simple selection’’ and change the name ''ACIC’’ to ''ACIC-greedy selection’’ in Figure 3. The comparison between them is to verify the effectiveness of the greedy intervention strategy.
>
> No real data: Thank you for your feedback. As pointed out by Reviewer WArj, the contribution is mainly theoretical. There is not existing theory for our problem, and we present a solid method with theoretical guarantee for that. We agree that it will be better if we could verify the effectiveness of the method with real data. However, there is not such a dataset in public, even if in many classical papers taking active interventions into account ([1]...[4] below) or identifying causal effects ([5]...[8] below). In the revised manuscript, we will add a simulated experiment as [9] to simulate the real data to the greatest extent.
>
> [1] Active Invariant Causal Prediction Experiment Selection through Stability.
>
> [2] Active Learning of Causal Networks with Intervention Experiments and Optimal Designs.
>
> [3] Two Optimal Strategies for Active Learning of Causal Models from Interventional Data.
>
> [4] Cost-optimal learning of causal graphs.
>
> [5] Estimating high-dimensional intervention effects from observational data.
>
> [6] Causal discovery from soft interventions with unknown targets: Characterization and learning.
>
> [7] General Identifiability with Arbitrary Surrogate Experiments.
>
> [8] A generalized back-door criterion.
>
> [9] Cost-effectively identifying causal effects when only response variable is observable.
>
> We are happy to respond further if there is something unclear.
>
> Best Regards

---

> > ### Comment · Reviewer_sQJg · 2021-08-26
> > **Thank you for the response**
> >
> > Thank you for the response! In the light of that and the discussions with the other reviewers I'm willing to increase my score from 4 to 5. The discussion found room for further improvement in the motivation and presentation, which I encourage the authors to consider (regardless of the outcome).

---

> > > ### Author Response · Authors · 2021-08-26
> > > **Thank you**
> > >
> > > Dear Reviewer sQJg:
> > >
> > > Thank you for your suggestions! We will improve the paper accordingly.
> > >
> > > Best Regards

---

> ### Author Response · Authors · 2021-08-21
> **Dear Reviewer sQJg**
>
> Thank you for your invaluable feedback once again. We were wondering whether our response has addressed your concerns. If you have any additional comments, please let us know, we would be happy to address them.

---

### Official Review · Reviewer_WArj · 2021-07-13

**Rating:** 7
**Confidence:** 3

**Summary:**

Based on the ancestral graph framework, the authors tackle the problem of identifying an interventional distribution $f (Y | do (X))$ by only observing the outcome $Y$ after each possible intervention. This work extended previous work by Wang et al. assuming no latent variables on the graph.

**Ethical Concerns:**

No.

**Limitations And Societal Impact:**

This work is purely theoretical so no immediate potential negative societal impact.

**Main Review:**

As most of us know, the graphical model that best approximates a DAG with latent factors is an Acyclic Mixed Directed Graphs (ADMGs), which preserves not only conditional independence constraints but also dormant conditional independence constraints after one performs fixing operations on the original ADMG. But because the characterization of MEC of an ADMG is still an open problem, most of the current works consider a relaxation of ADMG -- MAG, as in the current submission. Unlike ADMGs, MAGs do not preserve dormant conditional independence constraints; also, the characterization of MEC of MAG has been solved by Richardson and Spirtes (2002).

This paper seems to be an extension of an earlier work by Wang et al. 2020 ICML, in which Wang et al. also consider only observing the outcome after active interventions, but under causal sufficiency assumptions. This paper, however, considered a more difficult problem, for which unmeasured factors/confounders are allowed.

The theoretical development is new but it is not easy to evaluate if the intervention scheme considered in this paper is very practically relevant. To me, the most (or in my opinion, the only) meaningful causal discovery setting is in genomics where large-scale intervention data is available to actually test the results, but in genomics, it is rarely the case that we only get to observe $Y$ after the intervention. As a result, I give a score of 7 to this paper.

Besides the general comment, I have more specific comments or questions regarding the submission:

(1) More recently, the research group of Elias Barenboim has also solved the problem of characterizing the so-called interventional MEC of MAGs when one has mixed observational and interventional data. Though the authors decide to take a different route (i.e. local search), it seems to me that it is entirely possible to characterize the MEC of MAGs when interventional data is available but we only get to observe $Y$ in each intervention. Have the authors considered such a problem? If not, is it by any means helpful for settings similar to what the authors are trying to solve?

(2) In practice, decision variables might not be all observables except $Y$. In this setting, if I am only interested in $Y | do (X_{i})$ for $i$ from a subset of $[p]$, would there be a more efficient algorithm than simply applying ACIC but just focusing on this subset?

(3) Intervening decisions variables one by one also seems restrictive. It is quite reasonable to consider cases where multiple variables are intervened on. If one can intervene in more than one variable, what are the technical challenges of developing theory?

(4) A discussion on the potential generalization of this work to ADMG will be helpful to reach a more general audience of causal inference.

Minor points:
(1) Line 143: I would prefer to add its complete name when the authors first mention MCS.

(2) Line 115: "(identifying failing of GBC is counted)." Does it mean that "failing to identify $Y | do (X)$ by GBC"?

(3) After reading the paper, my understanding is that ACIC is sound but not necessarily complete. I would suggest the authors discuss this issue at the end of the paper and mention it in the introduction.

(4) The paragraph below Prop 1 seems hard to understand. In the second unidentifiable case, do you mean that a generalized backdoor set may not exist?

**Time Spent Reviewing:**

1.5

---

> ### Author Response · Authors · 2021-08-10
> **Response to Reviewer WArj**
>
> Thank you very much for your positive review! We will improve the paper accordingly.
>
> ''it is not easy to evaluate if the intervention scheme considered in this paper is very practically relevant’’: In this paper, although we restrict that only $Y$ is observable under intervention, the techniques could be easily generalized to any partial observational settings where only a part of variables (not only $Y$) are observable under hard intervention. In this paper, we consider an extreme scenario.
>
> The part of comments or questions regarding the submission:
>
> (1) It is an insightful question! In fact, we have ever considered such a characterization in our paper when we read ``Causal Discovery from Soft Interventions with Unknown Targets: Characterization & Learning, NeurIPS 2020``, which is a solid and in-depth paper, and the authors proposed a characterization in Theorem 1. The main reason that we do not present such a characterization is that when latent confounders exist, it is hard to propose a characterization that could directly guide the structure learning in practice. Even if it is as Theorem 1 of ``Causal Discovery from Soft Interventions with Unknown Targets: Characterization & Learning, NeurIPS 2020``, the third condition is hard to judge since it is necessary to consider *all* discriminating paths. And since we do not enumerate all possible MAGs to avoid the huge computation cost, such a characterization that implies which MAGs are equivalent seems not to contribute a lot to our method. On the other hand, essentially, an important step in our method is classifying all the MAGs consistent to a PAG without enumeration, such that the MAGs in one class share the same $P(Y|do(X))$. In another word, the MAGs that are classified to one class belong to an equivalent class given the observational data and the data of $Y$ under intervention on $X$. It could be seen as another kind of characterization.
>
> (2) If $X_i$ is a single variable, then by only collecting $Y$ under intervening on $X_i$ with some value $x_i$, we could identify the generalized-backdoor set for $(X_i,Y)$ if the set exists, which is implied as Theorem 9. Hence, if we only care about $P(Y|do(X_i))$, we could achieve it by once intervention on $X_i$. However, if $X_i$ is a set comprised of more than one variable, we possibly have to learn the structure by ACIC at first, and then search for a valid adjustment set by the criterion of Perković et al. 2015. It is not efficient yet.
>
>
> (3) ACIC heavily relies on Theorem 4.1 of ref [12], where the authors propose a checkable necessary and sufficient criteria for the existence of a generalized back-door set. However, it restricts that we consider a single intervention variable and a single outcome variable. Hence, if we want to generalize ACIC to adapt to the setting where many variables are intervened at the same time, we need a stronger criterion for the existence of a generalized back-door set that could apply to many intervention variables.
>
>
> (4) Thank you for your suggestion. It is very valuable. We will consider it in the future. In fact, if considering ''purity'' from the viewpoint of ADMG, the introduction of ''purity'' is to identify whether there are bi-directed edges between two variables in the corresponding ADMG. To the best of our knowledge, if we identify some directed edge $X_1\rightarrow X_2$ is pure in an MAG or PAG, then we could see that there is no bi-directed edge between them in ADMG. If we identify it is not pure, then we see that there is a bi-directed edge between $X_1$ and $X_2$ in ADMG. In the future, we will take a deeper and more rigorous study about this problem.
>
> The part of minor points:
>
> (1) Thank you! We will add the complete name.
>
> (2) That implies if our method concludes that there does not exist a valid generalized back-door set for $(X_i,Y)$ *even if* we know the true MAG and the purity of each edge, we *see this condition as* that the method has identified $P(Y|do(X_i))$, since in this case it is impossible to identify the causal effect by generalized back-door criterion no matter how many interventions we take.
>
> (3) Thank you for your suggestion. On line 386, we take it as an aspect to improve the method that how to completely update a PAG with background knowledge. Since there is no theoretical guarantee for the completeness of this step, the completeness of ACIC is not guaranteed. We will make it clear in the paper that the method is not yet complete.
>
> (4) Yes. For example, given a PAG $X\circ-\circ Y$, the causal effect of $X$ on $Y$ is unidentifiable due to the missing of the marks at $X$ and $Y$. It belongs to case 1. However, even if we are given an MAG $X\rightarrow Y$ with known purity matrix implying that the edge is not pure, there does not exist a generalized back-door set. It belongs to case 2. For case 1, we learn the unknown marks and possibly identify the causal effects. For case 2, it is not feasible to identify the causal effect by generalized back-door criterion.
>
> We are happy to respond further if there is something unclear.
>
> Best Regards

---

> ### Author Response · Authors · 2021-08-21
> **Dear Reviewer WArj**
>
> Thank you for your invaluable feedback once again. We were wondering whether our response has addressed your concerns. If you have any additional comments, please let us know, we would be happy to address them.

---

> > ### Comment · Reviewer_WArj · 2021-08-24
> > **Thanks**
> >
> > Dear authors,
> >
> > Thank you for your response to my questions. Your response definitely solidifies my relatively positive evaluation on your paper!
> >
> > After pondering for a moment on my concern about the motivation of your paper, I think I might not formulate my question very clearly in my comments. Could you further comment if I'm simply interested in Y | do (X) but after interventions, I might observe some "nuisance" vertices in addition to $Y$, will these additional observables be helpful (e.g. decreasing intervention burden) or is there any paper considering such a situation? It seems that decision-makers might be really interested in balancing between these two situations if there is some trade-off between measuring more variables and performing more interventions.
> >
> > Thanks in advance for your clarification!

---

> > > ### Author Response · Authors · 2021-08-25
> > > **The additional observables will be helpful**
> > >
> > > Dear Review WArj
> > >
> > >   Thank you very much for your valuable question! At first, please allow us to restate the setting you mentioned, to ensure that we did not misunderstand it. The setting is, we want to estimate the distribution of *only* $Y$ under intervention on any possible variable $X_i$ (i.e. the causal effects $P(Y|do(X_i))$ for $i=1,\cdots$). We may observe additional variables in addition to $Y$ under intervention.
> > >
> > >   Our answer about whether the additional observables could be helpful is definitely yes! The additional observables could help it by learning more structure information in the PAG, including discovering more circles and identifying the purity of more edges, which could save some intervention times. Before illustrating this gain in detail, we name some notations. Denote the intervention variable by $X$ and the observed variables under intervention by $(Y,Z)$, where $Z$ is an additional variable (or a variable set). We have $P(Y,Z|do(X=x))$ by the collected data after intervening on $X$ with the value $x$.
> > >
> > >   A trivial method based on ACIC is considering $Y$ and $Z$ respectively. That is, we obtain $P(Y|do(X=x))$ and $P(Z|do(X=x))$ by the interventional data. With $P(Y|do(X=x))$, we identify the generalized back-door set for $(X,Y)$ and some structure information by ACIC. And with $P(Z|do(X=x))$ we could also learn some structure information by ACIC (in this case we see $Z$ as the response variable $Y$ in ACIC.). However, in this way we do not use the interventional data efficiently, since we ignore the joint information of $Y$ and $Z$. A feasible method that takes the joint information into account is to exploit the invariance property under interventions, which is defined in Def 19 and detailed in Theorem 24 and Theorem 30 of "Causal Reasoning with Ancestral Graphs" (https://www.jmlr.org/papers/volume9/zhang08a/zhang08a.pdf). Next we give Example A to illustrate how the trivial method based on ACIC works and Example B to illustrate how to use the invariance property to learn the structure information.
> > >
> > >   Example A: Consider the true MAG $X \rightarrow Z \leftarrow T \rightarrow X\rightarrow Y \leftarrow T$ with all edges pure. By observational data, we have a PAG with circles at each position (due to no v-structures). If we only observe $Y$ under intervention (i.e. we observe $P(Y|do(X=x))$), ACIC could learn the true generalized back-door set for $(X,Y)$ and the structure information including pure $X\rightarrow Y$, $T\circ\rightarrow X$, and $T\rightarrow Y$. The other circles remain. In this discovered structure, $P(Y|do(T))$ and $P(Y|do(Z))$ remain unidentifiable. However, if we could observe $Z$ under intervention additionally (i.e. we also have $P(Z|do(X=x))$), this information helps us further learn the structure $X\rightarrow Z$ and $T\circ\rightarrow Z$ by ACIC. In this discovered structure $P(Y|do(Z))$ is additionally identified. Hence it is not necessary for us to intervene on $Z$. This example demonstrates that more observables help decreasing intervention burden by adopting the trivial method based on ACIC.
> > >
> > >   Example B: Consider the true MAG $X \rightarrow Z \rightarrow Y \leftarrow X$ with all edges pure. By obser. data, we have a PAG with circles at each position. If we only observe $Y$ under intervention, ACIC could learn that the true generalized back-door set for $(X,Y)$ is $\emptyset$, and learn the structure information including pure $X\rightarrow Y$ and $X\circ \rightarrow Z$. In this discovered graph, $P(Y|do(Z))$ is unidentifiable. If we could observe $Z$ under intervention additionaly, this information helps us further learn $Z\rightarrow Y$ and $X\rightarrow Y$ by ACIC. In the discovered structure $P(Y|do(Z))$ is also unidentifiable because we do not know whether $Z\rightarrow Y$ is pure (without latent confounders). In this case, the trivial method based on ACIC fails to decrease the intervention burden since we have to intervene on $Z$ later. If introducing Thm. 24 of "Causal Reasoning with Ancestral Graphs", which presents a necessary and sufficient graphical condition for whether $P(Y|Z)$ is invariant under an intervention on $X$, we could see if the edge $Z\rightarrow Y$ is pure, the three conditions of Thm. 24 are satisfied, thus the invariance holds. While if it is not pure, the second condition is violated (since there is an active path relative to $Z$ that is $X\rightarrow Z\leftarrow L \rightarrow Y$ where $L$ is a latent confounder), thus the invariance does not hold. Hence, since we have $P(Y|Z)$ by the observational data and also have $P_{do(X=x)}(Y|Z)$ by the interventional data, we could identify that $Z\rightarrow Y$ is pure by noticing $P(Y|Z)=P_{do(X=x)}(Y|Z)$, thus identify that {X} is a valid generalized back-door set for $(Z,Y)$  by the learned structure $X \rightarrow Z \rightarrow Y \leftarrow X$ with pure $Z\rightarrow Y$, without another intervention on $Z$.
> > >
> > >   By the two examples above, we show that the additional observations could help reduce the necessary intervention times. If pursuing a general method that is suitable for not only the two toy examples above, the trivial method based on ACIC could be applied directly by respectively considering $P(Y|do(X))$ and $P(Z|do(X))$. However, it may take some difficulties when using the result of invariance, since given a PAG, we *do not* want to enumerate each possible MAGs to evaluate the m-separations in the three graphical conditions in Thm. 24 and Thm.30 of "Causal Reasoning with Ancestral Graphs", which is not favored from the point of complexity. The key here is to avoid the enumeration of all MAGs when evaluating the m-separations. We leave this problem for the future work.
> > >
> > >   We agree that a more practical scenario is that a few variables could be observed simultaneously under intervention. In this paper, in contrast to many classical causal discovery methods exploiting full variables under experiments, we hope to guide readers to pay attention to the scenario that only partial variables are observable. We thus select an extreme scenario, that only the response variable is observed under intervention. How to take full advantage of the intervention data when there are more than one variable observed and how to balance the measurements and interventions are quite valuable and practical problems. We will try to address them in the future work. We sincerely appreciate your constructive question!
> > >
> > > Best Regards

---

> > > > ### Comment · Reviewer_WArj · 2021-08-25
> > > > **Thanks again!**
> > > >
> > > > Thank you so much for your thorough and educational response. In light of the discussion, I am now willing to increase my score from 6 to 7.

---

> > > > > ### Author Response · Authors · 2021-08-26
> > > > > **Re: Thanks again**
> > > > >
> > > > > Dear Review WArj:
> > > > >
> > > > >   Thank you for your response. We notice that your initial score is 6 and you increase it to a more negative score 5 after our rebuttal. Thus we are wondering whether there is still something unclear. If you have any additional comments, please let us know, we would be happy to address them.
> > > > >
> > > > > Best Regards

---

> > > > > > ### Comment · Reviewer_WArj · 2021-08-26
> > > > > > **my bad**
> > > > > >
> > > > > > Dear authors,
> > > > > >
> > > > > > Sorry, it's completely my bad. I was confused about the scoring rule. I meant increasing your score from 6 to 7. Somehow I mistakenly thought the smaller the score is the better.

---

> > > > > > > ### Author Response · Authors · 2021-08-26
> > > > > > > **Thank you**
> > > > > > >
> > > > > > > Dear Review WArj:
> > > > > > >
> > > > > > >   We are lucky to have the insightful discussion with you. Thank you.
> > > > > > >
> > > > > > > Best Regards

---

### Official Review · Reviewer_ot5a · 2021-07-16

**Rating:** 6
**Confidence:** 2

**Summary:**

The paper proposes a method to identify the causal effect of each treatment variable on a response variable under latent confounding. The method starts from a PAG and further learning its structure by using the Generalized Backdoor Criterion (GBC) or resorting to interventional data where only the response variable is observed. The process finishes when all effects have been identified. For this, the authors introduce a graphical condition based on the notions of "local MAG" and "critical variables" that allow them to determine the existence of a minimal conditional set (MCS) that is valid for adjustment for some effect P(Y | do(X)).

**Limitations And Societal Impact:**

I am satisfied with the discussion of the limitations provided in the paper.
I can't foresee any societal impact of the work that needs to be addressed at the moment.

**Main Review:**

The setting considered in the paper combines elements from structural learning and data-driven causal inference. As far as I know, the task of identifying all effects while learning the structure posed in the paper is novel. In particular, in the presence of latent confounding and having observational data plus interventional data where only the outcome is observed as input. Relevant related work seems to be properly cited.

The claims made in the paper seem well supported both by theoretical analysis and experimentation. While the method relies on the GBC, I wonder why the Generalized Adjustment Criterion (Perković et al. 2015) is not a better choice since it is known to be sufficient and complete for adjustment in the context of DAGs, MAGs, CPDAGs, and PAGs. The algorithms are described with enough detail and their complexity has been analyzed.

I find the paper well-written and clear, with a few exceptions. In particular, I find the wording and subsequent explanation of Theorem 9 confusing. What does it mean for a causal effect to be "identifiable by interventions"?
The experimental section is clear and provides a compelling comparison with other methods.

The results presented in the paper address a causal inference setting that could arise in many practical scenarios. Allowing for only the response variable to be observed under intervention provides flexibility compared to other methods that require observation of all non-intervened variables. The method also seems to improve, compared to previous work, in terms of computation complexity and the number of required interventions.

Other comments:
- Consider adding a title to Definition 3.
- Line 180: "GBC dose not fail"
- Line 347: "of first part" (missing article)

**Time Spent Reviewing:**

7

---

> ### Author Response · Authors · 2021-08-10
> **Response to Reviewer ot5a**
>
>
> Thank you very much for your positive review! We will improve the paper accordingly.
>
> 1. Why generalized adjustment criterion (GAC) is not a better choice: It is a very insightful question. Theorem 3.4 of Perković et al. 2015 provides a very strong result. However, it is not an efficient criterion for judging the existence of the adjustment set, as said in ''6 Discussion'' of Perković et al. 2015 that ''it would be desirable to have an easily checkable condition to determine if there exists any adjustment set at all, as done for the generalized back-door criterion for single interventions by Maathuis and Colombo (2015)''. To the best of our knowledge, if adopting the criterion of Perković et al. 2015, it needs to test for any subset of $V\backslash \{X,Y\}$, which takes a very huge cost. That is the main reason that we use GBC.
> In addition, in our paper we just consider the condition that $X$ and $Y$ are single variables, and $X$ is an ancestor of $Y$ (If $X$ is not an ancestor of $Y$, the situation is as (1) in Section 3.4 and we do not use Prop. 1). And we have Assumption 1. In this context, we have not found a counterexample where there does not exist a generalized back-door set but exists a generalized adjustment set. However, we do not have a rigorous proof for it. And we will try to achieve a conclusion in the future.
>
>
> 2. Theorem 9: We will correct it, e.g. ''Given a PAG $\mathcal{P}$ and the observational distribution of the variables, if there exists a valid generalized back-door set for $(X,Y)$ in the true MAG with the knowledge of the purity of each directed edge, then we can identify this set by only additional data of $Y$ under intervention on $X$.''
>
> We are happy to respond further if there is something unclear.
>
> Best Regards

---

> > ### Comment · Reviewer_ot5a · 2021-08-22
> > **Re: Response**
> >
> > Thanks for your answers. Please allow me a follow-up question.
> >
> > While the GAC was defined in Perkovic et al. (2015), didn't [Perkovic (2018)](https://arxiv.org/pdf/1606.06903.pdf) showed (sec. 4.3) how to construct, in linear time, a set that satisfies the criterion whenever any set satisfying the criterion exist?
> > Could you please clarify if this construction is not suitable for your goal in this paper?

---

> > > ### Author Response · Authors · 2021-08-24
> > > **Re:Re: Response**
> > >
> > > Dear Reviewer ot5a：
> > >
> > >   Thank you very much for the careful reviews and the extra opportunity to clarify our method. It is a very valuable question, which helps us improve the paper further. We admit that we did not read the paper "Complete Graphical Characterization and Construction of Adjustment Sets in Markov Equivalence Classes of Ancestral Graphs" (we call it Perkovic (2018) for short in the following) before. We were surprised to find that our previous conjecture that GAC and GBC are equivalent in our setting has been proved by Perkovic (2018). In detail, it is, when we identify $P(Y|do(\mathbf{X}))$ where $\lvert \mathbf{X} \rvert=1$, there is a generalized back-door set relative to $(\mathbf{X},\mathbf{Y})$ if and only if there is a set satisfying the generalized adjustment criterion relative to $(\mathbf{X},\mathbf{Y})$, which is indicated in Theorem 26 and the three following lines after Corollary 27 of Perkovic (2018). Since we consider singleton intervention, (1) "there exists a generalized adjustment set" $\Leftrightarrow$ (2) "there exists a generalized back-door set". And by Thm. 4.1 of the paper "A generalized back-door criterion", we conclude that (2) "there exists a generalized back-door set" $\Leftrightarrow$ (3) "D-SEP is a generalized back-door set". Hence we further conclude "D-SEP is a generalized back-door set" $\Leftrightarrow$ "there exists a generalized adjustment set".
> > >
> > >   After reading (Perkovic 2018) you mentioned, we try to answer your question about whether the construction of (Perkovic 2018) is suitable for our goal. Whenever any set satisfying the GAC criterion exists, it is feasible to construct a valid set by the method you mentioned. However, in our task of actively identifying causal effects, the case that the causal effect of $X$ on $Y$ is *not identifiable in a partial graph to be discovered* is also necessary to be considered. In this case, we need to find all possible causal effects, i.e. all possible generalized adjustment sets. The methods of Perkovic (2018) could not help it directly. (Note that finding all possible generalized adjustment sets in our task is totally different from constructing all adjustment sets by Perkovic (2018). For the former one, $P(Y|do(X))$ is *unidentifiable* by GAC in the given PAG $\mathcal{P}$. If a set is a generalized adjustment set in any an MAG $\mathcal{M}$ consistent to $\mathcal{P}$, then this set is a possible generalized adjustment set. For the latter one, $P(Y|do(X))$ is *identifiable* by GAC in the given PAG $\mathcal{P}$. The task is just to list all valid adjustment sets.)
> > >
> > >
> > >   Next we show how GBC could work. The main reason is that Thm. 4.1 of "A generalized back-door criterion" presents an *explicit* set - D-SEP, which could be as a generalized back-door set. The word "explicit" here is regarding the graph. In our method, when the causal effect of $X$ on $Y$ is not identifiable in the partial graph to be discovered, the step is to first find all local MAGs of $X$ by marking the circles at $X$, and in each local MAG we find all possible $P(Y|do(X))$ *without enumerating all MAGs consistent to the local MAG*. As we show in the main paper, there is an equivalent relationship between (MCS regarding) D-SEP and $P(Y|do(X))$. Hence the task of finding all possible causal effects in a local MAG is converted to the task of finding all (MCSs regarding) D-SEPs in the local MAG. And since D-SEP is an explicit set regarding the graph, as defined in Definition 1 of our paper, the target of finding all (MCSs regarding) D-SEPs could guide us which circles in the local MAG we should consider further, i.e. the different marks (arrowhead or tail) on these circles will lead to MAGs with different (MCSs regarding) D-SEPs, thus we need to discuss the different marks on these circles. Under this guideline, we define PD-SEP and critical variable, and present Thm 3.
> > >
> > >
> > >   After reading (Perkovic 2018), we further think about whether the results of Perkovic (2018) about GAC could be combined with our idea to tackle the problem of finding all possible adjustment sets instead of finding all possible generalized back-door sets. A plausible method is to use Adjust(X,Y,G) as a set instead of D-SEP. In each local MAG, we find Adjust(X,Y,G) and judge whether $Adjust(X,Y)$ m-separates $X$ and $Y$ in $G_{XY}^{pbd}$ as Theorem 7 of Perkovic (2018). However, here we encounter a problem that the judgment of m-separation *depends on* the circles in the partial graph. For example, given a PAG $X\circ-\circ Y \circ-\circ Z \circ-\circ X$, suppose a local MAG as $X -\circ Y \circ-\circ Z \circ-> X$. Here $Z\in Adjust(X,Y,G)$, we cannot judge the m-separation of $X$ and $Y$ given $Adjust(X,Y,G)$ because different marks on the circles at $Z$ will lead to different m-separations. A feasible method is to enumerate the possible marks on the circles at $Z$, for each possible marking of the circles at $Z$, we obtain a new partial graph $G'$ and judge whether $Adjust(X,Y,G')$ m-separates $X$ and $Y$ in $G_{XY}^{'pbd}$. When the graph is large, however, we need to consider carefully about the circles at which variable should be enumerated, to avoid getting into a situation that we enumerate all the circles, which is not favored from the point of complexity. In our method with GBC, the target of finding (MCSs regarding) D-SEP guides us to consider the circles at critical variable. When our target is to find (MCSs regarding) $Adjust(X,Y,G')$, how to avoid enumerating all circles should also be considered. This is the main challenge of using GAC. We will consider it in the future. If it could be addressed, the method right now could be applied to the setting where considering intervening on *more than* one variable $\mathbf{X}$ with GAC simultaneously. In the setting with $\lvert \mathbf{X}\rvert>1$, GAC is a better choice than GBC. Thus our method could be improved further in the setting with many variables intervened simultaneously.
> > >
> > >   We sincerely appreciate the suggestion of the paper. We promise to add the related paper and the discussion about the equivalent relationship between GAC and GBC under our setting in the revision. And we will try to introduce the result of Perkovic (2018) into the current framework in the future.
> > >
> > > Best Regards

---

> ### Author Response · Authors · 2021-08-21
> **Dear Reviewer ot5a**
>
> Thank you for your invaluable feedback once again. We were wondering whether our response has addressed your concerns. If you have any additional comments, please let us know, we would be happy to address them.

---

### Official Review · Reviewer_rzAn · 2021-07-16

**Rating:** 6
**Confidence:** 4

**Summary:**

This paper develops a method for identifying causal effects on a response variable from observational and interventional data, with the restriction that under interventions only the response variable can be measured. The method does not assume away latent confounders. Theoretical justifications and some empirical evaluations of the method are presented .

**Ethical Concerns:**



**Limitations And Societal Impact:**

Adequately addressed.

**Main Review:**

This paper tackles an interesting question and the proposed method has novel elements. However, details of the paper are not easy to follow. At the moment I am puzzled by the following aspects of the work:

1. The paper proposes that if the effect of a variable X on the response variable cannot be identified by the generalized backdoor criterion (GBC) even given the true MAG with all the purity information, "Fail" will be returned for this effect. But why not intervene on X in such a situation to directly estimate the effect? In fact, in Section 3.4, the "Fail" situation is supposed to be "detected" by intervening on X. Then why not use the interventional data there to estimate the effect in question instead of returning "Fail"?

2. It is also unclear to me how those MAGs that do not license identification by GBC can ever be ruled out in the proposed method. For such a MAG, any effect estimated from the interventional data seems to be consistent with it, as the MAG does not entail what the effect is by GBC.

3. The definition of local MAG is given on p. 5 relative to one variable, but Algorithm 1 seems to use local MAGs in a more general sense, which is not clearly defined (as far as I can see).

4. The statement of Theorem 9 is a little curious on two counts. First, as I indicated in point 1 above, the locution "the causal effect of X on Y is identifiable by GBC with the interventional data of Y under intervention on X" sounds odd because the causal effect of X on Y is identifiable with the interventional data of Y under intervention on X, and it is unclear why "identifiable by GBC" is so important. Second, the theorem sounds like a simple tautology: the causal effect is identifiable by GBC if GBC does not fail to identify it.

Minor comments:

i. In Proposition 1, since G is supposed to be an ancestral graph (as opposed to a partial ancestral graph), should "PossDe" be simply "De"?

ii. In Definition 3, the terminology "deterministic marks" is misleading. "Definite marks" or  "determinate marks" sounds better.

iii. Also in Definition 3, a local MAG is typically still a *partial graph*, so not an ancestral graph, right?

iv. In Definition 4, "descendant" or "possible descendant"?

**Time Spent Reviewing:**

5

---

> ### Author Response · Authors · 2021-08-10
> **Response to Reviewer rzAn**
>
> Thanks for your careful reviews. We will improve the paper accordingly.
>
> The ''ancestral graph'' on Line 72, Prop.1, and Def.3 should be ''partial ancestral graph''. We are sorry that this typo may influence the reading. In this paper, we see ancestral graph as a special case of partial ancestral graph.
>
> 1. Why returning failing instead of using the interventional data directly: The reason is twofold: (1). Strictly, $P(Y|do(X_i))$ is a series of distribution under different value $x_i$ that variable $X_i$ could attain. If we want to identify $P(Y|do(X_i))$ by interventional data, we need to intervene on $X_i$ with all possible values $x_i$ and estimate the distribution of $Y$ under $x_i$. If the number of possible value of $X_i$ is large or even unlimited, e.g. $X_i\in 1:10$ or $-1\leq X_i \leq 1$, it is impractical to intervene on each value $x_i$ and estimate corresponding $P(Y|do(X_i=x_i))$. This is the main reason for returning ''failing''. (2). Even if the attainable value for $X_i$ is limited, we possibly estimate $P(Y|do(X_i)) $ by intervening on $X_i$ with each possible value $x_i$. However, from the points of achieving target effect identification, i.e. identifying the causal effect of each variable $X$ on $Y$, such intervention information could not help learn the structure further. Hence we do not consider identifying such effects by a large number of interventions.
>
> 2. How MAGs that do not license identification by GBC are ruled out? If there exists an MAG that license identification by GBC and is consistent to the interventional data, then we rule out the MAGs that do not license identification according to Assumption 1. Let us give an example. Note in our paper the notion ''purity’’ is added to MAG since it influences causal effect identification by GBC. Suppose the true MAG $X\rightarrow Y$ and the edge is pure. By FCI, we have a PAG $X\circ-\circ Y$. By intervening on $X$ and collecting the data of $Y$ under intervention, we could see $X\rightarrow Y$. Since the purity of the edge will influence the identification of $P(Y|do(X))$, we need to judge the purity. If it is pure, it holds that $P(Y|do(X=x))=P(Y|X=x)$. If not pure, it implies that there are some latent confounders between them. There must be some *latent* $\mathbf{Z}$ such that $P(Y|do(X=x))=\int P(\mathbf{Z})P(Y|X=x,\mathbf{Z})d\mathbf{Z}$. According to a common sense that correlation does not equal to causation when there are confounders, we have Assumption 1 to assume that $\int P(\mathbf{Z})P(Y|X=x,\mathbf{Z})d\mathbf{Z}$ cannot equal to $P(Y|X=x)$ numerically in this case. That is, if we find that $P(Y|do(X))$ equals to $P(Y|X=x)$ by the observational data and interventional data, we see that there does not exist a latent confounder $\mathbf{Z}$. Thus the MAG $X\rightarrow Y$ with an impure edge is ruled out. And we conclude the graph should be $X\rightarrow Y$ with a pure edge. For all the MAGs that do not license identification $P(Y|do(X))$ by GBC, we cannot distinguish them from each other by the interventional data.
>
> 3. Thank you for pointing out this error that we did not notice. We will revise the second condition of Def. 3. In fact, we only need to restrict that all marks at $X$ are not circles in local MAGs, but ''*merely* marking circles at $X$ compared to $\mathcal{P}$’’ is not necessary. The theoretical part still holds. In the method, given a PAG, we first get local MAGs by only marking circles at $X$, as shown on Line 243. Then, even if we mark some circles out of $X$ as Algorithm 1, the oriented PAG is also a valid local MAG.
>
> 4. (1). Why "identifiable by GBC" is so important: please see the reply for the first question.
> (2). The theorem sounds like a simple tautology: Thank you. We will correct it, e.g. ''Given a PAG $\mathcal{P}$ and the observational distribution of the variables, if there exists a valid generalized back-door set for $(X,Y)$ in the true MAG with the knowledge of the purity of each directed edge, then we can identify this set by only additional data of $Y$ under intervention on $X$.''
>
> The part of minor comments:
>
> i. It is a typo. Here $G$ is a partial ancestral graph. In fact, this proposition applies to both PAG and MAG, following Theorem 4.1 of [12] with a slight generalization of visibility. To make Prop. 1 concise, we take MAG as a special case of PAG and define $G$ as a partial ancestral graph.
>
> ii. Thank you very much. We will correct it.
>
> iii. Thank you for pointing out the typo. $M$ is a partial ancestral graph!
>
> iv. It is ''Descendant''. $\mathcal{M}$ is an MAG. As footnote 2, we show in Lemma 18 that if $\mathcal{M}$ is consistent to $M$, all the descendants of $X$ in $\mathcal{M}$ are determinate in $M$ even if $\mathcal{M}$ is not known. The key point for the proof is the fact that all marks at $X$ are determinate (arrowheads or tails but not circles) in local MAG $M$.
>
> We are happy to respond further if there is something unclear.
>
> Best Regards

---

> ### Author Response · Authors · 2021-08-21
> **Dear Reviewer rzAn**
>
> Thank you for your invaluable feedback once again. We were wondering whether our response has addressed your concerns. If you have any additional comments, please let us know, we would be happy to address them.

---

> > ### Comment · Reviewer_rzAn · 2021-08-22
> > **Thanks**
> >
> > Thank you for the helpful response, in light of which I decided to increase my score to 6.

---

### Decision · Program_Chairs · 2021-09-27

**Decision:**

Accept (Poster)

**Comment:**

The paper explores a causal setting where we are interested in understanding the impact of possible interventions on a single response variable, using observational data (which might suffer from latent confounding) and limited interventional data. The limit on the interventional data is that we only observe the response variable, and nothing else. This work extends several other papers that have dealt with similar setting, with the important distinction of allowing latent (i.e. hidden) confounding in the observational data.
The paper is a theory paper, and the reviewers agreed that it contains a novel contribution and that it is technically correct.

The main concern of the reviewers was about the applicability of the specific scenario the authors propose; however, the overall decision is that the real theoretical contribution is important enough. Furthermore, I believe it is often difficult to know in advance which learning scenarios or which techniques will end up useful. Since the work is both rigorous and novel, the recommendation is to accept the paper.

The reviewers made some thoughtful suggestions in their reviews and the discussion period, and I encourage the authors to incorporate some of the conclusions in their revised version.